# Fast Pure Exploration via Frank-Wolfe

**Po-An Wang**
KTH Royal Institute of Technology
Stockholm, Sweden
wang9@kth.se

**Ruo-Chun Tzeng**
KTH Royal Institute of Technology
Stockholm, Sweden
rctzeng@kth.se

**Alexandre Proutiere**
EECS and Digital Futures
KTH, Stockholm, Sweden
alepro@kth.se

## Abstract

We study the problem of active pure exploration with fixed confidence in generic stochastic bandit environments. The goal of the learner is to answer a query about the environment with a given level of certainty while minimizing her sampling budget. For this problem, instance-specific lower bounds on the expected sample complexity reveal the optimal proportions of arm draws an Oracle algorithm would apply. These proportions solve an optimization problem whose tractability strongly depends on the structural properties of the environment, but may be instrumental in the design of efficient learning algorithms. We devise Frank-Wolfe-based Sampling (`FWS`), a simple algorithm whose sample complexity matches the lower bounds for a wide class of pure exploration problems. The algorithm is computationally efficient as, to learn and track the optimal proportion of arm draws, it relies on a single iteration of Frank-Wolfe algorithm applied to the lower-bound optimization problem. We apply `FWS` to various pure exploration tasks, including best arm identification in unstructured, thresholded, linear, and Lipschitz bandits. Despite its simplicity, `FWS` is competitive compared to state-of-art algorithms.

## 1 Introduction

Pure exploration in stochastic bandits [24] refers to the task of answering a given question about the reward distributions of the different arms, using as few arm pulls (or samples) as possible. The task may correspond to identifying the best arm [13], the top-$m$ arms [37], all $\epsilon$-good arms [27], a set of arms whose expected rewards exceed a given threshold [26], etc. To reduce the sample complexity of such a task, the learner needs to leverage as much as possible the information available about reward distributions, which typically comes as known structural properties of the set of their expected rewards. Exploiting particular structures (e.g., unimodal, Lipschitz, convex, linear) has been thoroughly studied in the regret minimization setting (see [6], and references therein), but less in the pure exploration framework, where most efforts have focused on linear structures [35, 20, 39, 36, 10, 18, 9].

In this paper, we investigate a generic learning problem proposed in [8] and covering the aforementioned pure exploration tasks with or without structure. Consider $K$ arms whose reward distributions $(\nu_1, \ldots, \nu_K)$ come from a one-dimensional exponential family and are of unknown means $\boldsymbol{\mu} = (\mu_1, \ldots, \mu_K)$. The parameter $\boldsymbol{\mu}$ is known to belong to $\Lambda \subset \mathbb{R}^K$, the set of possible instances. For each $\boldsymbol{\mu} \in \Lambda$, we assume that there is a unique true answer $i^\star(\boldsymbol{\mu})$ that belongs to the finite set $\mathcal{I}$ of possible answers[1] (e.g., for the best arm identification task, $i^\star(\boldsymbol{\mu}) = \arg\max_k \mu_k$). We consider pure

---

[1]Scenarios with several correct answers require a more involved analysis, see [7].

35th Conference on Neural Information Processing Systems (NeurIPS 2021).

exploration tasks in the *fixed confidence* setting where the learner wishes, for any possible $\boldsymbol{\mu} \in \Lambda$, to discover $i^\star(\boldsymbol{\mu})$ with a certain level of confidence $1 - \delta$, for some $\delta \in (0, 1)$. The learner's strategy is defined by (i) an adaptive sampling rule dictating the sequence of arm pulls, (ii) a stopping rule defining $\tau$, the round where, based on the data gathered so far, the learner decides to stop pulling arms, and (iii) a decision rule specifying her answer. The goal is to devise a $\delta$-PAC (it outputs the right answer with probability at least $1 - \delta$ for any $\boldsymbol{\mu} \in \Lambda$) strategy minimizing the expected sample complexity $\mathbb{E}_{\boldsymbol{\mu}}[\tau]$.

Using the same arguments as those used in [13] for classical MAB problems, we may derive a lower bound of the expected sample complexity satisfied by any $\delta$-PAC strategy. This lower bound, whose proof can be found in Appendix B for completeness, is given by $T^\star(\boldsymbol{\mu})\mathrm{kl}(\delta, 1 - \delta)$, where the characteristic time $T^\star(\boldsymbol{\mu})$ is defined through the following optimization problem:

$$T^\star(\boldsymbol{\mu})^{-1} = \sup_{\boldsymbol{\omega} \in \Sigma} \inf_{\boldsymbol{\lambda} \in \mathrm{Alt}(\boldsymbol{\mu})} \sum_{k=1}^{K} \omega_k d(\mu_k, \lambda_k), \tag{1}$$

where $\Sigma$ is the $(K - 1)$-dimensional simplex, $\mathrm{Alt}(\boldsymbol{\mu})$ is the set of *confusing* parameters $\boldsymbol{\lambda} \in \Lambda$ such that $i^\star(\boldsymbol{\mu}) \neq i^\star(\boldsymbol{\lambda})$, $\mathrm{kl}(a, b)$ is the KL divergence between two Bernoulli distributions of means $a$ and $b$, and $d(\mu_k, \lambda_k)$ denotes the KL divergence of arm-$k$ reward distributions under parameters $\boldsymbol{\mu}$ and $\boldsymbol{\lambda}$ . A solution $\boldsymbol{\omega}^\star(\boldsymbol{\mu})$ of (1) can be interpreted as an optimal *allocation*, in the sense that pulling each arm $i$ a proportion of round equal to $\omega_i^\star(\boldsymbol{\mu})$ (in expectation) constitutes an optimal sampling rule.

Most existing algorithms achieving an asymptotically (when $\delta$ goes to 0) minimal sample complexity leverage a Track-and-Stop (TaS) framework [13]. In each round $t$, they plug $\hat{\boldsymbol{\mu}}(t)$ the estimated expected arm rewards in the lower bound optimization problem (1), and track the allocation $w^\star(\hat{\boldsymbol{\mu}}(t))$. As already noticed in [28], the main drawback of the Track-and-Stop framework is that it requires a recurrent access to an Oracle able to solve (1) (actually existing analyses usually assume that the Oracle outputs the exact solution for any $\boldsymbol{\mu}$). (1) is a concave program but can become difficult to solve depending the underlying structure $\Lambda$. Indeed, for complex structures, identifying the most confusing parameters leading to the objective function $\inf_{\boldsymbol{\lambda} \in \mathrm{Alt}(\boldsymbol{\mu})} \sum_{k=1}^{K} \omega_k d(\mu_k, \lambda_k)$ can be hard.

**Contributions.** 1) Instead of solving (1) in each round as in the TaS framework, we propose an online iterative method to approach the optimal allocation of arm pulls. Specifically, we devise Frank-Wolfe-based Sampling (FWS), a computationally efficient algorithm that just relies, in each round, on a single iteration Frank-Wolfe (FW) algorithm applied to (1) instantiated at $\hat{\boldsymbol{\mu}}(t)$.
2) For a wide class of pure exploration problems with or without structure, we derive an upper bound of the expected sample complexity of FWS for any certainty level $\delta$, and show that this bound matches the lower bound $T^\star(\boldsymbol{\mu})\mathrm{kl}(\delta, 1 - \delta)$ asymptotically as $\delta$ goes to 0.
3) We illustrate the performance of FWS on various pure exploration problems, including best arm identification in unstructured, linear, and Lipschitz bandits. In all tested scenarios, and despite its simplicity, FWS matches the performance of the best existing algorithms.

The use of the FW algorithm has been suggested in [13] in the case of best arm identification problem in unstructured bandits. In this case, FW iterations take a very simple and intuitive form (see Example 1 introduced in §3). The corresponding sampling rule is referred to as Best Challenger in [13], and leads to algorithms with remarkably low sample complexity empirically – sometimes lower than that of TaS algorithms solving (1) in each round. So far however, as discussed in [28], the analysis of FW-type sampling rules, and even their convergence, have eluded researchers. Towards the design of FWS algorithm, we devise a simple variant of the FW algorithm that yields a sampling rule whose sample complexity can be analyzed. We confirm the asymptotic optimality of as well as its empirical superiority, not only for the case of best arm identification in unstructured bandits as predicted by [13], but also for a wide class of pure exploration problems. We believe that our analysis also brings interesting solutions to the three important obstacles we needed to tackle to devise and analyze a FW-type sampling rule: (i) the objective function in (1) is not smooth; (ii) its curvature becomes infinite in general close to the boundary of $\Sigma$; and (iii) the estimate $\hat{\boldsymbol{\mu}}(t)$ is evolving and might be far from $\boldsymbol{\mu}$.

## 2 Related Work

Best Arm Identification (BAI) has recently received a lot of attention, either in unstructured bandit problems, see [13, 33], or in problems with various kinds of structure, e.g., linear [35, 20, 39, 36,

10, 18, 9, 31], combinatorial [23, 19, 32], spectral [22], monotone [14], cascading [41]. For BAI in unstructured bandits with fixed confidence, [13] developed the celebrated Track-and-Stop framework leading to algorithms able to asymptotically converge towards the optimal allocation of arm draws, and in turn, to achieve the lowest sample complexity possible in the high confidence regime (as $\delta$ goes to 0). It is possible to apply the TaS framework to specific structures, as this was proposed in [18] for linear bandits. However, for more involved structures, this might become computationally too difficult. Indeed TaS requires the learner to repeatedly solve the optimization problem (1).

The authors of [8] propose and exploit an interpretation of the lower bound optimization problem (1) as the solution of a 2-players game – the $\omega$-player playing the 'sup' and the $\lambda$-player playing the 'inf'. The algorithm presented in [8] combines two zero-regret algorithms applied sequentially by the two players, and converge to an optimal allocation. Interestingly, the algorithm uses the optimism in face of uncertainty principle to remove the need of forced exploration (the $\omega$-player is fed with upper-confidence bounds on her rewards). As shown later, the algorithm does not perform as well as FWS. The applicability of the framework used in [8] remains unclear to us: in [9] and in [19], the authors claim that the framework cannot be applied to linear and combinatorial bandits, respectively.

In [28], the author proposes a solution close to ours. His algorithm, LMA (Lazy Miror Ascent), just runs in each round one iteration of a sub-gradient ascent algorithm applied to (1). Fortunately, the projection step usually involved in such algorithm is simple. Numerically, as illustrated later in the paper, we found that LMA may not be as efficient as TaS or FWS. We could try to explain this by remarking that LMA has similarities with the Exponential Weights algorithm (see Appendix F in [28]), an algorithm designed for adversarial online optimization problem, and may be too conservative in a stochastic setting.

As already mentioned in the introduction, FW-based algorithms for BAI in unstructured bandits have been mentioned first in [13] for their simplicity and good performance. Applying FW as if the objective function was smooth may fail at converging [28] experimentally. We believe that we manage to make, in our algorithm, the minimal modification of the FW-based algorithm so that convergence and asymptotic optimality are guaranteed. Finally note that [2] uses FW in a regret minimization problem but with a smooth objective function.

We conclude this section by mentioning existing works on the FW algorithm when applied to optimizing non-smooth functions. The proposed solutions consist by either smoothing objective function or enlarging the set of differential (this is the second approach we chose). [11, 15] apply FW on the randomly smoothed surrogate instead of the original non-smooth objective. However, computing the gradient at each iteration requires to query many time on the objective function, which may not be practical. [1, 29] use a proximal operator to replace the objective function, but as pointed out in [4], the smoothing parameters of the proximal operator are not trivial to tune. Our solution is close to those developed in [30, 4]. There, inspired by the approximate subdifferential [38], the authors propose to collect the set of the gradients in the neighborhood at each round. They show that these collection is continuous even when the objective functions is non-smooth, which allows for the use of FW. The way we deal with the non-smoothness issue is similar but simplified by the fact that the specific form of our objective function.

## 3 Preliminaries

We consider the pure exploration task described in the introduction. This section presents the additional assumptions made towards the design and analysis of our algorithm. These assumptions are here illustrated for the classical Best Arm Identification (BAI) task in unstructured bandits (see Example 1); they will be verified for all other examples of pure exploration problems presented in Section 5. This section also provides useful properties of the lower bound optimization problem (1), and finally describes our choice of stopping and decision rules.

### 3.1 Assumptions and properties of the lower bound optimization problem

The answer map $i^\star : \Lambda \to \mathcal{I}$ allows us to decompose $\Lambda$ into a union of non-overlapping sets: $\Lambda = \cup_{i \in \mathcal{I}} \mathcal{S}_i$, where $\mathcal{S}_i = \{\boldsymbol{\mu} \in \Lambda : i^\star(\boldsymbol{\mu}) = i\}$ for all $i \in \mathcal{I}$. The answer map is known (i.e., knowing $\boldsymbol{\mu}$ is enough to output the right answer), and hence without loss of generality, we can assume

that $\mathcal{S}_i \neq \emptyset$ for all $i \in \mathcal{I}$. Using this notation, the set of confusing parameters can be written as $\text{Alt}(\boldsymbol{\mu}) = \cup_{i \neq i^\star(\boldsymbol{\mu})} \mathcal{S}_i$.

**Assumption 1.** *For each $i \in \mathcal{I}$, $\mathcal{S}_i$ is an open set and the complementary of $\mathcal{S}_i$ is a finite union of convex sets. Namely, there exists a finite collection $\mathcal{J}_i$ of convex sets $\mathcal{C}_j^i$ s.t. $\Lambda \setminus \mathcal{S}_i = \cup_{j \in \mathcal{J}_i} \mathcal{C}_j^i$.*

*Example 1. The BAI task in unstructured bandits with Bernoulli rewards.* For this task, we have $\Lambda = (0,1)^K$, $\mathcal{I} = \{1, \ldots, K\}$, and for all arm $i$, the set of parameters for which arm $i$ is the best arm is $\mathcal{S}_i = \{\boldsymbol{\mu} \in \Lambda : \mu_i > \mu_k, \forall k \neq i\}$. We have: $\Lambda \setminus \mathcal{S}_i = \cup_{j \in \mathcal{J}_i} \mathcal{C}_j^i$ where $\mathcal{J}_i = \mathcal{I} \setminus \{i\}$ is the set of arms different than $i$ and $\mathcal{C}_j^i = \{\boldsymbol{\mu} \in \Lambda : \mu_j > \mu_i\}$ is the convex set of parameters for which arm $j$ is better than arm $i$. $\qquad\square$

Now under Assumption 1, we can decompose the lower bound optimization problem as follows: $T^\star(\boldsymbol{\mu})^{-1} = \sup_{\boldsymbol{\omega} \in \Sigma} F_{\boldsymbol{\mu}}(\boldsymbol{\omega})$ where $F_{\boldsymbol{\mu}}(\boldsymbol{\omega}) = \min_{j \in \mathcal{J}_{i^\star(\boldsymbol{\mu})}} f_j(\boldsymbol{\omega}, \boldsymbol{\mu})$ and for all $j \in \mathcal{J}_{i^\star(\boldsymbol{\mu})}$,

$$f_j(\boldsymbol{\omega}, \boldsymbol{\mu}) = \inf_{\boldsymbol{\lambda} \in \mathcal{C}_j^{i^\star(\boldsymbol{\mu})}} \sum_{k=1}^{K} \omega_k d(\mu_k, \lambda_k). \tag{2}$$

Note that (2) is convex program (by convexity of the KL divergence), and that $f_j$ is a concave function in $\boldsymbol{\omega}$ (as the minimum of concave functions). As a consequence, the objective function $F_{\boldsymbol{\mu}}$ is also concave, but not smooth. The following proposition summarizes insightful properties of the functions $f_j, j \in \mathcal{J}_{i^\star(\boldsymbol{\mu})}$. It is a consequence of the envelope theorem and proved in Appendix K.2.

**Proposition 1.** *Let $i \in \mathcal{I}$, $j \in \mathcal{J}_i$. Define for all $(\boldsymbol{\omega}, \boldsymbol{\mu}) \in \Sigma \times \mathcal{S}_i$,*

$$\overline{\boldsymbol{\lambda}_j(\boldsymbol{\omega}, \boldsymbol{\mu})} = \arg \min_{\boldsymbol{\lambda} \in \text{cl}(\mathcal{C}_j^i)} \sum_{k=1}^{K} \omega_k d(\mu_k, \lambda_k), \tag{3}$$

*where $\text{cl}(\mathcal{C}_j^i)$ is the closure of $\mathcal{C}_j^i$. Then under Assumption 1, $\overline{\boldsymbol{\lambda}_j(\boldsymbol{\omega}, \boldsymbol{\mu})}$ is unique for all $(\boldsymbol{\omega}, \boldsymbol{\mu}) \in \mathring{\Sigma} \times \mathcal{S}_i$, where $\mathring{\Sigma}$ is the interior of $\Sigma$. In addition, $f_j$ is continuously differentiable on $\mathring{\Sigma} \times \mathcal{S}_i$, and $\forall (\boldsymbol{\omega}, \boldsymbol{\mu}) \in \mathring{\Sigma} \times \mathcal{S}_i$,*

$$\nabla_{\boldsymbol{\omega}} f_j(\boldsymbol{\omega}, \boldsymbol{\mu}) = \sum_{k=1}^{K} d(\mu_k, \overline{\boldsymbol{\lambda}_j(\boldsymbol{\omega}, \boldsymbol{\mu})}_k) e_k, \tag{4}$$

*where $e_k$ denotes the $K$-dimensional vector whose $k$-th coordinate is 1 and whose other coordinates are 0.*

A key insight from the above result is that the objective function $F_{\boldsymbol{\mu}}$ is the minimum of a finite number of continuously differentiable functions. This observation will make the use of a slightly modifed FW algorithm possible (remember that the FW algorithm is known to converge for smooth functions only). We use an additional assumption on the gradient and curvature of $f_j$. A controlled curvature is an essential ingredient when analyzing the convergence of FW-based algorithms, see e.g. [17]. Define $\Sigma_\gamma = \{\boldsymbol{\omega} \in \Sigma : \min_k \omega_k \geq \gamma\}$ for any $\gamma \in (0, 1/K)$. Following [17], we define $C_\psi(\mathcal{K})$, the curvature constant of the concave differentiable function $\psi : \mathcal{K} \to \mathbb{R}$ with respect to the compact set $\mathcal{K}$, as

$$C_\psi(\mathcal{K}) = \sup_{\substack{\boldsymbol{x}, \boldsymbol{z} \in \mathcal{K} \\ \alpha \in (0,1] \\ \boldsymbol{y} = \boldsymbol{x} + \alpha(\boldsymbol{z} - \boldsymbol{x})}} \frac{1}{\alpha^2} \left[ \psi(\boldsymbol{x}) - \psi(\boldsymbol{y}) + \langle \boldsymbol{y} - \boldsymbol{x}, \nabla\psi(\boldsymbol{x}) \rangle \right]. \tag{5}$$

Refer to [17], for the intuition behind this defintion and examples.

**Assumption 2.** *For all $\boldsymbol{\mu} \in \Lambda$,*
*(i) there exists $L > 0$ such that $\forall j \in \mathcal{J}_{i^\star(\boldsymbol{\mu})}, \omega \in \Sigma, \|\nabla_{\boldsymbol{\omega}} f_j(\boldsymbol{\omega}, \boldsymbol{\mu})\|_\infty \leq L$;*
*(ii) there exists $D > 0$ such that $\forall \gamma \in (0, 1/K)$ and $\forall j \in \mathcal{J}_{i^\star(\boldsymbol{\mu})}, C_{f_j(\cdot, \boldsymbol{\mu})}(\Sigma_\gamma) \leq \frac{D}{\gamma}$.*

There is a simple way to verify whether a pure exploration problem satisfies Assumption 2, by looking at the second derivative of the function $y \mapsto d(x, y)$ at the points $(\mu_k, (\overline{\boldsymbol{\lambda}_j(\boldsymbol{\omega}, \boldsymbol{\mu})})_k)$ for all $k$. Refer to Appendix C for details.

*Example 1 (cont'd).* For unstructured bandits with Bernoulli rewards, we can easily compute $f_j$ and its gradient [13, 28]: for all $j \neq i^\star(\boldsymbol{\mu})$ and all $\boldsymbol{\omega} \in \overset{\circ}{\Sigma}$, define $m_j(\boldsymbol{\omega}, \boldsymbol{\mu}) = \frac{\omega_{i^\star(\boldsymbol{\mu})}\mu_{i^\star(\boldsymbol{\mu})} + \omega_j \mu_j}{\omega_{i^\star(\boldsymbol{\mu})} + \omega_j}$. Then $\overline{\boldsymbol{\lambda}_j(\boldsymbol{\omega}, \boldsymbol{\mu})}_k = \mu_k$ if $k \notin \{i^\star(\boldsymbol{\mu}), j\}$ and $\overline{\boldsymbol{\lambda}_j(\boldsymbol{\omega}, \boldsymbol{\mu})}_k = m_j(\boldsymbol{\omega}, \boldsymbol{\mu})$ otherwise. As a consequence:

$$\begin{cases} f_j(\boldsymbol{\omega}, \boldsymbol{\mu}) = \omega_{i^\star(\boldsymbol{\mu})} d(\mu_{i^\star(\boldsymbol{\mu})}, m_j(\boldsymbol{\omega}, \boldsymbol{\mu})) + \omega_j d(\mu_j, m_j(\boldsymbol{\omega}, \boldsymbol{\mu})), \\ \nabla_{\boldsymbol{\omega}} f_j(\boldsymbol{\omega}, \boldsymbol{\mu}) = d(\mu_{i^\star(\boldsymbol{\mu})}, m_j(\boldsymbol{\omega}, \boldsymbol{\mu})) \boldsymbol{e}_{i^\star(\boldsymbol{\mu})} + d(\mu_j, m_j(\boldsymbol{\omega}, \boldsymbol{\mu})) \boldsymbol{e}_j. \end{cases} \quad (6)$$

For this example, we can verify that Assumption 2 holds, either directly or using the tool described in Appendix C. $\qquad \square$

### 3.2 Stopping and decision rules

Next we present the two last components of the FWS algorithm, namely the stopping and decision rules. These components are standard and borrowed from the existing literature. We need a few notations. For any $t \geq 1$, let $A_t$ denote the arm selected in round $t$. Define $N_k(t) = \sum_{s=1}^{t} \mathbb{1}\{A_s = k\}$ the number of times arm $k$ has been selected up to round $t$, and by $\omega_k(t) = N_k(t)/t$ the corresponding empirical proportion of draw. When $N_k(t) > 0$, the empirical average reward of arm $k$ up to round $t$ is denoted by $\hat{\mu}_k(t) = \sum_{s=1}^{t} X_k(s)\mathbb{1}\{A_s = k\}/N_k(t)$, where $X_k(s)$ is the random reward received from pulling arm $k$ in round $s$.

Let us denote by $\tau$, the stopping time defining when the algorithm stops exploring and has to output a decision. Our decision rule is obviously to output the best empirical answer: $\hat{\imath}_\tau = i^\star(\hat{\boldsymbol{\mu}}(\tau))$.

For the stopping rule, as in other existing algorithms, we leverage a Generalized Likelihood Ratio Test (GLRT). Our test boils down to comparing $tF_{\hat{\boldsymbol{\mu}}(t)}(\boldsymbol{\omega}(t))$ to a threshold $\beta(t, \delta)$ (recall that $F_{\boldsymbol{\mu}}$ is the objective function of the lower bound optimization problem):

$$\tau = \inf\{t \geq 1 : tF_{\hat{\boldsymbol{\mu}}(t)}(\boldsymbol{\omega}(t)) \geq \beta(t, \delta)\}. \quad (7)$$

Many thresholds $\beta(t, \delta)$ have been proposed in the literature [21, 13, 18, 28]. For FWS and its analysis, we just need that the threshold statisfies the two following properties:

$$\forall t \geq 1, \ \left(tF_{\hat{\boldsymbol{\mu}}(t)}(\boldsymbol{\omega}(t)) \geq \beta(t, \delta)\right) \implies \left(\mathbb{P}_{\boldsymbol{\mu}}\left[i^\star(\hat{\boldsymbol{\mu}}(t)) \neq i^\star(\boldsymbol{\mu})\right] \leq \delta\right), \quad (8)$$

$$\exists c_1(\Lambda), c_2(\Lambda) > 0 \ : \ \forall t \geq c_1(\Lambda), \ \beta(t, \delta) \leq \log\left(\frac{c_2(\Lambda)t}{\delta}\right). \quad (9)$$

The first of the above properties will naturally imply that FWS returns the true answer with probability at least $1-\delta$ when stopping, whereas the second will be instrumental in the sample complexity analysis (there, $c_1(\Lambda), c_2(\Lambda)$ may depend on the set of possible instances, and on the reward distributions). In [21], the authors manage to provide, for any generic pure exploration task, a single threshold satisfying (8)-(9)). Unless otherwise mentioned, we will use the stopping rule implementing this threshold.

## 4 The FWS Algorithm and its Sample Complexity

In the FWS algorithm, we use the FW algorithm to learn an optimal allocation $\boldsymbol{\omega}^\star(\boldsymbol{\mu})$. In each round, an iteration of FW updates the allocation that the FWS algorithm aims at approaching using some tracking procedure. We describe this learning and tracking procedure below.

### 4.1 Adapting Frank-Wolfe to the non-smooth function $F_{\boldsymbol{\mu}}$

The FW algorithm [12] solves smooth convex programs by linearizing, in each iteration, the objective function and moving towards a minimizer of this linear function. Compared to the projected gradient and proximal methods, FW is computationally more efficient (e.g. it avoids the projection step), and is particularly well-suited when optimizing over polyhedra [3] (which is our case here). For a contemporary treatment of FW, refer to [17]. FW was suggested in [13] for BAI in unstructured bandits to update the allocation to be tracked. For this BAI problem, an iteration of the FW algorithm takes an intuitive form (see also Appendix A2 in [28]):

*Example 1 (cont'd).* For BAI in unstructured bandits, the optimal allocation $\boldsymbol{\omega}^\star(\boldsymbol{\mu})$ is the maximizer of the function $\boldsymbol{\omega} \mapsto F_{\boldsymbol{\mu}}(\boldsymbol{\omega}) = \min_j f_j(\boldsymbol{\omega}, \boldsymbol{\mu})$. $F_{\boldsymbol{\mu}}$ is smooth at points when the minimum is realized at a single arm $j^\star = \arg\min_j f_j(\boldsymbol{\omega}, \boldsymbol{\mu})$, and there, in view of (6), its gradient is

$\nabla F_{\boldsymbol{\mu}}(\boldsymbol{\omega}) = d(\mu_{i^{\star}(\boldsymbol{\mu})}, m_{j^{\star}}(\boldsymbol{\omega}, \boldsymbol{\mu})) \boldsymbol{e}_{i^{\star}(\boldsymbol{\mu})} + d(\mu_{j^{\star}}, m_{j^{\star}}(\boldsymbol{\omega}, \boldsymbol{\mu})) \boldsymbol{e}_{j^{\star}}$. Now in an iteration of the FW algorithm, one would follow the direction given by $\arg\max_{\boldsymbol{\omega}' \in \Sigma} \boldsymbol{\omega}'^{\top} \nabla F_{\boldsymbol{\mu}}(\boldsymbol{\omega})$. This direction is $\boldsymbol{e}_{j^{\star}}$ if $d(\mu_{j^{\star}}, m_{j^{\star}}(\boldsymbol{\omega}, \boldsymbol{\mu})) > d(\mu_{i^{\star}(\boldsymbol{\mu})}, m_{j^{\star}}(\boldsymbol{\omega}, \boldsymbol{\mu}))$, and $\boldsymbol{e}_{i^{\star}(\boldsymbol{\mu})}$ otherwise. This is precisely what the FW-type sampling rule suggested in [13] is doing: in round $(t+1)$, the best challenger is defined as $j^{\star} = \arg\min_{j} f_{j}(\boldsymbol{\omega}(t), \hat{\boldsymbol{\mu}}(t))$, and the arm selected corresponds to the direction given by $\arg\max_{\boldsymbol{\omega}' \in \Sigma} \boldsymbol{\omega}'^{\top} \nabla F_{\hat{\boldsymbol{\mu}}(t)}(\boldsymbol{\omega}(t))$, i.e., it is either the best challenger $j^{\star}$ or the best empirical arm $i^{\star}(\hat{\boldsymbol{\mu}}(t))$. $\qquad\square$

The convergence analysis of FW usually requires that the objective function is smooth, and that its curvature can be controlled. When applying FW-type algorithms to design an optimal sampling rule (a rule that converges to the allocation $\boldsymbol{\omega}^{\star}(\boldsymbol{\mu})$ maximizing $F_{\boldsymbol{\mu}}$), we face three issues: (i) $F_{\boldsymbol{\mu}}$ is not smooth; (ii) $F_{\boldsymbol{\mu}}$ has an unbounded curvature close to the boundary of $\Sigma$; (iii) $\boldsymbol{\mu}$ is unknown initially, so the FW iteration in round $t$ can be applied to $F_{\hat{\boldsymbol{\mu}}(t)}$ only. We discuss below how we circumvent these issues in the design of our algorithm.

**(i) Non-smoothness of $F_{\boldsymbol{\mu}}$.** In view of Proposition 1, $F_{\boldsymbol{\mu}}$ is the minimum of a finite number of smooth concave functions $f_{j}$. Hence at points where two of these functions are equal in $\boldsymbol{\omega}$, $F_{\boldsymbol{\mu}}$ is not differentiable in $\boldsymbol{\omega}$. The FW algorithm has been adapted to cope with non-smooth functions, see e.g. [30]. Typically, one constructs continuous approximations of the gradient close to non-smooth points of the functions. This construction often involves the $r$-subdifferential [16][2], which would be too costly to compute for $F_{\boldsymbol{\mu}}$. Instead, we can leverage the fact $F_{\boldsymbol{\mu}}$ is the minimum of concave functions, and construct the called $r$-*subdifferential subspace*: for $r \in (0, 1)$,

$$H_{F_{\boldsymbol{\mu}}}(\boldsymbol{\omega}, r) = \mathrm{cov}\left\{ \nabla f_{j}(\boldsymbol{\omega}, \boldsymbol{\mu}) : j \in \mathcal{J}_{i^{\star}(\boldsymbol{\mu})}, f_{j}(\boldsymbol{\omega}, \boldsymbol{\mu}) < F_{\boldsymbol{\mu}}(\boldsymbol{\omega}) + r \right\}, \tag{10}$$

where $\mathrm{cov}\{S\}$ denotes the convex hull of the set $S$. This choice greatly simplifies because it does not require to compute the gradient of $f_{j}$ in a neighborhood of $\boldsymbol{\omega}$. Since the $f_{j}$ are continuously differentiable, we can prove that $\boldsymbol{\omega} \mapsto H_{F_{\boldsymbol{\mu}}}(\boldsymbol{\omega}, r)$ is a continuous (i.e. upper- and lower-hemicontinuous). Using the $r$-subdifferential subspace, the modified FW update is given as follows. Let $\boldsymbol{x}(t)$ be the estimated optimizer of $F_{\boldsymbol{\mu}}$ in round $t$. In round $(t+1)$, it is updated as:

$$\begin{cases} \boldsymbol{z}(t+1) = \mathrm{argmax}_{\boldsymbol{z} \in \Sigma} \min_{h \in H_{F_{\boldsymbol{\mu}}}(\boldsymbol{x}(t), r_{t})} \langle \boldsymbol{z} - \boldsymbol{x}(t), h \rangle \quad \textit{(ties broken arbitrarily)}, \\ \boldsymbol{x}(t+1) = \frac{t}{t+1}\boldsymbol{x}(t) + \frac{1}{t+1}\boldsymbol{z}(t+1). \end{cases} \tag{11}$$

Of course in the `FWS` algorithm, $\boldsymbol{\mu}$ is unknown, and will be simply replaced by $\hat{\boldsymbol{\mu}}(t)$ in the above update. The way we choose the sequence of parameters $\{r_{t}\}_{t \geq 1}$ will be discussed later. Computing $\boldsymbol{z}(t)$ is equivalent to solving a zero-sum game, which can be further formulated as a LP [40] (Chapter 20). Refer to Appendix H for a detailed description of this LP.

**(ii) Unbounded curvature of $F_{\boldsymbol{\mu}}$ and (iii) unknown $\boldsymbol{\mu}$.** These two issues are solved by a single trick. We impose that in the FW iterations, the update directions $\boldsymbol{z}(t)$ cover all $\boldsymbol{e}_{k}, k = 1, \ldots, K$ sufficiently often. This ensures that the target allocation $\boldsymbol{x}(t)$ stays away from the boundary of $\Sigma$, which in turn allows us to control the curvature of $F_{\hat{\boldsymbol{\mu}}(t)}$ thanks to Assumption 2. This imposed constraint can be seen as a sort of forced exploration, and further implies (thanks to our tracking procedure) that each arm is played often enough. Now, with this kind of forced exploration, $\hat{\boldsymbol{\mu}}(t)$ will concentrate around the true $\boldsymbol{\mu}$.

### 4.2 Algorithm

The `FWS` algorithm proceeds as follows. `FWS` maintains a target allocation, denoted by $\boldsymbol{x}(t)$, its empirical allocation $\boldsymbol{\omega}(t)$, and the empirical average rewards $\hat{\boldsymbol{\mu}}(t)$ after round $t$. After an initialization phase ($K$ rounds where each arm is selected), `FWS` alternates between forced exploration and FW updates. More precisely:
*Forced exploration* occurs at rounds $t$ where $\sqrt{\lfloor t/K \rfloor}$ is an integer and at those where $\hat{\boldsymbol{\mu}}(t-1) \notin \Lambda$ (in this case, we cannot compute the objective function). In forced exploration round $t$, the target allocation is updated towards the center of the simplex: $\boldsymbol{x}(t) = \frac{t-1}{t}\boldsymbol{x}(t-1) + \frac{1}{t}(1/K, \ldots, 1/K)$.
*FW updates* happen in other rounds. There, the target allocation is updated according to our adapted version of FW (11), where in round $t$ the unknown $\boldsymbol{\mu}$ is replaced by $\hat{\boldsymbol{\mu}}(t-1)$. In the successive FW

---

[2]For $r \in (0, 1)$, the $r$-subdifferential of $\psi : \mathcal{K} \to \mathbb{R}$ (where $\mathcal{K} \subset \mathbb{R}^{K}$ is compact and convex) is defined as $\partial_{r}\psi(\boldsymbol{x}) = \{h \in \mathbb{R}^{K} : \psi(\boldsymbol{y}) < \psi(\boldsymbol{x}) + \langle \boldsymbol{y} - \boldsymbol{x}, h \rangle + r \text{ for all } \boldsymbol{y} \in \mathcal{K}\}$.

updates, we use $r$-subdifferential subspaces with varying parameter $r$. For the analysis of FWS, we will select a sequence of parameters $\{r_t\}_{t\geq 1}$ with an appropriate decay rate.

After the target allocation is updated in round $t$, the algorithm tracks this allocation by selecting the arm maximizing over $k$ the ratio $x_k(t)/\omega_k(t-1)$. Finally, FWS, whose pseudo-code is presented below, uses the stopping and decision rules described in §3.2.

---

**Algorithm 1:** FWS algorithm

---

**Input:** Confidence level $\delta$, sequence $\{r_t\}_{t\geq 1}$
**Initialization:** Sample each arm once and update $\boldsymbol{\omega}(K)$, $\boldsymbol{x}(K) = (\frac{1}{K}, \ldots, \frac{1}{K})$, and $\hat{\boldsymbol{\mu}}(K)$
$t \leftarrow K$
**While** $(tF_{\hat{\boldsymbol{\mu}}(t)}(\boldsymbol{\omega}(t)) < \beta(\delta, t)$ or $\hat{\boldsymbol{\mu}}(t-1) \notin \Lambda)$
$\qquad t \leftarrow t+1$
$\qquad$ **If** $(\sqrt{\lfloor t/K \rfloor} \in \mathbb{N}$ or $\hat{\boldsymbol{\mu}}(t-1) \notin \Lambda)$ *(forced exploration)* $\boldsymbol{z}(t) \leftarrow (\frac{1}{K}, \ldots, \frac{1}{K})$
$\qquad$ **Else** *(FW update)*

$$\qquad\qquad \boldsymbol{z}(t) \leftarrow \underset{\boldsymbol{z} \in \Sigma}{\operatorname{argmax}} \min_{h \in H_{F_{\hat{\boldsymbol{\mu}}(t-1)}}(\boldsymbol{x}(t-1), r_t)} \langle \boldsymbol{z} - \boldsymbol{x}(t-1), h \rangle \quad \textit{(ties broken arbitrarily)}$$

$\qquad$ Update $\boldsymbol{x}(t) \leftarrow \frac{t-1}{t}\boldsymbol{x}(t-1) + \frac{1}{t}\boldsymbol{z}(t)$
$\qquad$ Sample the arm $A_t \leftarrow \operatorname{argmax}_k x_k(t)/\omega_k(t-1)$ $\quad$ *(ties broken arbitrarily)*
$\qquad$ Update $\boldsymbol{\omega}(t)$ and $\hat{\boldsymbol{\mu}}(t)$
**Output:** $i^\star(\hat{\boldsymbol{\mu}}(t))$

---

### 4.3 Sample complexity

In the following theorem, we establish the asymptotic optimality of FWS.

**Theorem 1.** *Consider the* FWS *algorithm with a sequence $\{r_t\}_{t\geq 1}$ of strictly positive reals satisfying (i) $\lim_{t\to\infty} \frac{1}{t}\sum_{s=1}^t r_s = 0$, and (ii) $\lim_{t\to\infty} tr_t = \infty$. Under Assumptions 1, 2, the algorithm terminates in finite time almost surely and is $\delta$-PAC. Its sample complexity $\tau$ satisfies:*

$$\forall \boldsymbol{\mu} \in \Lambda, \quad \mathbb{P}_{\boldsymbol{\mu}}\left[\limsup_{\delta\to 0} \frac{\tau}{\log(1/\delta)} \leq T^\star(\boldsymbol{\mu})\right] = 1, \quad and \quad \limsup_{\delta\to 0} \frac{\mathbb{E}_{\boldsymbol{\mu}}[\tau]}{\log(1/\delta)} \leq T^\star(\boldsymbol{\mu}).$$

The proof is given in Appendix I. We sketch the proof of the guarantees in expectation. The proof relies on classical concentration results, but more critically combines continuity arguments (developed in Appendix K) to account for the varying $\hat{\boldsymbol{\mu}}(t)$, and tools to analyze the convergence of the modified FW algorithm (reported in Appendix L).
1. First using concentration inequalities and the fact that FWS includes forced exploration rounds, we can define, for round $t$, a "good" event $\mathcal{E}_t$ under which $\hat{\boldsymbol{\mu}}(t)$ is very close to $\boldsymbol{\mu}$ and such that $\sum_{t=1}^\infty \mathbb{P}_{\boldsymbol{\mu}}[\mathcal{E}_t^c] < \infty$. Then, several continuity arguments have to be made. In Lemma 6 (Appendix K) we show that $\boldsymbol{\mu} \mapsto F_{\boldsymbol{\mu}}$ is continuous (w.r.t. the uniform convergence norm). In Theorem 3 (Appendix K) we also prove that the solution $\boldsymbol{z}(t+1)$ of the FW update (11) is continuous in $\boldsymbol{\mu}$. The arguments above allow us to analyze the convergence of the FW updates almost as if $\hat{\boldsymbol{\mu}}(t)$ was replaced by $\boldsymbol{\mu}$ provided that the event $\mathcal{E}_t$ occurs.
2. Now we can study under the event $\mathcal{E}_t$, the impact of the FW update on the target allocation. The main step of our proof is Theorem 6 (Appendix L) characterizing how $F_{\boldsymbol{\mu}}(\boldsymbol{x}(t))$ get closer to $F_{\boldsymbol{\mu}}(\boldsymbol{\omega}^\star(\boldsymbol{\mu}))$ in each FW update. We then deduce that after a time $T_1$, $F_{\hat{\boldsymbol{\mu}}(t)}(\boldsymbol{x}(t))$ is a good approximation of $F_{\boldsymbol{\mu}}(\boldsymbol{\omega}^\star(\boldsymbol{\mu}))$.
3. We conclude the proof using similar arguments as those in [13]. According to our stopping rule, $t > \tau$ if and only if $tF_{\hat{\boldsymbol{\mu}}(t)}(\boldsymbol{\omega}(t)) > \beta(t, \delta)$. Hence $\mathbb{E}_{\boldsymbol{\mu}}[\tau] = \sum_{t=1}^\infty \mathbb{P}_{\boldsymbol{\mu}}[\tau > t] = \sum_{t=1}^\infty \mathbb{P}_{\boldsymbol{\mu}}[tF_{\hat{\boldsymbol{\mu}}(t)}(\boldsymbol{\omega}(t)) \leq \beta(t, \delta)]$ which can be approximately upper bounded by $T_1 + \sum_{t=T_1}^\infty \mathbb{P}_{\boldsymbol{\mu}}[\mathcal{E}_t^c] + \sum_{t=1}^\infty \mathbb{P}_{\boldsymbol{\mu}}[tF_{\boldsymbol{\mu}}(\boldsymbol{\omega}^\star(\boldsymbol{\mu})) \leq \beta(t, \delta)]$. The proof is concluded by remarking that in view of the property (9) of our stopping threshold, the last sum is close to $T^\star(\boldsymbol{\mu})\log(1/\delta)$ as $\delta \to 0$.
Note that our proof of Theorem 1 accounts for the possibility in certain structures (e.g. linear) of

having multiple optimal allocations (these allocations form a convex set). We just reason in terms of the objective function (as in [18] for linear bandits).

Under the following additional assumption, we can derive non-asymptotic sample complexity upper bound for FWS. The proof of the following theorem is presented in Appendix N.

**Assumption 3.** *For any $\boldsymbol{\mu} \in \Lambda$, there exist constants $\kappa, E > 0$, s.t. if $\|\boldsymbol{\pi} - \boldsymbol{\mu}\|_\infty \leq \kappa$, then $\boldsymbol{\pi} \in \mathcal{S}_{i^\star(\boldsymbol{\mu})}, \forall \boldsymbol{\omega} \in \overset{\circ}{\Sigma}, \ j \in \mathcal{J}_{i^\star(\boldsymbol{\mu})}, \nabla_{\boldsymbol{\pi}} d(\pi_k, \overline{\boldsymbol{\lambda}_j(\boldsymbol{\omega}, \boldsymbol{\pi})}_k)$ is continuous and $\left\|\nabla_{\boldsymbol{\pi}} d(\pi_k, \overline{\boldsymbol{\lambda}_j(\boldsymbol{\omega}, \boldsymbol{\pi})}_k)\right\|_1 \leq E, \ \forall k = 1, \ldots, K$.*

**Theorem 2.** *Consider the FWS algorithm with a sequence $\{r_t\}_{t \geq 1}$ as in Theorem 1. Under Assumptions 1, 2, and 3, the sample complexity $\tau$ of the algorithm satisfies: for any $\boldsymbol{\mu} \in \Lambda, \delta \in (0, 1)$, and any $\epsilon < \min\{\kappa E/2, 1\}, \tilde{\epsilon} < 1$,*

$$
\mathbb{E}_{\boldsymbol{\mu}}[\tau] \leq \frac{1 + \tilde{\epsilon}}{F_{\boldsymbol{\mu}}(\boldsymbol{\omega}^\star(\boldsymbol{\mu})) - 6\epsilon} \left[\log\left(\frac{(1 + \tilde{\epsilon})c_2(\Lambda)e}{\delta(F_{\boldsymbol{\mu}}(\boldsymbol{\omega}^\star(\boldsymbol{\mu})) - 6\epsilon)}\right) + \log\log\left(\frac{(1 + \tilde{\epsilon})c_2(\Lambda)}{\delta(F_{\boldsymbol{\mu}}(\boldsymbol{\omega}^\star(\boldsymbol{\mu})) - 6\epsilon)}\right)\right]
$$
$$
+ \Psi(K, D, E, L, c_1(\Lambda), \epsilon) + T_{\epsilon, L}^{\frac{5}{4}},
$$

*where $T_{\epsilon, L}$ is a constant such if $t \geq T_{\epsilon, L}$, then $\sum_{s=1}^{t} r_s < t\epsilon$ and $tr_t > L$. The constant $\Psi$ is polynomial in $(D, E, L, c_1(\Lambda), 1/\epsilon)$ and exponential in $K$. The precise definition of $\Psi$ is given in Appendix N.*

## 5 Examples and Experiments for Linear Bandits

### 5.1 Examples

Our framework can be applied to many pure exploration problems, including BAI in unstructured (see Example 1), linear, Lipschitz bandits. It further covers threshold bandits (the problem of identifying all arms with rewards greater than a threshold), linear threshold bandits, top-$m$ bandits (where we wish to identify the best $m$ arms), and dueling bandits. All these examples are presented in Appendix. Using numerical experiments, we show that FWS is competitive with state-of-the-art algorithms for BAI in unstructured, linear, and Lipschitz bandits, see Appendices D-E-F, respectively. To the best of our knowledge, we report the first results for BAI in Lipschitz bandits. We quote some of our results for BAI in linear bandits below.

When facing a new pure exploration problem, one can check whether it falls into our framework, by first directly verifying Assumption 1. In Appendix C, we provide a simple sufficient condition ensuring that Assumption 2 holds, and explain why all the aforementioned pure exploration problems satisfy this condition.

### 5.2 BAI in linear bandits

Linear bandits constitute arguably the most popular and important bandit problems with structure, and have found many applications [25, 5]. BAI in linear bandits has received a lot of attention recently, see §2. To model linear bandits, we slightly modify our framework. The reason for this modification is that the linear structure is so strong that using our initial framework, the set $\Lambda$ would be small, and we would have problems ensuring that $\hat{\boldsymbol{\mu}}(t) \in \Lambda$ after some reasonable time $t$. Alternatively (rather than modifying the framework), we could modify the FWS algorithm so that $\hat{\boldsymbol{\mu}}(t)$ is projected onto $\Lambda$.

Consider a set of $K$ arms. Arm $k$ is attached a $d$-dimensional feature vector $\boldsymbol{a}_k$ and its average reward $\langle \boldsymbol{a}_k, \boldsymbol{\mu} \rangle$, where $\boldsymbol{\mu} \in \mathbb{R}^d$ is unknown. Without loss of generality, we assume that $\{\boldsymbol{a}_k\}_{k \in [K]}$ spans $\mathbb{R}^d$. We modify the definition of $\Lambda$ as follows: $\Lambda = \{\boldsymbol{\mu} \in \mathbb{R}^d : \exists k \in [K] \text{ s.t.} \langle \boldsymbol{a}_k - \boldsymbol{a}_i, \boldsymbol{\mu} \rangle > 0, \forall i \neq k\}$. Hence $\boldsymbol{\mu}$ parametrizes the average rewards of the arms, but $\mu_k$ is not the average reward of arm $k$. The true answer is $i^\star(\boldsymbol{\mu}) = \text{argmax}_k \langle \boldsymbol{a}_k, \boldsymbol{\mu} \rangle$. The lower bound optimization problem (1) becomes: $\sup_{\boldsymbol{\omega} \in \Sigma} F_{\boldsymbol{\mu}}(\boldsymbol{\omega})$ where $F_{\boldsymbol{\mu}}(\boldsymbol{\omega}) = \inf_{\boldsymbol{\lambda} \in \text{Alt}(\boldsymbol{\mu})} \frac{1}{2}(\boldsymbol{\mu} - \boldsymbol{\lambda})^\top \sum_k \omega_k \boldsymbol{a}_k \boldsymbol{a}_k^\top (\boldsymbol{\mu} - \boldsymbol{\lambda})$ and $\text{Alt}(\boldsymbol{\mu}) = \{\boldsymbol{\lambda} \in \Lambda : \exists k \neq i^\star(\boldsymbol{\mu}) \text{ s.t.} \langle \boldsymbol{a}_k - \boldsymbol{a}_{i^\star(\boldsymbol{\mu})}, \boldsymbol{\lambda} \rangle > 0\}$, see e.g. [18]. From there, we can reproduce our framework: for Assumption 1, for all $j \neq i^\star(\boldsymbol{\mu})$, $\mathcal{C}_j^{i^\star(\boldsymbol{\mu})} = \{\boldsymbol{\lambda} \in \Lambda : \langle \boldsymbol{a}_j - \boldsymbol{a}_{i^\star(\boldsymbol{\mu})}, \boldsymbol{\lambda} \rangle > 0\}$; as for

the functions $f_j$, they are defined through:

$$\overline{\boldsymbol{\lambda}_j(\boldsymbol{\omega}, \boldsymbol{\mu})} = \boldsymbol{\mu} + \left( \frac{\langle \boldsymbol{a}_{i^\star(\boldsymbol{\mu})} - \boldsymbol{a}_j, \boldsymbol{\mu} \rangle}{\left\| \boldsymbol{a}_{i^\star(\boldsymbol{\mu})} - \boldsymbol{a}_j \right\|_{V_{\boldsymbol{\omega}}^{-1}}^2} V_{\boldsymbol{\omega}}^{-1} \right) \left( \boldsymbol{a}_j - \boldsymbol{a}_{i^\star(\boldsymbol{\mu})} \right),$$ (12)

where $V_{\boldsymbol{\omega}} = \sum_k \omega_k \boldsymbol{a}_k \boldsymbol{a}_k^\top$. In the FWS algorithm for linear bandits, we use the Least-Squares Estimator (LSE) $\hat{\boldsymbol{\mu}}(t)$ given past observations, see [18] or Appendix E for an explicit expression. It can be readily seen that this slight modification of our framework does not affect the validity of Theorem 1. We just need to use the concentration inequalities derived in [18] for $\hat{\boldsymbol{\mu}}(t)$ in the first step of its proof.

**Numerical experiments.** We consider the example proposed by [35]. The unknown parameter $\boldsymbol{\mu} = \boldsymbol{e}_1$ and there are $d + 1$ arms, $\boldsymbol{e}_1, \cdots, \boldsymbol{e}_d, \cos(\phi)\boldsymbol{e}_1 + \sin(\phi)\boldsymbol{e}_2$ in $\mathbb{R}^d$, where $(\boldsymbol{e}_1, \cdots, \boldsymbol{e}_d)$ form the standard orthonormal basis. We set $d = 6$ and $\phi = 0.1$. To assess the performance of the FWS algorithm, we compare with the following algorithms: the Lazy Track and Stop algorithm (LT) from [18]; LineGame-C (CG-C) and LineGame (Lk-C) from [9] and implemented by [34]; the XY-Adaptive algorithm (XY-A) from [35]. For information, we also run the Round Robin algorithm RR selecting each equally. For comparison, we finally compute the sample complexity lower bound $\text{LB}_{\text{lin}}(\delta)$ (equal to $T^\star(\boldsymbol{\mu})\text{kl}(\delta, 1 - \delta)$).

Except for XY-A, all algorithms implement the same stopping rule defined in (7) with threshold $\beta(t, \delta) = \log((\log(t) + 1)/\delta)$ (this threshold was initially suggested in [13], and is also used in [34] for CG-C and Lk-C). For XY-A, we use the stopping rule advocated in the corresponding papers. Refer to Appendix E for the detailed implementations.

In Table 1, we present the sample complexity (the number of samples gathered before the algorithm stops) averaged over 1000 runs for the various algorithms and for different confidence levels $\delta \in \{0.1, 0.01, 0.001, 0.0001\}$. In Appendix E, we provide detailed results, e.g. including box-plots (to show how confident we are about the values displayed in Table 1), as well as the empirical allocations achieved under the various algorithms.

Table 1: Sample complexity for the linear bandit benchmark example of [35], averaged over 1000 runs. Refer to Appendix E for details, including box-plots.

| | FWS | LT | CG-C | Lk-C | XY-A | RR | $\text{LB}_{\text{lin}}(\delta)$ |
|---|---|---|---|---|---|---|---|
| $\delta = 0.1$ | 1 030 | 919 | 2 498 | 2 319 | 7 016 | 5 451 | 359 |
| $\delta = 0.01$ | 1 614 | 1 464 | 3 501 | 3 431 | 7 779 | 8 814 | 920 |
| $\delta = 0.001$ | 2 229 | 1 982 | 4 324 | 4 326 | 9 090 | 12 101 | 1 408 |
| $\delta = 0.0001$ | 2 839 | 2 518 | 5 118 | 5 120 | 9 723 | 15 314 | 1 881 |

## 6   Conclusion

We have developed FWS, a computationally and statistically efficient algorithm for active pure exploration in bandit problems with fixed confidence. In each round, FWS performs a single iteration of a modified FW algorithm to approach an optimal allocation of arm draws predicted by the asymptotic lower bound. In the FWS algorithm, the FW iterations aim at maximizing a non-smooth function. Our main contribution is here to adapt the design of FW so that its convergence can be analyzed even for this non-smooth function. FW-based pure exploration algorithms have been discussed in the literature, with the belief that they would perform well. We confirm this belief, and even establish the asymptotic optimality of FWS in wide class of pure exploration problems.

Many interesting research directions could be investigated. Our analysis of the sample complexity in the moderate confidence regime has the advantage of being applicable to generic pure exploration problems, but may not be always tight. For bandits with specific structures, we may refine the analysis in this regime to get better upper bounds. We are also interested in investigating whether the iterative approach used in the FWS algorithm can be extended to more complex problems such as learning an optimal policy in MDPs, as well as to regret minimization problems. There, instance-specific regret lower bounds and the corresponding optimal exploration process are characterized by the solution of an optimization problem, just as in pure exploration problems.

## Acknowledgments and Disclosure of Funding

The authors would like to thank the anonymous reviewers whose comments helped us to improve the manuscript. R.-C Tzeng is supported by ERC Advanced Grant REBOUND (834862). A. Proutiere's research is supported by the Wallenberg AI, Autonomous Systems and Software Program (WASP) funded by the Knut and Alice Wallenberg Foundation. This work was also in part financially supported by Digital Futures.

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
