# A    Table of Notations

| Setting: pure exploration task | |
| --- | --- |
| $K$ | Number of arms |
| $[m]$ for any $m \in \mathbb{N}$ | The set $\{1, 2 \ldots, m\}$ |
| $\nu_k$ | Reward distribution for arm $k$ |
| $X_k(t)$ | Random reward received from pulling arm $k$ in round $t$ |
| $\boldsymbol{\mu} \in \mathbb{R}^K$ | Vector of the expected rewards of the various arms |
| $\Lambda$ | Set of all possible parameters $\boldsymbol{\mu}$ |
| $\mathcal{I}$ | Set of the answers |
| $i^\star(\boldsymbol{\mu})$ | Correct answer for parameter $\boldsymbol{\mu}$ |
| $\mathcal{S}_i$ | Set of parameters for which $i$ is the correct answer |
| $\delta$ | Targeted confidence level |

| Lower bound properties | |
| --- | --- |
| $\boldsymbol{\omega}$ | Vector of the proportions of arm draws |
| $\Sigma$ | Simplex |
| $\mathring{\Sigma}$ | Interior of $\Sigma$ |
| $\Sigma_\gamma$ | $\{\boldsymbol{\omega} \in \Sigma : \min_k \omega_k \geq \gamma\}$ |
| $\boldsymbol{e}_k$ | The $K$-dimensional vector with a 1 in the $k$-th coordinate and 0's elsewhere. |
| $\mathbb{E}_{\boldsymbol{\mu}}$ and $\mathbb{P}_{\boldsymbol{\mu}}$ | The expectation and probability measure corresponding to the parameter $\boldsymbol{\mu}$ |
| $\mathrm{Alt}(\boldsymbol{\mu})$ | Set of confusing parameters for $\boldsymbol{\mu}$ |
| $\boldsymbol{\omega}^\star(\boldsymbol{\mu})$ | Optimal allocation for parameter $\boldsymbol{\mu}$ |
| $T^\star(\boldsymbol{\mu})$ | Characteristic time for parameter $\boldsymbol{\mu}$ |
| $d(\mu, \mu')$ | KL divergence between the distributions parametrized by $\mu$ and $\mu'$ |
| $\mathrm{kl}(a, b)$ | KL divergent between two Bernoulli distributions of means $a$ and $b$ |

| Assumptions on the objective function | |
| --- | --- |
| $\mathcal{J}_i$ | Finite set of indexes associated with answer $i \in \mathcal{I}$ |
| $\mathcal{C}_j^i$ where $j \in \mathcal{J}_i$ | A convex set in $\Lambda \setminus \mathcal{S}_i$ |
| $f_j(\boldsymbol{\omega}, \boldsymbol{\mu})$ | $\inf_{\boldsymbol{\lambda} \in \mathcal{C}_j^{i^\star(\boldsymbol{\mu})}} \sum_{k=1}^K \omega_k d(\mu_k, \lambda_k)$ |
| $\mathrm{cl}(\mathcal{K})$ | The closure of $\mathcal{K}$ |
| $\overline{\boldsymbol{\lambda}_j(\boldsymbol{\omega}, \boldsymbol{\mu})}$ | $\arg\min_{\boldsymbol{\lambda} \in \mathrm{cl}(\mathcal{C}_j^{i^\star(\boldsymbol{\mu})})} \sum_{k=1}^K \omega_k d(\mu_k, \lambda_k)$ |
| $F_{\boldsymbol{\mu}}(\boldsymbol{\omega})$ | $\min_{j \in \mathcal{J}_{i^\star(\boldsymbol{\mu})}} f_j(\boldsymbol{\omega}, \boldsymbol{\mu})$ |
| $C_\psi(\mathcal{K})$ | $\sup_{\substack{\boldsymbol{x}, \boldsymbol{z} \in \mathcal{K} \\ \alpha \in (0,1] \\ \boldsymbol{y} = \boldsymbol{x} + \alpha(\boldsymbol{z} - \boldsymbol{x})}} \frac{1}{\alpha^2} \left[ \psi(\boldsymbol{x}) - \psi(\boldsymbol{y}) + \langle \boldsymbol{y} - \boldsymbol{x}, \nabla \psi(\boldsymbol{x}) \rangle \right]$ |
| $L$ | Upper bound of $\|\nabla_{\boldsymbol{\omega}} f_j(\boldsymbol{\omega}, \boldsymbol{\mu})\|_\infty$ |
| $D$ | Upper bound of $\gamma C_{f_j(\cdot, \boldsymbol{\mu})}(\Sigma_\gamma)$ |
| $\tau$ | Stopping rule |
| $\hat{i}_\tau$ | Decision rule |
| $\beta(t, \delta)$ | Stopping threshold |
| $c_1(\Lambda), c_2(\Lambda)$ | The constants needed for property of $\beta(t, \delta)$ (see (8)) |

| Notations for FWS | |
| --- | --- |
| $N_k(t)$ | Number of pulls of arm $k$ up to $t$ |
| $\omega_k(t)$ | $N_k(t)/t$ |
| $A_t$ | The arm pulled in time $t$ |
| $\hat{\mu}_k(t)$ | $\sum_{s=1}^t X_k(s) \mathbb{1}\{A_s = k\}/N_k(t)$ |
| $H_{F_{\boldsymbol{\mu}}}(\boldsymbol{\omega}, r)$ | r-subdifferential subspace |
| $\boldsymbol{x}(t)$ | The allocation tracked at time $t$ |
| $\boldsymbol{z}(t)$ | The solution for FW update at time $t$ |
| $\{r_t\}_{t \geq 1}$ | A sequence of positive numbers for FWS |
| $T_{\epsilon, L}$ | Constant needed for the assumption on $\{r_t\}_{t \geq 1}$ |
| $\kappa, E$ | Constants needed for Assumption 3 |

# B Proof of the Lower Bound of $\mathbb{E}_{\boldsymbol{\mu}}[\tau]$

**Definition 1.** *A $\delta$-PAC strategy with stopping rule $\tau$ and decision rule $\hat{\imath}_\tau$ is a strategy such that for any $\boldsymbol{\mu} \in \Lambda$, $\mathbb{P}_{\boldsymbol{\mu}}(\tau < \infty) = 1$ and $\mathbb{P}_{\boldsymbol{\mu}}(\hat{\imath}_\tau \neq i^\star(\boldsymbol{\mu})) \leq \delta$.*

**Proposition 2.** *Let $\delta \in (0,1)$ and $\boldsymbol{\mu} \in \Lambda$. For any $\delta$-PAC strategy,*

$$\mathbb{E}_{\boldsymbol{\mu}}[\tau] \geq T^\star(\boldsymbol{\mu}) \mathrm{kl}(\delta, 1 - \delta), \tag{13}$$

*where*

$$T^\star(\boldsymbol{\mu})^{-1} = \sup_{\boldsymbol{\omega} \in \Sigma} \inf_{\boldsymbol{\lambda} \in \mathrm{Alt}(\boldsymbol{\mu})} \sum_{k=1}^{K} \omega_k d(\mu_k, \lambda_k). \tag{14}$$

Note that $\mathrm{kl}(\delta, 1 - \delta) \approx \log(1/\delta)$ as $\delta \to 0$. Hence (13) yields that

$$\liminf_{\delta \to 0} \frac{\mathbb{E}_{\boldsymbol{\mu}}[\tau]}{\log(\frac{1}{\delta})} \geq T^\star(\boldsymbol{\mu}). \tag{15}$$

*Proof.* Consider a $\delta$-PAC strategy. Let $\boldsymbol{\lambda} \in \mathrm{Alt}(\boldsymbol{\mu})$. Let $\mathbb{P}_{\boldsymbol{\mu}}$ and $\mathbb{P}_{\boldsymbol{\lambda}}$ denote the probability measures generated by the parameter $\boldsymbol{\mu}$ and $\boldsymbol{\lambda}$, respectively. $\tau$ is a stopping time w.r.t. the filtration $(\mathcal{F}_t)_{t \geq 1}$ where $\mathcal{F}_t = \sigma(A_1, X_{A_1}(1), \ldots, A_t, X_{A_t}(t))$, and where $A_t$ is the arm selected under the algorithm in round $t$ and $X_{A_t}(t)$ is the corresponding reward. According to Definition 1, $\tau$ is almost surely finite, and Lemma 19 in [28] directly implies that

$$\sum_{k=1}^{K} \mathbb{E}_{\boldsymbol{\mu}}[N_k(\tau)] d(\mu_k, \lambda_k) \geq \mathrm{kl}(\mathbb{P}_{\boldsymbol{\mu}}(\mathcal{E}), \mathbb{P}_{\boldsymbol{\lambda}}(\mathcal{E})), \tag{16}$$

where $\mathcal{E}$ can be any $\mathcal{F}_\tau$-measurable event. With the choice, $\mathcal{E} = \{\hat{\imath}_\tau = i^\star(\boldsymbol{\lambda})\}$, the definition of $\delta$-PAC strategy and $\boldsymbol{\lambda} \in \mathrm{Alt}(\boldsymbol{\mu})$ imply that the right-hand side of inequality (16) is $\mathrm{kl}(\mathbb{P}_{\boldsymbol{\mu}}(\mathcal{E}), \mathbb{P}_{\boldsymbol{\lambda}}(\mathcal{E})) \geq \mathrm{kl}(\delta, 1 - \delta)$. (16) holds for any $\boldsymbol{\lambda} \in \mathrm{Alt}(\boldsymbol{\mu})$. Thus,

$$\mathrm{kl}(\delta, 1 - \delta) \leq \inf_{\boldsymbol{\lambda} \in \mathrm{Alt}(\boldsymbol{\mu})} \mathbb{E}_{\boldsymbol{\mu}}[\tau] \sum_{k=1}^{K} \frac{\mathbb{E}_{\boldsymbol{\mu}}[N_k(\tau)]}{\mathbb{E}_{\boldsymbol{\mu}}[\tau]} d(\mu_k, \lambda_k)$$

$$\leq \mathbb{E}_{\boldsymbol{\mu}}[\tau] \sup_{\boldsymbol{\omega} \in \Sigma} \inf_{\boldsymbol{\lambda} \in \mathrm{Alt}(\boldsymbol{\mu})} \sum_{k=1}^{K} \omega_k d(\mu_k, \lambda_k). \tag{17}$$

This completes the proof. $\qquad \square$

# C  A Generic Method to Verify Assumptions 2

Recall that Assumption 2 is:

**Assumption 2.** For all $\boldsymbol{\mu} \in \Lambda$,
(i) there exists $L > 0$ such that $\forall j \in \mathcal{J}_{i^\star(\boldsymbol{\mu})}$, $\left\| \nabla_{\boldsymbol{\omega}} f_j(\boldsymbol{\omega}, \boldsymbol{\mu}) \right\|_\infty \leq L$;
(ii) there exists $D > 0$ such that $\forall \gamma \in (0, 1/K)$ and $\forall j \in \mathcal{J}_{i^\star(\boldsymbol{\mu})}$, $C_{f_j(\cdot, \boldsymbol{\mu})}(\Sigma_\gamma) \leq \frac{D}{\gamma}$.

**Notation.** In this appendix, we often use the function $d : (\mu, \pi) \mapsto d(\mu, \pi)$, defined as the KL divergence between reward distributions parametrized $\mu$ and $\pi$. $\pi$ will denote its second argument. For example, $\frac{\partial d}{\partial \pi}(\mu, \lambda)$ is the partial derivate of $d$ w.r.t. its second argument evaluated at the point $(\mu, \lambda)$.

## C.1  Preliminaries: BAI in unstructured bandits

Before we introduce a generic way to check the assumption, we discuss the insightful case of the BAI problem in unstructured bandits with Bernoulli rewards. In this case, the gradients of $f_j$'s are:

$$\forall j \neq i^\star(\boldsymbol{\mu}), \ \nabla f_j(\boldsymbol{\omega}, \boldsymbol{\mu}) = d(\mu_{i^\star(\boldsymbol{\mu})}, m_j(\boldsymbol{\omega}, \boldsymbol{\mu})) \boldsymbol{e}_{i^\star(\boldsymbol{\mu})} + d(\mu_j, m_j(\boldsymbol{\omega}, \boldsymbol{\mu})) \boldsymbol{e}_j,$$

where

$$m_j(\boldsymbol{\omega}, \boldsymbol{\mu}) = \frac{\omega_{i^\star(\boldsymbol{\mu})} \mu_{i^\star(\boldsymbol{\mu})} + \omega_j \mu_j}{\omega_{i^\star(\boldsymbol{\mu})} + \omega_j}.$$

We deduce that $\left\| \nabla f_j(\boldsymbol{\omega}, \boldsymbol{\mu}) \right\|_\infty \leq L = \max_{k \neq i^\star(\boldsymbol{\mu})} d(\mu_{i^\star(\boldsymbol{\mu})}, \mu_k)$ for any $\boldsymbol{\omega} \in \Sigma$. Hence, this constant $L$ satisfies Assumption 2 (i) and depends $\boldsymbol{\mu}$ only. As for the Assumption 2 (ii), the Hessian $\nabla^2_{\boldsymbol{\omega}, \boldsymbol{\omega}} f_j(\boldsymbol{\omega})$ has elements almost all equal to 0 except for those corresponding to the basis $\boldsymbol{e}_{i^\star(\boldsymbol{\mu})}, \boldsymbol{e}_j$. Extracting the non-zero elements of the Hessian, we get:

$$\begin{bmatrix} \frac{m_j(\boldsymbol{\omega},\boldsymbol{\mu})-\mu_{i^\star(\boldsymbol{\mu})}}{m_j(\boldsymbol{\omega},\boldsymbol{\mu})(1-m_j(\boldsymbol{\omega},\boldsymbol{\mu}))} \frac{(\mu_{i^\star(\boldsymbol{\mu})}-\mu_j)\omega_j}{(\omega_{i^\star(\boldsymbol{\mu})}+\omega_j)^2} & \frac{m_j(\boldsymbol{\omega},\boldsymbol{\mu})-\mu_{i^\star(\boldsymbol{\mu})}}{m_j(\boldsymbol{\omega},\boldsymbol{\mu})(1-m_j(\boldsymbol{\omega},\boldsymbol{\mu}))} \frac{(\mu_j-\mu_{i^\star(\boldsymbol{\mu})})\omega_{i^\star(\boldsymbol{\mu})}}{(\omega_{i^\star(\boldsymbol{\mu})}+\omega_j)^2} \\[2em] \frac{m_j(\boldsymbol{\omega},\boldsymbol{\mu})-\mu_j}{m_j(\boldsymbol{\omega},\boldsymbol{\mu})(1-m_j(\boldsymbol{\omega},\boldsymbol{\mu}))} \frac{(\mu_{i^\star(\boldsymbol{\mu})}-\mu_j)\omega_j}{(\omega_{i^\star(\boldsymbol{\mu})}+\omega_j)^2} & \frac{m_j(\boldsymbol{\omega},\boldsymbol{\mu})-\mu_j}{m_j(\boldsymbol{\omega},\boldsymbol{\mu})(1-m_j(\boldsymbol{\omega},\boldsymbol{\mu}))} \frac{(\mu_j-\mu_{i^\star(\boldsymbol{\mu})})\omega_{i^\star(\boldsymbol{\mu})}}{(\omega_{i^\star(\boldsymbol{\mu})}+\omega_j)^2} \end{bmatrix}.$$

Notice that $\left| \frac{m_j(\boldsymbol{\omega},\boldsymbol{\mu})-\mu_{i^\star(\boldsymbol{\mu})}}{m_j(\boldsymbol{\omega},\boldsymbol{\mu})(1-m_j(\boldsymbol{\omega},\boldsymbol{\mu}))} \right| < 4\mu_{i^\star(\boldsymbol{\mu})}$. Thus $\left\| \nabla^2_{\boldsymbol{\omega},\boldsymbol{\omega}} f_j(\boldsymbol{\omega}, \boldsymbol{\mu}) \right\|_\infty$ (defined as the maximum over rows of the $L_1$-norm of a row), is smaller than $\frac{4L\mu_{i^\star(\boldsymbol{\mu})}}{\gamma}$ when $\boldsymbol{\omega} \in \Sigma_\gamma$. Invoking Lemma 1.2.2 in [3] (more precisely, in its proof), one can immediately deduce that $\nabla f_j$ is $\frac{D}{\gamma}$-Lipschitz, where $D = 4L\mu_{i^\star(\boldsymbol{\mu})}$. Finally, Lemma 7 in [24] implies that a function with gradient $\frac{D}{\gamma}$-Lipschitz satisfies Assumption 2 (ii).

From the above observations, we note that the value of $m_j(\boldsymbol{\omega}, \boldsymbol{\mu})$, or equivalently the most confusing parameter $\overline{\boldsymbol{\lambda}_j(\boldsymbol{\omega}, \boldsymbol{\mu})}$, plays an essential role in our assumptions. In view of Proposition 1, $\nabla_{\boldsymbol{\omega}} f_j(\boldsymbol{\omega}, \boldsymbol{\mu}) = \sum_k d(\mu_k, \overline{\boldsymbol{\lambda}_j(\boldsymbol{\omega}, \boldsymbol{\mu})}_k) \boldsymbol{e}_k$. First, if $d(\mu_k, \overline{\boldsymbol{\lambda}_j(\boldsymbol{\omega}, \boldsymbol{\mu})}_k)$ is bounded for any $k \in [K]$, $\boldsymbol{\omega} \in \overset{\circ}{\Sigma}$ and $j \in \mathcal{J}_{i^\star(\boldsymbol{\mu})}$, then Assumption 2 (i) holds because the $k$-th component of $\nabla_{\boldsymbol{\omega}} f_j(\boldsymbol{\omega}, \boldsymbol{\mu})$ is exactly $d(\mu_k, \overline{\boldsymbol{\lambda}_j(\boldsymbol{\omega}, \boldsymbol{\mu})}_k)$. Then, the chain rule yields:

$$\left( \nabla^2_{\boldsymbol{\omega}, \boldsymbol{\omega}} f_j(\boldsymbol{\omega}, \boldsymbol{\mu}) \right)_{k, k'} = \left( \frac{\partial d}{\partial \pi}(\mu_k, \overline{\boldsymbol{\lambda}_j(\boldsymbol{\omega}, \boldsymbol{\mu})}_k) \right) \frac{\partial}{\partial \omega_{k'}} \overline{\boldsymbol{\lambda}_j(\boldsymbol{\omega}, \boldsymbol{\mu})}_k. \tag{18}$$

For the BAI in unstructured bandits, we can derive Assumption 2 (ii) if $\frac{\partial d}{\partial \pi}(\mu_k, \overline{\boldsymbol{\lambda}_j(\boldsymbol{\omega}, \boldsymbol{\mu})}_k)$ is bounded and $\left\| \nabla_{\boldsymbol{\omega}} \overline{\boldsymbol{\lambda}_j(\boldsymbol{\omega}, \boldsymbol{\mu})} \right\|_\infty$ is shown to scale as $\mathcal{O}(\frac{1}{\min_k \omega_k})$. Sometimes, however, $\nabla_{\boldsymbol{\omega}} \overline{\boldsymbol{\lambda}_j(\boldsymbol{\omega}, \boldsymbol{\mu})}$ is not easy to compute. Next we provide a sufficient condition for Assumption 2 which is easier to check.

## C.2  Constraint function and a sufficient condition for Assumption 2

**Constraint function.** To state our sufficient condition, we introduce the constraint function $c_j^i$ to describe the set $\mathcal{C}_j^i$. Let us fix $i \in \mathcal{I}$ and $j \in \mathcal{J}_i$. The constraint function $c_j^i : \Lambda \setminus \mathcal{S}_i \to \mathbb{R}$ is a

mapping such that:

$$\mathcal{C}_j^i = \left\{ \boldsymbol{\mu} \in \Lambda \setminus \mathcal{S}_i : c_j^i(\boldsymbol{\mu}) > 0 \right\}.$$

Namely, we can define $\mathcal{C}_j^i$ by using $c_j^i$. For concreteness, we list below examples in which there is a constraint function.

*Example 1 – BAI in unstructured bandits with Bernoulli rewards.* For this task, we have $\Lambda = (0,1)^K$, $\mathcal{I} = \{1, \ldots, K\}$, and for all arm $i$, the set of parameters for which arm $i$ is the best arm is $\mathcal{S}_i = \{\boldsymbol{\mu} \in \Lambda : \mu_i > \mu_k, \forall k \neq i\}$. We have: $\Lambda \setminus \mathcal{S}_i = \cup_{j \in \mathcal{J}_i} \mathcal{C}_j^i$ where $\mathcal{J}_i = \mathcal{I} \setminus \{i\}$ is the set of arms different than $i$ and $\mathcal{C}_j^i = \{\boldsymbol{\mu} \in \Lambda : \mu_j > \mu_i\}$. Thus a constraint function is $c_j^i(\boldsymbol{\mu}) = \mu_j - \mu_i$.

*Example 2 – Threshold bandits.* In this task, the objective is to identify all arms whose average rewards are above a threshold $\mathfrak{I}$. With Bernoulli rewards, we have $\Lambda = (0,1)^K$ and the set of possible answers is $\mathcal{I} = 2^{[K]}$. We can decompose the set $\Lambda \setminus \mathcal{S}_{\mathcal{A}} = \cup_{k \in [K]} \mathcal{C}_k^{\mathcal{A}}$, where

$$\mathcal{C}_k^{\mathcal{A}} = \begin{cases} \{\boldsymbol{\mu} \in \Lambda \setminus \mathcal{S}_{\mathcal{A}} : \mu_k > \mathfrak{I}\} & \text{if } k \notin \mathcal{A}, \\ \{\boldsymbol{\mu} \in \Lambda \setminus \mathcal{S}_{\mathcal{A}} : \mu_k < \mathfrak{I}\} & \text{if } k \in \mathcal{A}. \end{cases}$$

Then a constraint function is: $c_k^{\mathcal{A}}(\boldsymbol{\mu}) = (\mathbb{1}\{k \notin \mathcal{A}\} - \mathbb{1}\{k \in \mathcal{A}\})(\mu_k - \mathfrak{I})$.

*Example 3 – Top-$m$ bandits.* The task is to identify the best $m$ arms. Assuming Bernoulli rewards, we have $\Lambda = \{\boldsymbol{\mu} \in (0,1) : \mu_{[1]} \geq \ldots \mu_{[m]} > \mu_{[m+1]}\}$, where $\mu_{[k]}$ denotes the average reward of the arm with the $k$-th highest reward. The set of possible answers is $\mathcal{I} = \{\mathcal{A} \in [K] : |\mathcal{A}| = m\}$. Define $\mathcal{J}_{\mathcal{A}} = \{j \notin \mathcal{A}\}$ and $\mathcal{C}_j^{\mathcal{A}} = \{\boldsymbol{\mu} \in \Lambda \setminus \mathcal{S}_{\mathcal{A}} : \mu_j > \min_{k \in \mathcal{A}} \mu_k\}$. Then, we have: $\Lambda \setminus \mathcal{S}_{\mathcal{A}} = \cup_{j \notin \mathcal{J}_{\mathcal{A}}} \mathcal{C}_j^{\mathcal{A}}$ and a constraint function can be $c_j^{\mathcal{A}}(\boldsymbol{\mu}) = \mu_j - \min_{k \in \mathcal{A}} \mu_k$.

As illustrated in the above examples, the constraint functions $c_j^i$ depend on the pure exploration task, but are simple and usually differentiable. The following lemma provides a sufficient condition for 2, involving the constraint functions only. In all the examples considered in this paper, this lemma can be applied. Its proof, provided at the end of this appendix, combines the Lagrange multiplier theorem and the implicit function theorem, and leverages similar techniques as those developed in [14, 2].

**Lemma 1.** *Let $\boldsymbol{\mu} \in \Lambda$. Assume that, for any $j \in \mathcal{J}_{i^\star(\boldsymbol{\mu})}$,*
*(a) $c_j^{i^\star(\boldsymbol{\mu})}$ is twice differentiable at the point $(\overline{\boldsymbol{\lambda}_j(\boldsymbol{\omega}, \boldsymbol{\mu})})$ and $\nabla^2 c_j^{i^\star(\boldsymbol{\mu})}(\overline{\boldsymbol{\lambda}_j(\boldsymbol{\omega}, \boldsymbol{\mu})}) = 0, \forall \boldsymbol{\omega} \in \overset{\circ}{\Sigma}$,*
*(b) the reward distributions are Gaussian or Bernoulli,*
*(c) there is a constant $M > 0$ such that*

$$\max\left\{\left|d(\mu_k, \overline{\boldsymbol{\lambda}_j(\boldsymbol{\omega}, \boldsymbol{\mu})_k})\right|, \left|\frac{\partial d}{\partial \pi}(\mu_k, \overline{\boldsymbol{\lambda}_j(\boldsymbol{\omega}, \boldsymbol{\mu})_k})\right|\right\} \leq M, \ \forall k \in [K], \boldsymbol{\omega} \in \overset{\circ}{\Sigma}.$$

*Then Assumption 2 holds.*

### C.3 Applications of Lemma 1

We apply Lemma 1 to verify Assumption 2 for the pure exploration tasks presented Appendix D, E, F, and G.

**Conditions (a) and (b).** These conditions hold trivially because in all examples, the constraint functions are linear, and we consider only Bernoulli or Gaussian rewards.

**Condition (c).** First observe that:
for Bernoulli rewards,

$$d(\mu, \pi) = \mu \log \frac{\mu}{\pi} + (1 - \mu) \log \frac{1 - \mu}{1 - \pi} \text{ and } \frac{\partial d}{\partial \pi}(\mu, \pi) = \frac{-\mu}{\pi} + \frac{1 - \mu}{1 - \pi}, \ \forall \mu, \pi \in (0, 1),$$

and for Gaussian rewards,

$$d(\mu, \pi) = \frac{1}{2}(\mu - \pi)^2 \text{ and } \frac{\partial d}{\partial \pi}(\mu, \pi) = \pi - \mu, \ \forall \mu, \pi \in \mathbb{R}.$$

In the case of BAI for unstructured bandits (Appendix D), $\overline{\boldsymbol{\lambda}_j(\boldsymbol{\omega}, \boldsymbol{\mu})}_k = \mu_k$ if $k \notin \{i^\star(\boldsymbol{\mu}), j\}$ and $\overline{\boldsymbol{\lambda}_j(\boldsymbol{\omega}, \boldsymbol{\mu})}_k = m_j(\boldsymbol{\omega}, \boldsymbol{\mu})$ otherwise, where $m_j(\boldsymbol{\omega}, \boldsymbol{\mu}) = \frac{\omega_{i^\star(\boldsymbol{\mu})}\mu_{i^\star(\boldsymbol{\mu})} + \omega_j\mu_j}{\omega_{i^\star(\boldsymbol{\mu})} + \omega_j}, \forall \boldsymbol{\omega} \in \mathring{\Sigma}$. Hence, (c) holds for $m_j(\boldsymbol{\omega}, \boldsymbol{\mu})$ is bounded in the interval $[\mu_j, \mu_{i^\star(\boldsymbol{\mu})}]$.

For BAI in linear bandits (Appendix E), according to (12), we have that for any $\boldsymbol{\omega} \in \mathring{\Sigma}$,

$$
\left\| \overline{\boldsymbol{\lambda}_j^{\text{BAI}}(\boldsymbol{\omega}, \boldsymbol{\mu})} \right\|_\infty \leq \|\boldsymbol{\mu}\|_\infty + \left\| \left( \frac{\langle \boldsymbol{a}_{i^\star} - \boldsymbol{a}_j, \boldsymbol{\mu} \rangle}{\|\boldsymbol{a}_{i^\star} - \boldsymbol{a}_j\|_{V_{\boldsymbol{\omega}}^{-1}}^2} V_{\boldsymbol{\omega}}^{-1} \right)(\boldsymbol{a}_j - \boldsymbol{a}_{i^\star}) \right\|_\infty
$$
$$
\leq \|\boldsymbol{\mu}\|_\infty + \|\boldsymbol{a}_{i^\star} - \boldsymbol{a}_j\|_\infty \|\boldsymbol{\mu}\|_\infty < \infty.
$$

Thus, (c) holds. The condition can be checked similarly for the threshold linear bandits. As for BAI in Lipschitz bandits (Appendix F), each component of the most confusing parameter (see (31)) is bounded in the interval of $[\min_k \mu_k, \max_k \mu_k]$, and thus, (c) holds. Finally, for the threshold bandit problem with monotone structure and the top-m arm problem in dueling bandits (Appendix G), (c) directly holds as the most confusing parameter $\overline{\boldsymbol{\lambda}_j(\boldsymbol{\omega}, \boldsymbol{\mu})}$ and $\overline{\boldsymbol{\lambda}_\sigma(\boldsymbol{\omega}, \boldsymbol{\mu})}$ are fixed for any $\boldsymbol{\omega} \in \mathring{\Sigma}$.

### C.4 Proof of Lemma 1

We first prove that Assumption 2 (i) holds. Let $L = M$. Observe that $\|\nabla_{\boldsymbol{\omega}} f_j(\boldsymbol{\omega}, \boldsymbol{\mu})\|_\infty = \max_k \left| d(\mu_k, \overline{\boldsymbol{\lambda}_j(\boldsymbol{\omega}, \boldsymbol{\mu})}_k) \right| < M$, then (i) holds directly.

Let us verify Assumption 2 (ii). Let $\gamma \in (0, \frac{1}{K})$. We will prove that for $\boldsymbol{\omega} \in \Sigma_\gamma$, $\left\| \nabla_{\boldsymbol{\omega}, \boldsymbol{\omega}}^2 f_j(\boldsymbol{\omega}, \boldsymbol{\mu}) \right\|_\infty$ is bounded by $\frac{D}{\gamma}$ by some constant $D > 0$. This will imply that Assumption 2 (ii) holds, see e.g.,[3, 24].

We have:

$$
\begin{cases}
\frac{\partial^2 d}{\partial \pi^2}(\mu, \pi) = \frac{\mu}{\pi^2} + \frac{1-\mu}{(1-\pi)^2} \geq 1, \forall \mu, \pi \in (0, 1), & \text{for Bernoulli rewards,} \\
\frac{\partial^2 d}{\partial \pi^2}(\mu, \pi) = 1, \ \forall \mu, \pi \in \mathbb{R}, & \text{for Gaussian rewards.}
\end{cases} \tag{19}
$$

We deduce that $\frac{\partial^2 d}{\partial \pi^2}(\mu, \pi) \geq 1$. Now, recall that $\overline{\boldsymbol{\lambda}_j(\boldsymbol{\omega}, \boldsymbol{\mu})}$ is the solution of the following optimization problem:

$$
\min_{\boldsymbol{\pi} \in \Lambda, c_j^{i^\star(\boldsymbol{\mu})}(\boldsymbol{\pi}) \geq 0} \sum_k \omega_k d(\mu_k, \pi_k).
$$

Let $\mathcal{L} : \mathbb{R}^K \times \mathbb{R}^K \times \mathbb{R} \times \mathbb{R}^K \mapsto \mathbb{R}^K$ be the Lagrangian defined as

$$
\mathcal{L}(\boldsymbol{\omega}, \boldsymbol{\mu}, \alpha, \boldsymbol{\pi}) = \sum_k \omega_k d(\mu_k, \pi_k) - \alpha c_j^{i^\star(\boldsymbol{\mu})}(\boldsymbol{\pi}).
$$

The solution $\overline{\boldsymbol{\lambda}_j(\boldsymbol{\omega}, \boldsymbol{\mu})}$ can be identified by solving $\nabla_\alpha \mathcal{L}(\boldsymbol{\omega}, \boldsymbol{\mu}, \alpha, \overline{\boldsymbol{\lambda}_j(\boldsymbol{x}, \boldsymbol{\mu})}) = -c_j^{i^\star(\boldsymbol{\mu})}(\overline{\boldsymbol{\lambda}_j(\boldsymbol{\omega}, \boldsymbol{\mu})})$ and

$$
\nabla_{\boldsymbol{\pi}} \mathcal{L}(\boldsymbol{\omega}, \boldsymbol{\mu}, \alpha, \overline{\boldsymbol{\lambda}_j(\boldsymbol{\omega}, \boldsymbol{\mu})}) = \sum_k \omega_k \frac{\partial d}{\partial \pi}(\mu_k, \overline{\boldsymbol{\lambda}_j(\boldsymbol{\omega}, \boldsymbol{\mu})}_k) \boldsymbol{e}_k - \alpha \nabla c_j^{i^\star(\boldsymbol{\mu})}(\overline{\boldsymbol{\lambda}_j(\boldsymbol{\omega}, \boldsymbol{\mu})}). \tag{20}
$$

By differentiating (20) with respect to $\boldsymbol{\pi}$, we get the Hessian of $\nabla_{\boldsymbol{\pi}, \boldsymbol{\pi}}^2 \mathcal{L}(\boldsymbol{\omega}, \boldsymbol{\mu}, \alpha, \overline{\boldsymbol{\lambda}_j(\boldsymbol{\omega}, \boldsymbol{\mu})})$: it is a diagonal matrix and for any $k \in [K]$, its $(k, k)$-th entry is

$$
\nabla_{\boldsymbol{\pi}, \boldsymbol{\pi}}^2 \mathcal{L}(\boldsymbol{\omega}, \boldsymbol{\mu}, \alpha, \overline{\boldsymbol{\lambda}_j(\boldsymbol{\omega}, \boldsymbol{\mu})})_{k,k} = \omega_k \frac{\partial^2 d}{\partial^2 \pi}(\mu_k, \overline{\boldsymbol{\lambda}_j(\boldsymbol{\omega}, \boldsymbol{\mu})}_k) - \alpha \nabla^2 c_j^{i^\star(\boldsymbol{\mu})}(\overline{\boldsymbol{\lambda}_j(\boldsymbol{\omega}, \boldsymbol{\mu})})
$$
$$
= \omega_k \frac{\partial^2 d}{\partial^2 \pi}(\mu_k, \overline{\boldsymbol{\lambda}_j(\boldsymbol{\omega}, \boldsymbol{\mu})}_k)
$$
$$
\geq \omega_k. \tag{21}
$$

Since $\boldsymbol{\omega} \in \Sigma_\gamma$, we deduce from (21) that $\nabla_{\boldsymbol{\pi}, \boldsymbol{\pi}}^2 L(\boldsymbol{\omega}, \boldsymbol{\mu}, \alpha, \overline{\boldsymbol{\lambda}_j(\boldsymbol{x}, \boldsymbol{\mu})})$ is invertible, and that we can apply the implicit function theorem:

$$
\nabla_{\boldsymbol{\omega}} \overline{\boldsymbol{\lambda}_j(\boldsymbol{\omega}, \boldsymbol{\mu})} = - \left( \nabla_{\boldsymbol{\pi}, \boldsymbol{\pi}}^2 \mathcal{L}(\boldsymbol{\omega}, \boldsymbol{\mu}, \alpha, \overline{\boldsymbol{\lambda}_j(\boldsymbol{\omega}, \boldsymbol{\mu})}) \right)^{-1} \left( \nabla_{\boldsymbol{\omega}} \nabla_{\boldsymbol{\pi}} \mathcal{L}(\boldsymbol{\omega}, \boldsymbol{\mu}, \alpha, \overline{\boldsymbol{\lambda}_j(\boldsymbol{\omega}, \boldsymbol{\mu})}) \right). \tag{22}
$$

In addition, we have:

$$\left\| \left( \nabla^2_{\boldsymbol{\pi},\boldsymbol{\pi}} \mathcal{L}(\boldsymbol{\omega}, \boldsymbol{\mu}, \alpha, \overline{\boldsymbol{\lambda}_j(\boldsymbol{\omega}, \boldsymbol{\mu})}) \right)^{-1} \right\|_\infty \leq \frac{1}{\gamma}. \tag{23}$$

To derive an upper bound of $\left\| \nabla_{\boldsymbol{\omega}} \overline{\boldsymbol{\lambda}_j(\boldsymbol{\omega}, \boldsymbol{\mu})} \right\|_\infty$, we compute the second factor in the r.h.s. of (22) by differentiating (20) with respect to $\boldsymbol{\omega}$. We can see that $\nabla_{\boldsymbol{\omega}} \nabla_{\boldsymbol{\pi}} \mathcal{L}(\boldsymbol{\omega}, \boldsymbol{\mu}, \alpha, \overline{\boldsymbol{\lambda}_j(\boldsymbol{\omega}, \boldsymbol{\mu})}$ is a diagonal matrix whose $(k, k)$-th entry is $\frac{\partial d}{\partial \pi}(\mu_k, \overline{\boldsymbol{\lambda}_j(\boldsymbol{\omega}, \boldsymbol{\mu})_k})$ for any $k \in [K]$. Combining this observation with (22)-(23), we deduce that

$$\left\| \nabla_{\boldsymbol{\omega}} \overline{\boldsymbol{\lambda}_j(\boldsymbol{\omega}, \boldsymbol{\mu})} \right\|_\infty \leq \frac{\max_k \left| \frac{\partial d}{\partial \pi}(\mu_k, \overline{\boldsymbol{\lambda}_j(\boldsymbol{\omega}, \boldsymbol{\mu})_k}) \right|}{\gamma} \leq \frac{M}{\gamma}, \tag{24}$$

where the last inequality stems from (c). Finally, using (24), (c), and (18), we can upper bound $\left\| \nabla^2_{\boldsymbol{\omega}, \boldsymbol{\omega}} f_j(\boldsymbol{\omega}, \boldsymbol{\mu}) \right\|_\infty$ as:

$$\left\| \nabla^2_{\boldsymbol{\omega}, \boldsymbol{\omega}} f_j(\boldsymbol{\omega}, \boldsymbol{\mu}) \right\|_\infty = \max_k \sum_{k'=1}^K \left| \left( \frac{\partial d}{\partial \pi}(\mu_k, \overline{\boldsymbol{\lambda}_j(\boldsymbol{\omega}, \boldsymbol{\mu})_k}) \right) \frac{\partial}{\partial \omega_{k'}} \overline{\boldsymbol{\lambda}_j(\boldsymbol{\omega}, \boldsymbol{\mu})_k} \right| \leq \frac{M^2 K}{\gamma}.$$

We have proved Assumption 2 (ii) with $D = M^2 K$. $\qquad\qquad\qquad\qquad\qquad\qquad\square$

# D  BAI in Unstructured Bandits

**About all our experiments.** All the experiments are executed on a machine with Intel Core i5 at 1.8 GHz with 8 GB RAM. We implemented all the algorithms[3] in `Julia 1.5.4` and part of the baselines are taken from the implementation by [31, 45]. Throughout the experiments, we fix the parameters of our Frank-Wolfe-based sampling (`FWS`): $r_t = t^{-0.9}/K$, where $K$ is the number of arms.

## D.1  Preliminaries and competing algorithms

The BAI in unstructured bandits with Bernoulli rewards has been treated in Example 1 (in the main document). It is obtained by assuming $\Lambda = \{\boldsymbol{\mu} \in (0,1)^K : \exists i \in [K] \text{ s.t } \mu_i > \mu_k, \forall k \neq i\}$. The set of answers is $\mathcal{I} = [K]$ and $i^\star(\boldsymbol{\mu}) = \operatorname{argmax}_{k \in [K]} \mu_k$ and hence $\mathcal{S}_i = \{\boldsymbol{\mu} \in (0,1)^K : \mu_i > \mu_k, \forall k \neq i\}$. For this BAI, we set $\mathcal{J}_i = [K] \setminus i$ and $\mathcal{C}_j^i = \{\boldsymbol{\mu} \in \Lambda : \mu_j > \mu_i\}$. Obviously, $\{\mathcal{S}_i\}_{i \in [K]}$ are open sets and $\{\mathcal{C}_j^i\}_{j \neq i}$ are convex sets, so Assumption 1 holds. As already mentioned, we have: $\forall (\boldsymbol{\omega}, \boldsymbol{\mu}) \in \mathring{\Sigma} \times \mathcal{S}_{i^\star(\boldsymbol{\mu})}, \forall j \neq i^\star(\boldsymbol{\mu})$,

$$\overline{\boldsymbol{\lambda}_j(\boldsymbol{\omega}, \boldsymbol{\mu})} = m_j(\boldsymbol{\omega}, \boldsymbol{\mu})\boldsymbol{e}_{i^\star(\boldsymbol{\mu})} + m_j(\boldsymbol{\omega}, \boldsymbol{\mu})\boldsymbol{e}_j + \sum_{k \neq j, i^\star(\boldsymbol{\mu})} \mu_k \boldsymbol{e}_k,$$

$$f_j(\boldsymbol{\omega}, \boldsymbol{\mu}) = \omega_{i^\star(\boldsymbol{\mu})} d(\mu_{i^\star(\boldsymbol{\mu})}, m_j(\boldsymbol{\omega}, \boldsymbol{\mu})) + \omega_j d(\mu_j, m_j(\boldsymbol{\omega}, \boldsymbol{\mu})),$$

$$\nabla f_j(\omega, \boldsymbol{\mu}) = d(\mu_{i^\star(\boldsymbol{\mu})}, m_j(\boldsymbol{\omega}, \boldsymbol{\mu}))\boldsymbol{e}_{i^\star(\boldsymbol{\mu})} + d(\mu_j, m_j(\boldsymbol{\omega}, \boldsymbol{\mu}))\boldsymbol{e}_j,$$

where

$$m_j(\boldsymbol{\omega}, \boldsymbol{\mu}) = \frac{\omega_{i^\star(\boldsymbol{\mu})}\mu_{i^\star(\boldsymbol{\mu})} + \omega_j \mu_j}{\omega_{i^\star(\boldsymbol{\mu})} + \omega_j}.$$

Assumption 2 is verified in Appendix C.

**`FWS` algorithm and the FW update.** To illustrate the implementation of `FWS`, we provide an example on how the FW update (11) is implemented. This update is translated into a zero-sum game that can be solved using any LP solver. Refer to Appendix H for a discussion on how to get this game in general pure exploration problems.

Let $K = 3$. Assume that we are in round $t$, and that we wish to apply the FW update:

$$\boldsymbol{z}(t) \leftarrow \operatorname*{argmax}_{\boldsymbol{z} \in \Sigma} \min_{h \in H_{F_{\hat{\boldsymbol{\mu}}(t-1)}}(\boldsymbol{x}(t-1), r_t)} \langle \boldsymbol{z} - \boldsymbol{x}(t-1), h \rangle \quad \text{(ties broken arbitrarily)}.$$

Further assume that $i^\star(\hat{\boldsymbol{\mu}}(t-1)) = 3$ and that $f_1(\boldsymbol{x}(t-1), \hat{\boldsymbol{\mu}}(t-1)) \vee f_2(\boldsymbol{x}(t-1), \hat{\boldsymbol{\mu}}(t-1)) < F_t(\boldsymbol{x}(t-1)) + r_t$. We then create a $3 \times 2$ payoff matrix $M$, whose $(k,j)$-th entry is $M_{k,j} = \langle \boldsymbol{e}_k - \boldsymbol{x}(t-1), \nabla f_j(\boldsymbol{x}(t-1), \hat{\boldsymbol{\mu}}(t-1)) \rangle$ for all $k = 1, 2, 3$ and $j = 1, 2$. In this example, the update can be formulated as

$$\max_{\boldsymbol{z} \in \Sigma} \min_{\boldsymbol{y} \in \mathbb{R}^2} \boldsymbol{z}^\top M \boldsymbol{y} \tag{25}$$

$$\text{s.t. } y_1, y_2 \geq 0 \text{ and } y_1 + y_2 = 1.$$

Let $(\boldsymbol{z}^\star, \boldsymbol{y}^\star)$ denote the solution of (25). Then we have

$$\boldsymbol{z}(t) = \boldsymbol{x}(t-1) + \sum_{k=1}^3 z_k^\star(\boldsymbol{e}_k - \boldsymbol{x}(t-1)) = \boldsymbol{z}^\star.$$

A standard method to solve the zero-sum game (25) is to apply any LP solver to the following problem [37, 48, 52]:

$$\max_{\boldsymbol{z} \in \Sigma, u \in \mathbb{R}} u \tag{26}$$

$$\text{s.t. } (\boldsymbol{z}^\top M)_1, (\boldsymbol{z}^\top M)_2 \geq u.$$

The solution of (26) provides $\boldsymbol{z}^\star$ and the value of (25). Appendix H explains why and gives a short introduction for the transformation of a zero-sum game to an LP.

**Competing algorithms.** The list of algorithms used for comparison is provided below.

---

[3]https://github.com/rctzeng/NeurIPS2021-Fast-Pure-Exploration-via-Frank-Wolfe

- `FWS`: Our algorithm with parameters $r_t = t^{-0.9}/K$, where $K$ is the number of arms.

- `T-D`: Track-and-Stop [28] with D-Tracking implemented by [31].

- `D-C`: AdaHedge as the $\boldsymbol{\lambda}$-player and Best-Response as the $\boldsymbol{\omega}$-player described in Section 3.1 in [12] implemented by [31].

- `M-C`: Lazy Mirror Ascent by [38] implemented by [31]. This method is very sensitive to the learning rate $\eta_t = 1/(L\sqrt{t})$ ($L > 0$ is a hyperparameter). Note that the implementation [31] chooses $L$ assuming the knowledge of $\boldsymbol{\mu}$. This choice is for experimental comparison only and cannot be used in real-world scenarios.

- `O-C`: Optimistic Track and Stop [12] implemented by [31].

- `RR`: Sample arms in a round-robin manner.

- `LB(δ)`: $T^\star(\boldsymbol{\mu})\mathrm{kl}(\delta, 1 - \delta)$.

The Track-and-Stop algorithm has two versions, one with D-tracking (directly tracking the optimal allocation) and another one with C-tracking (tracking the cumulative optimal allocation). We found that D-tracking always performs better than C-tracking numerically. Hence, we report the performance of Track-and-Stop with D-tracking only.

**Stopping rule.** In all the algorithms, we use the same stopping rule (7) for unstructured bandits, with the same threshold $\beta(t, \delta) = \log((\log(t) + 1)/\delta)$, suggested in [20].

## D.2 Numerical experiments

**Bernoulli rewards.** In the first experiment, we consider Bernoulli rewards with $\boldsymbol{\mu} = [0.3, 0.21, 0.2, 0.19, 0.18]$ used in [28]. We average our results over 3000 runs. In Table 2, we provide the sample complexity for various confidence levels $\delta \in \{0.1, 0.01, 0.001, 0.0001\}$. To provide a more detailed comparison, at the confidence level $\delta = 0.01$, we show the sample complexity in box-plot in Figure 3a and compare the allocation of arm draws achieved under the various algorithms in Table 3.

In Figure 1, we plot the number of rounds (the median over all runs) `FWS` is in force exploration or the $r$-subdifferential subspace used in `FWS` contains the gradient of only one function (in this round, our FW update coincides with the traditional FW update as if the objective function was smooth). In Figure 2, we provide the distribution of the number of functions involved in the FW updates. It is interesting to note that 60% of the time our update differs from the usual FW update.

Table 2: Sample complexity in unstructured bandits with Bernoulli rewards with $\boldsymbol{\mu} = [0.3, 0.21, 0.2, 0.19, 0.18]$ and $\delta = 0.01$, averaged over 3000 runs.

|  | FWS | T-D | D-C | M-C | O-C | RR | LB($\delta$) |
|---|---|---|---|---|---|---|---|
| $\delta = 0.1$ | 1 365 | 1 337 | 1 859 | 1 668 | 1 818 | 2 326 | 574 |
| $\delta = 0.01$ | 2 125 | 2 066 | 2 674 | 2 509 | 2 706 | 3 460 | 1 471 |
| $\delta = 0.001$ | 2 899 | 2 823 | 3 465 | 3 362 | 3 584 | 4 555 | 2 252 |
| $\delta = 0.0001$ | 3 645 | 3 589 | 4 279 | 4 231 | 4 457 | 5 621 | 3 008 |

Table 3: Allocation of arm draws in unstructured bandits with Bernoulli rewards with $\boldsymbol{\mu} = [0.3, 0.21, 0.2, 0.19, 0.18]$ and $\delta = 0.01$, averaged over 3000 runs.

|  | FWS | T-D | D-C | M-C | O-C | RR | $\boldsymbol{\omega}^\star(\boldsymbol{\mu})$ |
|---|---|---|---|---|---|---|---|
| $\boldsymbol{a}_1$ | 34.08 | 35.60 | 29.40 | 31.72 | 31.31 | 20.00 | 32.59 |
| $\boldsymbol{a}_2$ | 24.82 | 23.47 | 21.46 | 22.37 | 20.94 | 20.00 | 25.15 |
| $\boldsymbol{a}_3$ | 17.45 | 17.30 | 17.94 | 17.81 | 17.99 | 20.00 | 17.66 |
| $\boldsymbol{a}_4$ | 13.38 | 13.22 | 16.11 | 15.06 | 15.79 | 20.00 | 13.24 |
| $\boldsymbol{a}_5$ | 10.27 | 10.42 | 15.08 | 13.04 | 13.97 | 20.00 | 10.36 |

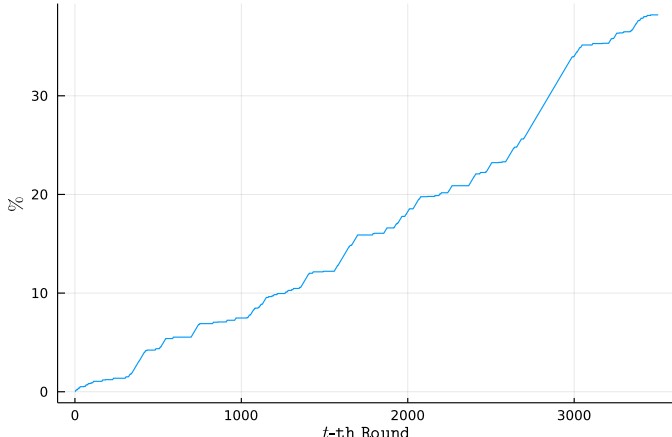

Figure 1: Number of rounds where `FWS` is in forced exploration or the FW update in `FWS` corresponds to the usual FW update. BAI in unstructured bandits with Bernoulli rewards and $\boldsymbol{\mu} = [0.3, 0.21, 0.2, 0.19, 0.18]$, $\delta = 0.0001$.

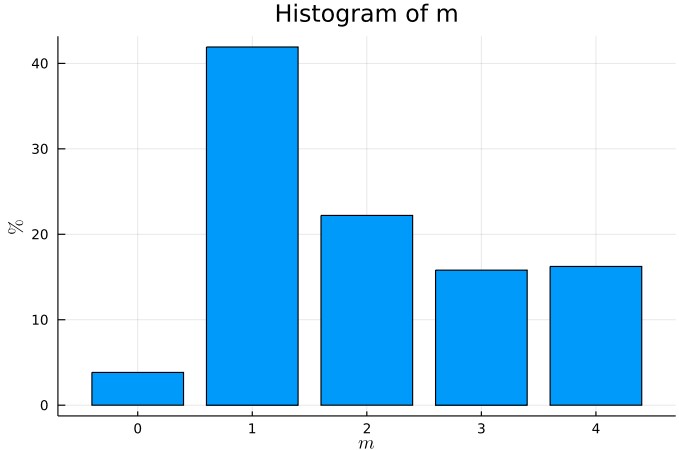

Figure 2: Proportions of rounds where we have $m$ functions in the linear program involved in the FW updates (i.e. $m = \left| \{ j : f_j(\boldsymbol{x}(t), \hat{\boldsymbol{\mu}}(t)) < F_{\hat{\boldsymbol{\mu}}(t-1)}(\boldsymbol{x}(t)) + r_t \} \right|$, and $m = 0$ in forced exploration). BAI in unstructured bandits with Bernoulli rewards and $\boldsymbol{\mu} = [0.3, 0.21, 0.2, 0.19, 0.18]$, $\delta = 0.0001$.

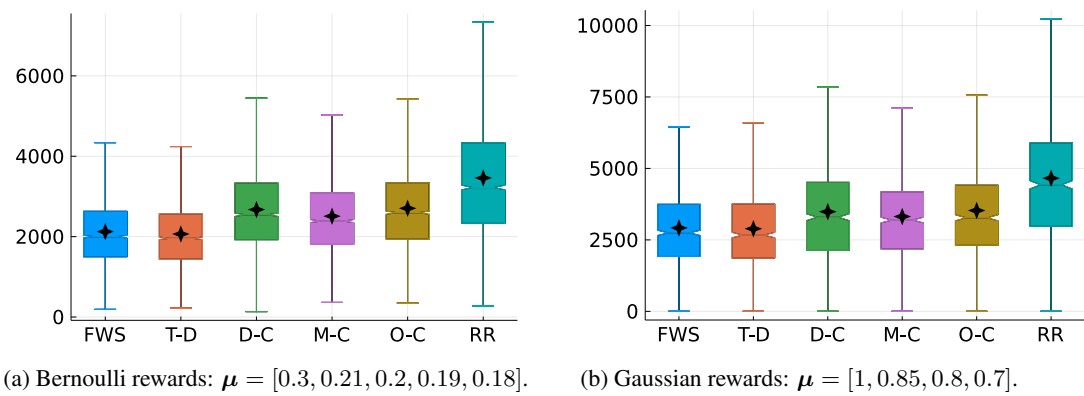

(a) Bernoulli rewards: $\boldsymbol{\mu} = [0.3, 0.21, 0.2, 0.19, 0.18]$.    (b) Gaussian rewards: $\boldsymbol{\mu} = [1, 0.85, 0.8, 0.7]$.

Figure 3: Sample complexity for the unstructured best-arm identification problem at $\delta = 0.01$, plotted in boxplots where the stars represent the averaged sample complexity and the outliers are hidden.

**Gaussian rewards.** In the second experiment, we consider Gaussian rewards with means $\boldsymbol{\mu} = [1, 0.85, 0.8, 0.7]$ and unit variance as proposed in [38]. The results are averaged over 1000 runs. In Table 4, we compare the sample complexity for $\delta \in \{0.1, 0.01, 0.001, 0.0001\}$. At the confidence level $\delta = 0.01$, we show the sample complexity in box-plot in Figure 3b and compare the allocation of arm draws achieved under the various algorithms in Table 5.

Table 4: Sample complexity in unstructured bandits with Gaussian rewards and $\boldsymbol{\mu} = [1, 0.85, 0.8, 0.7]$ and $\delta = 0.01$, averaged over 1000 runs.

|  | FWS | T-D | D-C | M-C | O-C | RR | LB($\delta$) |
|---|---|---|---|---|---|---|---|
| $\delta = 0.1$ | 1 857 | 1 874 | 2 286 | 2 160 | 2 272 | 2 994 | 791 |
| $\delta = 0.01$ | 2 919 | 2 891 | 3 487 | 3 313 | 3 528 | 4 659 | 2 026 |
| $\delta = 0.001$ | 3 990 | 4 000 | 4 640 | 4 449 | 4 732 | 6 219 | 3 101 |
| $\delta = 0.0001$ | 5 056 | 5 038 | 5 739 | 5 575 | 5 896 | 7 855 | 4 142 |

Table 5: Allocation of arm draws in unstructured bandits with Gaussian rewards $\boldsymbol{\mu} = [1, 0.85, 0.8, 0.7]$ and $\delta = 0.01$, averaged over 1000 runs.

|  | FWS | T-D | D-C | M-C | O-C | RR | $\omega^\star(\boldsymbol{\mu})$ |
|---|---|---|---|---|---|---|---|
| $a_1$ | 41.05 | 42.00 | 34.37 | 37.46 | 39.70 | 25.00 | 41.25 |
| $a_2$ | 36.01 | 36.04 | 30.94 | 32.47 | 31.60 | 25.00 | 37.93 |
| $a_3$ | 16.94 | 16.11 | 19.56 | 18.51 | 18.92 | 25.00 | 15.21 |
| $a_4$ | 6.00 | 5.85 | 15.13 | 11.56 | 9.78 | 25.00 | 5.61 |

In all these results, we observe that FWS and T-D exhibit very close performance. FWS is as efficient as T-D. The two algorithms outperform other algorithms. We further observe that the allocations achieved under FWS and T-D are closer to the optimal allocation $\omega^\star(\boldsymbol{\mu})$ than those of other algorithms.

# E  Linear Bandits

## E.1  Preliminaries and competing algorithms

Linear bandits have been extensively applied in online advertisement, [34, 9], and have become the most relevant of bandit problems with structure. BAI in linear bandits has been investigated for example in [25, 46, 13].

Consider a bandit problem with $K$ arms and Gaussian rewards. Each arm $k$ is associated with a $d$-dimensional vector $\boldsymbol{a}_k$. Without loss of generality, we assume that $\{\boldsymbol{a}_k\}_{k\in[K]}$ spans the space $\mathbb{R}^d$. We study two learning tasks: BAI and the so-called threshold bandit task where the objective is to identify the set of arms whose expected rewards are above a given threshold.

We slightly modify our framework as described in Section 5. We define for the two tasks:

$$\Lambda_{\text{BAI}} = \left\{ \boldsymbol{\mu} \in \mathbb{R}^d : \exists k \in [K] \text{ s.t.} \langle \boldsymbol{a}_k - \boldsymbol{a}_i, \boldsymbol{\mu} \rangle > 0, \forall i \neq k \right\}, \qquad i^\star_{\text{BAI}}(\boldsymbol{\mu}) = \operatorname*{argmax}_k \langle \boldsymbol{a}_k, \boldsymbol{\mu} \rangle;$$

$$\Lambda_{\mathfrak{I}} = \left\{ \boldsymbol{\mu} \in \mathbb{R}^d : \langle \boldsymbol{a}_k, \boldsymbol{\mu} \rangle \neq \mathfrak{I}, \forall k \in [K] \right\}, \qquad i^\star_{\mathfrak{I}}(\boldsymbol{\mu}) = \left\{ k \in [K] : \langle \boldsymbol{a}_k, \boldsymbol{\mu} \rangle > \mathfrak{I} \right\}.$$

For BAI, $\mathcal{S}_i = \{ \boldsymbol{\mu} \in \mathbb{R}^d : \langle \boldsymbol{a}_k - \boldsymbol{a}_i, \boldsymbol{\mu} \rangle > 0, \forall i \neq k \}$ is clearly an open set. For the threshold problem, $\mathcal{S}_A = \{ \boldsymbol{\mu} \in \mathbb{R}^d : \langle \boldsymbol{a}_k, \boldsymbol{\mu} \rangle > \mathfrak{I}, \forall k \in A \}$, where $A$ is a subset of $[K]$, is an open set too. To implement our algorithm, we introduce $V_{\boldsymbol{\omega}} = \sum_k \omega_k \boldsymbol{a}_k \boldsymbol{a}_k^\top$, and build the Least-Squares Estimator of $\boldsymbol{\mu}$:

$$\hat{\boldsymbol{\mu}}(t) = V_{\boldsymbol{\omega}(t)}^\dagger \sum_{s=1}^t X_{A_t}(t) \boldsymbol{a}_{A_t}.$$

For this LSE, we use in the analysis the concentration results derived in Lemmas 3 and 4 in [25]. The objective function that has to be maximized to get an optimal allocation is:

$$F_{\boldsymbol{\mu}}(\boldsymbol{\omega}) = \inf_{\boldsymbol{\lambda} \in \text{Alt}(\boldsymbol{\mu})} \frac{\|\boldsymbol{\mu} - \boldsymbol{\lambda}\|_{V_{\boldsymbol{\omega}}}^2}{2},$$

where $\text{Alt}(\boldsymbol{\mu})$ depends on the task. Let us describe the most confusing parameter $\overline{\boldsymbol{\lambda}_j(\boldsymbol{\omega}, \boldsymbol{\mu})}$ for both tasks.

**BAI.**  Let $\boldsymbol{\mu} \in \Lambda_{\text{BAI}}$ and $j \neq i^\star_{\text{BAI}}(\boldsymbol{\mu})$, let $\mathcal{C}_j^{i^\star_{\text{BAI}}(\boldsymbol{\mu})} = \{ \boldsymbol{\lambda} \in \Lambda^{\text{BAI}} : \langle \boldsymbol{\lambda}, \boldsymbol{a}_j \rangle > \langle \boldsymbol{\lambda}, \boldsymbol{a}_{i^\star_{\text{BAI}}(\boldsymbol{\mu})} \rangle \}$, which is a convex set (as any convex combination of two points in $\mathcal{C}_j^{i^\star_{\text{BAI}}(\boldsymbol{\mu})}$ is still in $\mathcal{C}_j^{i^\star_{\text{BAI}}(\boldsymbol{\mu})}$). Applying the Lagrange multiplier theorem (see Appendix of [13]), we get that:

$$\overline{\boldsymbol{\lambda}_j^{\text{BAI}}(\boldsymbol{\omega}, \boldsymbol{\mu})} = \boldsymbol{\mu} + \left( \frac{\langle \boldsymbol{a}_{i^\star_{\text{BAI}}(\boldsymbol{\mu})} - \boldsymbol{a}_j, \boldsymbol{\mu} \rangle}{\left\| \boldsymbol{a}_{i^\star_{\text{BAI}}(\boldsymbol{\mu})} - \boldsymbol{a}_j \right\|_{V_{\boldsymbol{\omega}}^{-1}}^2} V_{\boldsymbol{\omega}}^{-1} \right) \left( \boldsymbol{a}_j - \boldsymbol{a}_{i^\star_{\text{BAI}}(\boldsymbol{\mu})} \right), \ \forall \boldsymbol{\omega} \in \mathring{\Sigma}. \tag{27}$$

**Threshold bandit.**  For all $j \in [K]$, we let $\mathcal{C}_j^{i^\star_{\mathfrak{I}}(\boldsymbol{\mu})} = \{ \boldsymbol{\lambda} \in \Lambda_{\mathfrak{I}} : \text{sign}(\mathfrak{I} - \langle \boldsymbol{a}_j, \boldsymbol{\lambda} \rangle) \neq \text{sign}(\mathfrak{I} - \langle \boldsymbol{a}_j, \boldsymbol{\mu} \rangle) \}$, which is a convex set (as again any convex combination of two points in $\mathcal{C}_j^{i^\star_{\mathfrak{I}}(\boldsymbol{\mu})}$ is still in $\mathcal{C}_j^{i^\star_{\mathfrak{I}}(\boldsymbol{\mu})}$). Likewise, the Lagrange multiplier theorem yields that

$$\overline{\boldsymbol{\lambda}_j^{\mathfrak{I}}(\boldsymbol{\omega}, \boldsymbol{\mu})} = \boldsymbol{\mu} + \text{sign}(\mathfrak{I} - \langle \boldsymbol{a}_j, \boldsymbol{\mu} \rangle) \left( \frac{(\mathfrak{I} - \langle \boldsymbol{a}_j, \boldsymbol{\mu} \rangle)}{\|\boldsymbol{a}_j\|_{V_{\boldsymbol{\omega}}^{-1}}^2} V_{\boldsymbol{\omega}}^{-1} \right) \boldsymbol{a}_j, \ \forall \boldsymbol{\omega} \in \mathring{\Sigma}. \tag{28}$$

$\nabla_{\boldsymbol{\omega}} f_j(\boldsymbol{\omega}, \boldsymbol{\mu})$ can be obtained by directly plugging (27) or (28) into (4) and Proposition 1 shows that $f_j(\boldsymbol{\omega}, \boldsymbol{\mu}) = \langle \boldsymbol{\omega}, \nabla f_j(\boldsymbol{\omega}, \boldsymbol{\mu}) \rangle$. We have checked Assumption 2 in Appendix C.3, using Lemma 1.

**Competing algorithms.** For BAI in linear structure, we compare the performance of the following algorithms.

- FWS: Our algorithm with parameters $r_t = t^{-0.9}/K$.

- `LT`: Lazy Track and Stop (`LT`) by [25].
- `CG-C` and `Lk-C`: LineGame-C (`CG-C`) and LineGame (`Lk-C`) from [13] implemented by [45].
- `XY-A`: XY-Adaptive [46]. The hyperparameter $\alpha$ is set equal to $0.1$ as done by [46].
- `RR`: Round Robin
- `LB`$_{\text{lin}}(\delta)$: $T^\star(\boldsymbol{\mu})\text{kl}(\delta, 1-\delta)$ exploiting the linear structure

To our best knowledge, the linear threshold bandit problem was only studied in [13]. Hence for this problem, we only compare our algorithm with `CG-C`, `Lk-C` and `RR`.

Stopping rules. For BAI in linear bandits, except for `XY-A`, all algorithms use the same stopping rule (7), with the same threshold $\beta(t,\delta) = \log((\log(t)+1)/\delta)$ (Note that the implementations in [45] make this choice as well).
For linear threshold bandits, we also use the stopping rule (7) with the same threshold $\beta(t,\delta) = \log((\log(t)+1)/\delta)$ for all algorithms.

## E.2 Numerical experiments

**BAI in linear bandits.** We consider the example proposed by [46]. The unknown parameter is $\boldsymbol{\mu} = \boldsymbol{e}_1$ and there are $d+1$ arms, $\boldsymbol{e}_1, \cdots, \boldsymbol{e}_d, \cos(\phi)\boldsymbol{e}_1 + \sin(\phi)\boldsymbol{e}_2$ in $\mathbb{R}^d$, where $\boldsymbol{e}_1, \cdots, \boldsymbol{e}_d$ form the standard orthonormal basis. We set $d = 6$ and $\phi = 0.1$.

In Table 6, we provide the sample complexity of the various algorithms averaged over 1000 runs for various confidence levels $\delta \in \{0.1, 0.01, 0.001, 0.0001\}$. To provide a more detailed comparison, at the confidence level $\delta = 0.01$, we show the sample complexity in box-plot in Figure 6a and compare the allocation of arm draws achieved under the various algorithms in Table 7.

Table 6: Sample complexity for BAI in linear bandits for the benchmark example of [46], averaged over 1000 runs.

|  | FWS | LT | CG-C | Lk-C | XY-A | RR | LB$_{\text{lin}}(\delta)$ |
|---|---|---|---|---|---|---|---|
| $\delta = 0.1$ | 1 030 | 919 | 2 498 | 2 319 | 7 016 | 5 451 | 359 |
| $\delta = 0.01$ | 1 614 | 1 464 | 3 501 | 3 431 | 7 779 | 8 814 | 920 |
| $\delta = 0.001$ | 2 229 | 1 982 | 4 324 | 4 326 | 9 090 | 12 101 | 1 408 |
| $\delta = 0.0001$ | 2 839 | 2 518 | 5 118 | 5 120 | 9 723 | 15 314 | 1 881 |

Table 7: Allocation of arm draws for the benchmark example of [46] at $\delta = 0.01$, averaged over 1000 runs.

|  | FWS | LT | CG-C | Lk-C | XY-A | RR | $\omega^\star(\boldsymbol{\mu})$ |
|---|---|---|---|---|---|---|---|
| $\boldsymbol{a}_1$ | 1.02 | 4.4 | 13.64 | 13.10 | 9.35 | 14.29 | 0.38 |
| $\boldsymbol{a}_2$ | 94.22 | 91.11 | 35.75 | 36.66 | 69.80 | 14.28 | 97.72 |
| $\boldsymbol{a}_3$ | 0.93 | 1.02 | 12.46 | 12.25 | 4.35 | 14.28 | 0.38 |
| $\boldsymbol{a}_4$ | 0.93 | 1.01 | 12.43 | 12.22 | 3.63 | 14.28 | 0.38 |
| $\boldsymbol{a}_5$ | 0.93 | 1 | 12.38 | 12.25 | 4.45 | 14.28 | 0.38 |
| $\boldsymbol{a}_6$ | 0.93 | 1 | 12.44 | 12.23 | 2.64 | 14.28 | 0.38 |
| $\boldsymbol{a}_7$ | 1.01 | 0.46 | 0.89 | 1.29 | 5.78 | 14.28 | 0.38 |

All the results above suggest that `FWS` is really competitive with the state-of-art algorithm, `LT`, and it achieves an allocation closer to the optimal allocation than its competitors. In Figure 4, we plot the number of rounds (the median over all runs) `FWS` is in force exploration or the $r$-subdifferential subspace used in `FWS` contains the gradient of only one function (in this round, our FW update coincides with the traditional FW update as if the objective function was smooth). In Figure 5, we provide the distribution of the number of functions involved in the FW updates. Most of the time, the update used in `FWS` coincides with the usual FW update (which contrasts with the case of BAI in unstructured bandits).

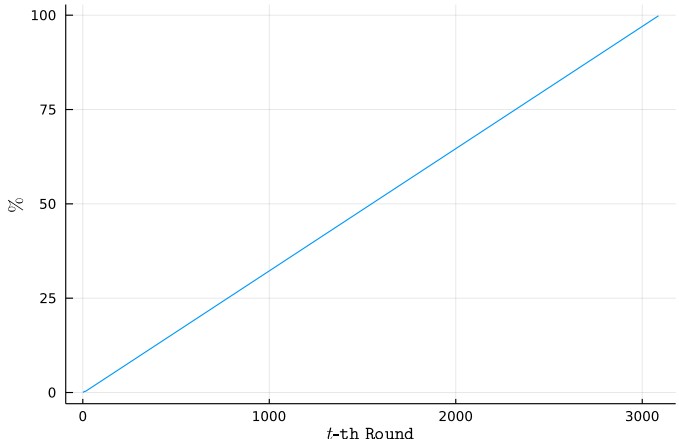

Figure 4: Number of rounds where the FW update in `FWS` corresponds to the usual FW update, and the force exploration. BAI in linear bandits for the benchmark example of [46] with $\delta = 0.0001$.

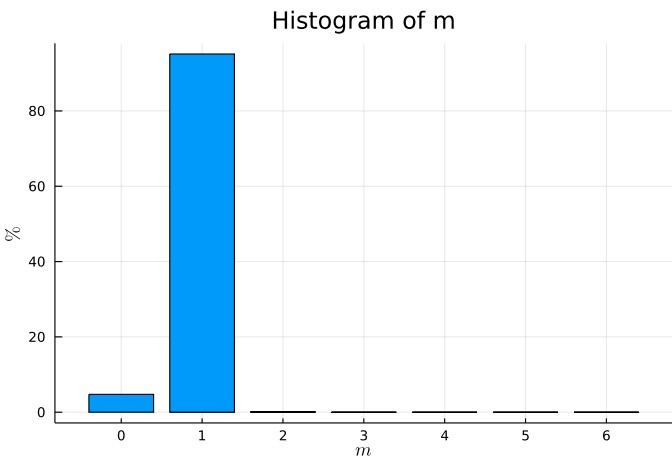

Figure 5: Proportions of rounds where we have $m$ functions in the linear program involved in the FW updates (i.e. $m = \left|\{j : f_j(\boldsymbol{x}(t), \hat{\boldsymbol{\mu}}(t)) < F_{\hat{\boldsymbol{\mu}}(t-1)}(\boldsymbol{x}(t)) + r_t\}\right|$, and $m = 0$ in forced exploration). BAI in linear bandits for the benchmark example of [46] with $\delta = 0.0001$.

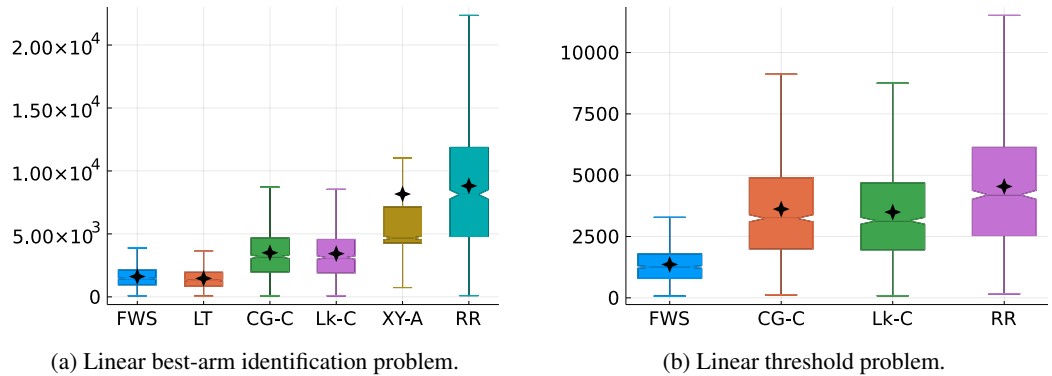

(a) Linear best-arm identification problem.

(b) Linear threshold problem.

Figure 6: Sample complexity for linear bandits at $\delta = 0.01$, plotted in boxplots where the stars represent the averaged sample complexity and the outliers are hidden.

**Threshold bandit.** Consider the linear threshold bandit problem, obtained by modifying the example of [46]: $\boldsymbol{\mu} = \boldsymbol{e}_1$ and there are $d + 1$ actions associated with $\boldsymbol{e}_1, \cdots, \boldsymbol{e}_d, \cos(\phi)\boldsymbol{e}_1 + \sin(\phi)\boldsymbol{e}_2$ in $\mathbb{R}^d$, where $(\boldsymbol{e}_1, \cdots, \boldsymbol{e}_d)$ form the standard orthonormal basis.

Table 8: Sample complexity for the linear threshold bandit problem, averaged over 1000 runs.

|  | FWS | CG-C | Lk-C | RR | $\text{LB}_{\text{lin}}(\delta)$ |
|---|---|---|---|---|---|
| $\delta = 0.1$ | 874 | 2 673 | 2 511 | 2 904 | 374 |
| $\delta = 0.01$ | 1 362 | 3 618 | 3 494 | 4 540 | 957 |
| $\delta = 0.001$ | 1 865 | 4 437 | 4 372 | 6 213 | 1465 |
| $\delta = 0.0001$ | 2 398 | 5 163 | 5 132 | 7 807 | 1957 |

Table 9: Allocation of arm draws for the linear threshold bandit problem at $\delta = 0.01$, averaged over 1000 runs.

|  | FWS | CG-C | Lk-C | RR | $\omega^{\star}(\boldsymbol{\mu})$ |
|---|---|---|---|---|---|
| $\boldsymbol{a}_1$ | 19.62 | 19.08 | 19.19 | 14.30 | 0.38 |
| $\boldsymbol{a}_2$ | 1.26 | 12.94 | 12.72 | 14.28 | 1.13 |
| $\boldsymbol{a}_3$ | 1.34 | 12.75 | 12.45 | 14.28 | 1.17 |
| $\boldsymbol{a}_4$ | 1.35 | 12.71 | 12.43 | 14.28 | 1.17 |
| $\boldsymbol{a}_5$ | 1.30 | 12.73 | 12.45 | 14.28 | 1.17 |
| $\boldsymbol{a}_6$ | 1.28 | 12.73 | 12.46 | 14.28 | 1.17 |
| $\boldsymbol{a}_7$ | 73.84 | 17.07 | 18.31 | 14.28 | 93.80 |

We set $d = 6$, $\phi = 0.01$, and the threshold $\mathfrak{I} = 0.9$. The goal is to identify all arms whose mean is larger than the $\mathfrak{I}$.

In Table 8, we provide the sample complexity of the various algorithms for $\delta \in \{0.1, 0.01, 0.001, 0.0001\}$ and averaged over 1000 runs. To provide a more detailed comparison, at the confidence level $\delta = 0.01$, we show the sample complexity in box-plot in Figure 6b and compare the allocation of arm draws achieved under the various algorithms in Table 9.

We have the similar observations as those made for the BAI task. FWS clearly outperforms its competitors.

# F  BAI in Lipschitz Bandits

## F.1  Preliminaries and competing algorithms

We consider the BAI task in the following Lipschitz bandit. There is a finite number $K$ of arms. Each arm $k$ is associated with a feature vector or *position* $\boldsymbol{a}_k \in \mathbb{R}^d$ for some $d \in \mathbb{N}$. The reward distributions are assumed to be Gaussian with a fixed and known variance. The mapping from the arm feature vector to the corresponding average reward is known to be Lipschitz, which means that:

$$\Lambda = \left\{ \boldsymbol{\mu} \in \mathbb{R}^K : \exists i \in [K] \text{ s.t. } \mu_i > \mu_k, \forall k \neq i \text{ and } |\mu_k - \mu_{k'}| < \ell \left\| \boldsymbol{a}_k - \boldsymbol{a}_{k'} \right\|_\infty, \ \forall k, k' \in [K] \right\},$$

where the constant $\ell > 0$ is known in advance. The answer map is $i^\star(\boldsymbol{\mu}) = \arg\max_i \mu_i$, since we consider a BAI task. Let us fix some $i \in [K]$ and $\boldsymbol{\mu} \in \mathcal{S}_i$, $\mathrm{Alt}(\boldsymbol{\mu})$ can be divided into a union of sets $\cup_{j \neq i} \{ \boldsymbol{\lambda} \in \Lambda : \lambda_j > \lambda_i \}$. Hence $\mathcal{J}_i = [K] \setminus \{i\}$. Assumption 1 holds as the set

$$\mathcal{S}_i = \left\{ \boldsymbol{\mu} \in \mathbb{R}^K : \mu_i > \mu_k, \forall k \neq i \text{ and } |\mu_k - \mu_{k'}| < \ell \left\| \boldsymbol{a}_k - \boldsymbol{a}_{k'} \right\|_\infty, \ \forall k, k' \in [K] \right\}$$

is open set for all $i \in [K]$ and

$$\mathcal{C}_j^i = \left\{ \boldsymbol{\lambda} \in \Lambda : \lambda_j > \lambda_i, \text{ and } |\lambda_k - \lambda_{k'}| < \ell \left\| \boldsymbol{a}_k - \boldsymbol{a}_{k'} \right\|_\infty, \ \forall k, k' \in [K] \right\}$$

is a convex set, for all $j \neq i$ (one can readily check that any convex combination of any two points in $\mathcal{C}_j^i$ is still in $\mathcal{C}_j^i$).

**Most confusing parameter.** Unlike in the previous examples, there is no close form for the $\overline{\boldsymbol{\lambda}_j(\boldsymbol{\omega}, \boldsymbol{\mu})}$. However, there is a simple strategy to compute it efficiently. Fix $\boldsymbol{\mu} \in \Lambda$, a suboptimal arm $j \neq i^\star(\boldsymbol{\mu})$, and $\boldsymbol{\omega} \in \mathring{\Sigma}$. The most confusing parameter solves (2), which in the case of Gaussian rewards translates to:

$$\min_{\boldsymbol{\lambda} \in \mathcal{C}_j^{i^\star(\boldsymbol{\mu})}} \sum_{k=1}^K \frac{\omega_k (\lambda_k - \mu_k)^2}{2}. \tag{29}$$

Observe that for the solution of (29), we should have $\lambda_j = \lambda_{i^\star(\boldsymbol{\mu})}$. Hence, the problem (29) can be simplified by setting $\lambda_j = \lambda_{i^\star(\boldsymbol{\mu})} = \theta \in [\mu_j, \mu_{i^\star(\boldsymbol{\mu})}]$ and by remarking that other values are decided by exploiting Lipschitz structure and minimizing the distance to $\boldsymbol{\mu}$. After simplification, we get a single-parameter (here $\theta$) optimization problem:

$$\min_{\theta \in [\mu_j, \mu_{i^\star(\boldsymbol{\mu})}]} \sum_k \frac{\omega_k}{2} \left\{ [(\theta - \ell \left\| \boldsymbol{a}_k - \boldsymbol{a}_j \right\|_\infty - \mu_k)^+]^2 + [(\mu_k - \theta - \ell \left\| \boldsymbol{a}_k - \boldsymbol{a}_{i^\star(\boldsymbol{\mu})} \right\|_\infty)^+]^2 \right\}. \tag{30}$$

Figure 7 provides a simple example to explain this transformation.

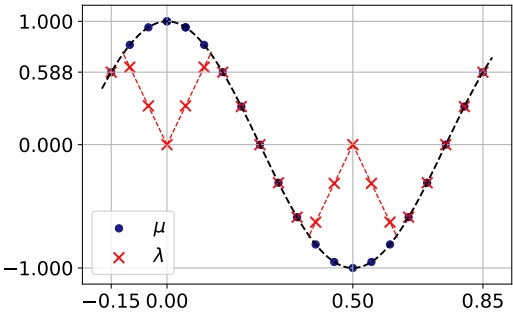

Figure 7: An example of most confusing parameters $\boldsymbol{\lambda} \in \mathcal{C}_j^{i^\star(\boldsymbol{\mu})}$. Along the $x$-axis, we have arm positions, and on the $y$-axis, the average rewards. Dots represent $\boldsymbol{\mu}$ and crosses $\boldsymbol{\lambda}$. Note that $\lambda_{i^\star(\boldsymbol{\mu})} = \lambda_j = \theta$ and that other components of $\boldsymbol{\lambda}$ are selected to get a minimal modification of $\boldsymbol{\mu}$ to satisfy the Lipschitz constraint. $\mu_k = \cos(2\pi(-0.15 + 0.05k)), \forall k = 0, 1, \ldots, 20$, $\ell = 2\pi$ and $\theta = 0$.

The minimal value of the above problem (30) is exactly $f_j(\boldsymbol{\omega}, \boldsymbol{\mu})$ and for any $k$, the $k$-th component of $\overline{\boldsymbol{\lambda}_j(\boldsymbol{\omega}, \boldsymbol{\mu})}$ is

$$\min\{\max\{\theta_j^\star - \ell \left\| \boldsymbol{a}_k - \boldsymbol{a}_j \right\|_\infty, \mu_k\}, \theta_j^\star + \ell \left\| \boldsymbol{a}_k - \boldsymbol{a}_{i^\star(\boldsymbol{\mu})} \right\|_\infty\}, \tag{31}$$

where $\theta_j^\star$ is the solution of the problem (30). $\theta_j^\star$ can be found by using simple binary search. $\nabla_{\boldsymbol{\omega}} f_j(\boldsymbol{\omega}, \boldsymbol{\mu})$ can be obtained by directly plugging $\overline{\boldsymbol{\lambda}_j(\boldsymbol{\omega}, \boldsymbol{\mu})}$ into the equation (4) and Proposition 1 shows that $f_j(\boldsymbol{\omega}, \boldsymbol{\mu}) = \langle \boldsymbol{\omega}, \nabla f_j(\boldsymbol{\omega}, \boldsymbol{\mu}) \rangle$. To check Assumption 2, we can use Lemma 1 as shown in Appendix C.

**Implementing** FWS. When the Lipschitz constant $\ell$ is tight, (i.e. $\max_{k \neq k'} \frac{|\mu_k - \mu_{k'}|}{\|\boldsymbol{a}_k - \boldsymbol{a}_{k'}\|_\infty} \approx \ell$), we need accurate estimate of $\boldsymbol{\mu}$ so that $\hat{\boldsymbol{\mu}}(t)$ satisfies the Lipschitz assumption, and belongs to $\Lambda$. In this case, the Lipschitz structure is too strong and in its initial design, FWS may use numerous rounds of forced exploration so that finally $\hat{\boldsymbol{\mu}}(t) \in \Lambda$. To circumvent this issue, we could project $\hat{\boldsymbol{\mu}}(t)$ to $\Lambda$. We use another solution that consists in artificially enlarging $\Lambda$. To this aim, we pick a Lipschitz constant $\ell'$ larger than $\ell$, and define

$$\Lambda_{\ell'} = \left\{ \boldsymbol{\mu} \in \mathbb{R}^K : \exists i \in [K] \text{ s.t. } \mu_i > \mu_k, \forall k \neq i \text{ and } |\mu_k - \mu_{k'}| < \ell' \|\boldsymbol{a}_k - \boldsymbol{a}_{k'}\|_\infty, \ \forall k, k' \in [K] \right\}.$$

Note that $\Lambda \subset \Lambda_{\ell'}$. Now when $\hat{\boldsymbol{\mu}}(t) \notin \Lambda$ (although there exists an unique empirical best arm under $\hat{\boldsymbol{\mu}}(t)$), we can find $\ell' > \ell$ s.t. $\hat{\boldsymbol{\mu}}(t) \in \Lambda_{\ell'}$. Inspired by this observation, we just replace in FWS the condition $\hat{\boldsymbol{\mu}}(t) \notin \Lambda$ by $\hat{\boldsymbol{\mu}}(t) \notin \Lambda_{\ell_{\text{pseudo}}(t)}$, where

$$\ell_{\text{pseudo}}(t) = \max \left\{ \ell, \max_{k \neq k'} \frac{|\hat{\mu}_k(t) - \hat{\mu}_{k'}(t)|}{\|\boldsymbol{a}_k - \boldsymbol{a}_{k'}\|_\infty} \right\}.$$

Note that $\ell_{\text{pseudo}}(t) \geq \ell$ and $\hat{\boldsymbol{\mu}}(t) \in \Lambda$ if and only if $\ell_{\text{pseudo}}(t) = \ell$. After this change, FWS does not have a forced exploration round each time $\hat{\boldsymbol{\mu}}(t) \notin \Lambda$. Removing this forced exploration condition does not affect our analysis.

**Competing algorithms.** As far as we know, this paper is the first to consider BAI in Lipschitz bandits. Hence, for this task, we just investigate the following algorithms and baselines:

- FWS: Our algorithm with parameters $r = t^{-0.9}/K$, where $K$ is the number of arms.
- T-D: Track and Stop [28] with D-Tracking. T-D is the strongest baseline without prior knowledge of the structure, and we include it to estimate the gains achieved when exploiting the Lipschitz structure.
- M-C: Here we use $\ell_{\text{pseudo}}(t)$, introduced above, and Proposition 1 to construct the subdifferential for LMA [38]. The learning rate $\eta_t$ is chosen with the knowledge of $\boldsymbol{\mu}$ (as discussed previously). Note that there is not known theoretical guarantees for LMA in Lipschitz bandits.
- LB$_{\text{Lip}}(\delta)$: $T^\star(\boldsymbol{\mu})\text{kl}(\delta, 1 - \delta)$ with Lipschitz structure.

Stopping rules. FWS and M-C use the stopping rule (7) for Lipschitz bandits while T-D uses the same stopping rule for the unstructured bandits. The threshold is set equal to $\beta(t, \delta) = \log((\log(t) + 1)/\delta)$ for both algorithms.

### F.2 Numerical experiments

We consider two experiments.

**Experiment L1.** In this experiment, the average rewards are given by the $\ell$-Lipschitz function $f(x) = 9\cos(x)/(x^2 + 10)$, where $\ell = 0.9$. We have 20 arms with mean $\boldsymbol{\mu} = [f(x_1), \cdots, f(x_{20})]$, where $x_i = 1.25 + 0.25(i - 1), \forall i = 1, \cdots, 20$, as shown in Figure 8.

In Table 10, we provide the sample complexity of the various algorithms averaged over 100 runs for confidence levels $\delta \in \{0.1, 0.01\}$. To provide a more detailed comparison, at the confidence level $\delta = 0.01$, we show the sample complexity in box-plot in Figure 11a and compare the allocation of arm draws achieved under the various algorithms in Table 11. Observe that FWS outperforms M-C, and manages to almost halve the sample complexity compared to T-D. Exploiting the structure yields critical improvements.

In Figure 9, we plot the number of rounds (the median over all runs) FWS is in force exploration or the $r$-subdifferential subspace used in FWS contains the gradient of only one function (in this round, our FW update coincides with the traditional FW update as if the objective function was smooth). In Figure 10, we provide the allocation that number of functions are involved in FW update.

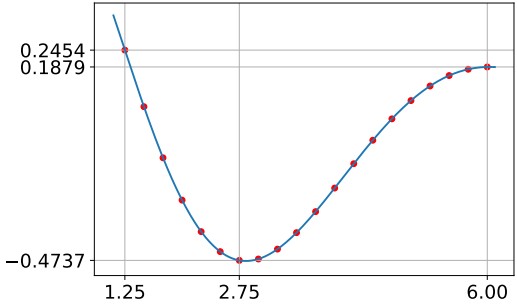

Figure 8: Experiment L1. The positions of the arms ($x$-axis) and their expected rewards ($y$-axis).

Table 10: Sample complexity for Experiment L1 averaged over 100 runs.

|  | FWS | M-C | T-D | $\text{LB}_{\text{Lip}}(\delta)$ |
|---|---|---|---|---|
| $\delta = 0.1$ | 21 791 | 30 999 | 41 182 | 6 798 |
| $\delta = 0.01$ | 30 051 | 41 481 | 56 810 | 17 415 |

Table 11: The average rewards and the allocation of arm draws (%) in Experiment L1 with $\delta = 0.01$ averaged over 100 runs.

|  |  | $a_1$ | $a_2$ | $a_3$ | $a_4$ | $a_5$ | $a_6$ | $a_7$ | $a_8$ | $a_9$ | $a_{10}$ |
|---|---|---|---|---|---|---|---|---|---|---|---|
|  | Positions | 1.25 | 1.5 | 1.75 | 2.0 | 2.25 | 2.5 | 2.75 | 3.0 | 3.25 | 3.5 |
|  | $\mu$ | 0.25 | 0.05 | -0.12 | -0.27 | -0.38 | -0.44 | -0.47 | -0.47 | -0.44 | -0.38 |
| Allocation (%) | FWS | 29.15 | 1.14 | 0.34 | 0.39 | 0.15 | 0.15 | 0.14 | 0.14 | 0.14 | 0.15 |
|  | M-C | 24.56 | 2.88 | 2.79 | 1.93 | 1.67 | 1.59 | 1.57 | 1.57 | 1.58 | 1.65 |
|  | T-D | 35.08 | 0.85 | 0.39 | 0.38 | 0.38 | 0.38 | 0.38 | 0.38 | 0.38 | 0.38 |
|  |  | $a_{11}$ | $a_{12}$ | $a_{13}$ | $a_{14}$ | $a_{15}$ | $a_{16}$ | $a_{17}$ | $a_{18}$ | $a_{19}$ | $a_{20}$ |
|  | Positions | 3.75 | 4.0 | 4.25 | 4.5 | 4.75 | 5.0 | 5.25 | 5.5 | 5.75 | 6.0 |
|  | $\mu$ | -0.31 | -0.23 | -0.14 | -0.06 | 0.01 | 0.07 | 0.12 | 0.16 | 0.18 | 0.19 |
| Allocation (%) | FWS | 0.17 | 0.21 | 0.29 | 0.43 | 0.79 | 1.56 | 3.04 | 7.25 | 23.00 | 31.57 |
|  | M-C | 1.92 | 2.17 | 2.68 | 2.90 | 3.09 | 3.41 | 4.35 | 6.90 | 13.00 | 17.78 |
|  | T-D | 0.38 | 0.38 | 0.38 | 0.41 | 0.52 | 1.09 | 2.59 | 7.53 | 19.44 | 28.30 |

**Experiment L2.** In the second experiment, we consider the arms with positions and average rewards presented in the second and third rows of Table 12, respectively. The reward function is Lipschitz, and the learner is informed that this function has a Lipschitz constant $\ell = 0.01$. This example is chosen because identifying the best arm $a_1$ is hard without leveraging the Lipschitz structure. Indeed, to identify $a_1$, the learner will need to select $a_1$ and $a_6$ (the second best arm) often. Imagine now that the average rewards of these two arms are well known. If the learner is not aware of the Lipschitz structure, she will need to further explore all other arms. However, if she is aware that the reward function is $0.01$-Lipschitz, knowing that the average reward of $a_6$ is roughly 1, she will deduce that the average rewards of all other arms (except $a_1$) must be in the interval $[0.96, 1.04]$ ($a_2$ and $a_{10}$ are at a distance 4 from $a_6$). These arms are then worse than $a_1$, and an informed learner does not really need to explore them. In summary, we expect that exploiting the structure in L2 will bring significant improvement in the sample complexity.

In Table 13, we report the sample complexity of the various algorithms averaged over 100 runs for confidence levels $\delta \in \{0.1, 0.01\}$. To provide a more detailed comparison, at the confidence level $\delta = 0.01$, we show the sample complexity in box-plot in Figure 11b and compare the allocation of arm draws achieved under the various algorithms in Table 12. Observe that again, FWS outperforms M-C, and in this experiment, it manages to almost divide the sample complexity by factor 3 compared to T-D. As expected, exploiting the structure yields an even greater improvement than in Experiment L1.

Table 12: Average rewards and allocation of arm draws (%) at $\delta = 0.01$ averaged over 100 runs.

|  |  | $a_1$ | $a_2$ | $a_3$ | $a_4$ | $a_5$ | $a_6$ | $a_7$ | $a_8$ | $a_9$ | $a_{10}$ |
|---|---|---|---|---|---|---|---|---|---|---|---|
|  | Positions | 0 | 96 | 97 | 98 | 99 | 100 | 101 | 102 | 103 | 104 |
|  | $\mu$ | 1.06 | 0.99 | 0.99 | 0.99 | 0.99 | 1 | 0.99 | 0.99 | 0.99 | 0.99 |
| Allocations (%) | FWS | 26.63 | 7.30 | 8.45 | 6.47 | 7.69 | 12.39 | 7.26 | 7.93 | 7.95 | 7.92 |
|  | M-C | 27.85 | 7.76 | 8.05 | 8.16 | 7.99 | 8.81 | 8.21 | 7.86 | 8.09 | 7.22 |
|  | T-D | 25.37 | 7.84 | 10.06 | 8.10 | 5.50 | 15.20 | 7.06 | 9.16 | 5.59 | 6.11 |

Table 13: Sample complexity for Experiment L2 averaged over 100 runs.

|  | FWS | M-C | T-D | $\text{LB}_{\text{Lip}}(\delta)$ |
|---|---|---|---|---|
| $\delta = 0.1$ | 29 308 | 35 582 | 75 154 | 6 046 |
| $\delta = 0.01$ | 41 909 | 47 759 | 98 188 | 15 490 |

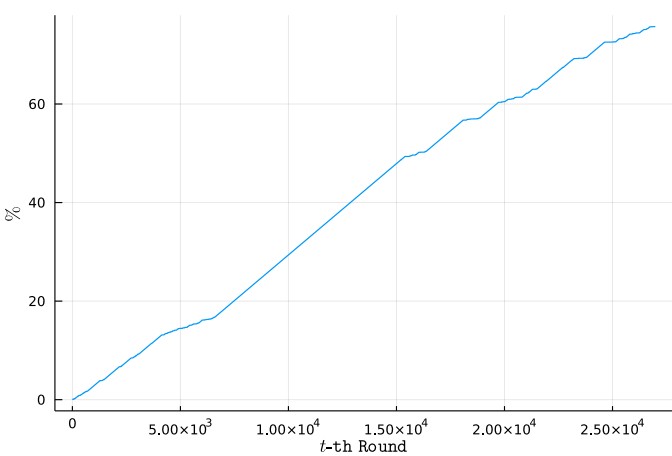

Figure 9: Number of rounds where FWS is in forced exploration or the FW update in FWS corresponds to the usual FW update. Experiment L1.

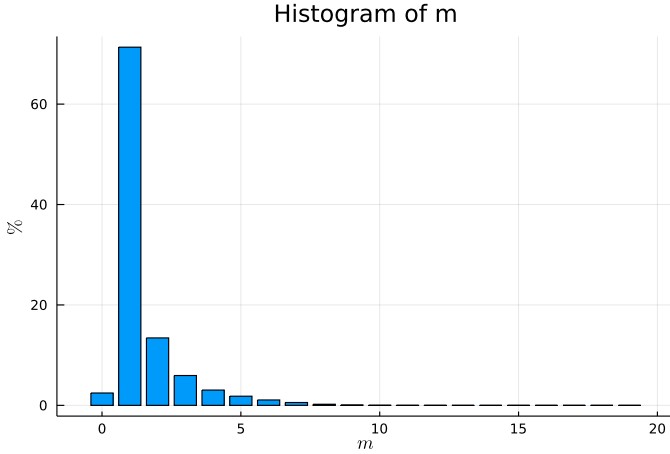

Figure 10: Proportions of rounds where we have $m$ functions in the linear program involved in the FW updates (i.e. $m = \left|\{j : f_j(\boldsymbol{x}(t), \hat{\boldsymbol{\mu}}(t)) < F_{\hat{\boldsymbol{\mu}}(t-1)}(\boldsymbol{x}(t)) + r_t\}\right|$, and $m = 0$ in forced exploration). Experiment L1.

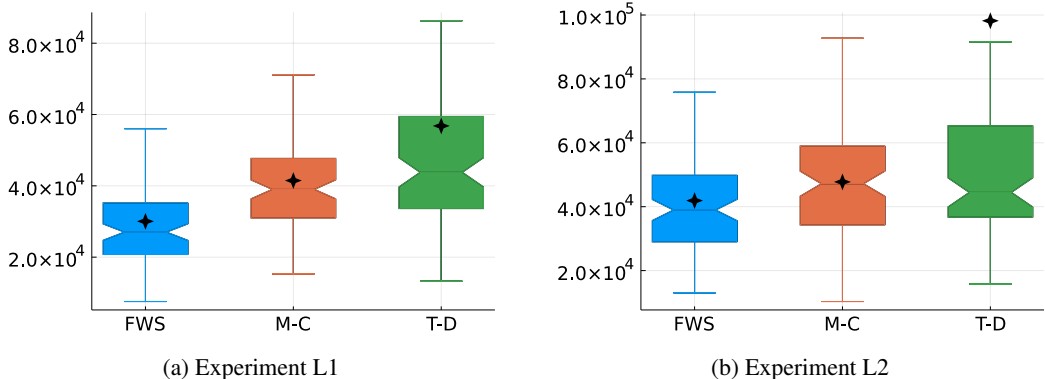

(a) Experiment L1                    (b) Experiment L2

Figure 11: Sample complexity averaged over 100 runs at $\delta = 0.01$. The stars in the boxplots represent the averaged sample complexity and the outliers are hidden.

# G  Additional Examples

In this section, we present two additional examples to illustrate the applicability of our framework. We do not report any numerical experiments on these.

## G.1  Threshold problem in monotone bandits

This task has applications in clinical trials. The learner aims at determining the maximum tolerable dose (MTD) (the maximum amount of the drug that can be given to a person without any potential danger). Arms represent the increasing doses, and the risk the potential adverse effects is drawn from a Gaussian distribution whose average increases with the dose. The learner is given a threshold of tolerance risk $\mathfrak{I} \in \mathbb{R}$, and wish to identify the first arm with risk that exceeds this threshold. Refer to [21] for details.

This pure exploration task can be investigated using our framework. We have: $\Lambda = \{\boldsymbol{\mu} \in \mathbb{R}^K : \mu_1 < \mu_2 < \ldots < \mu_K, \text{ and } \mu_k \neq \mathfrak{I}, \forall k \in [K]\}$. the set of possible answer is $\mathcal{I} = [K] \cup \emptyset$: if the true answer is $k$, arm $k$ is the last arm below the threshold, and $\emptyset$ refers to the case where all arms have a risk above the threshold. The set of parameters $\boldsymbol{\mu}$ for which $k$ is the correct answer is $\mathcal{S}_k = \{\boldsymbol{\mu} \in \Lambda : \mu_k < \mathfrak{I} < \mu_{k+1}\}$, which is an open set. Observe that it is also convex. Similarly $\mathcal{S}_\emptyset = \{\boldsymbol{\mu} \in \Lambda : \mu_1 > \mathfrak{I}\}$ is also open and convex.

Let us now identify the set of confusing parameters. To simplify the presentation, we assume that $\boldsymbol{\mu}$ is such that $1 < i^\star(\boldsymbol{\mu}) < K$. The set of confusing parameters can be decomposed as $\text{Alt}(\boldsymbol{\mu}) = \cup_{u \neq i^\star(\boldsymbol{\mu})} \mathcal{C}_u^{i^\star(\boldsymbol{\mu})}$, where $\mathcal{C}_u^{i^\star(\boldsymbol{\mu})} = \mathcal{S}_u$ is convex. Assumption 1 is hence verified. Let $u \neq i^\star(\boldsymbol{\mu})$ and $\boldsymbol{\omega} \in \overset{\circ}{\Sigma}$. Elementary calculus yields that

$$\overline{\boldsymbol{\lambda}_u(\boldsymbol{\omega}, \boldsymbol{\mu})} = \begin{cases} \boldsymbol{\mu} + \sum_{s=u}^{i^\star(\boldsymbol{\mu})} (\mathfrak{I} - \mu_s) \boldsymbol{e}_s & \text{if } u < i^\star(\boldsymbol{\mu}), \\ \boldsymbol{\mu} + \sum_{s=i^\star(\boldsymbol{\mu})}^{u} (\mathfrak{I} - \mu_s) \boldsymbol{e}_s, & \text{otherwise.} \end{cases}$$

This implies that

$$\nabla_{\boldsymbol{\omega}} f_u(\boldsymbol{\omega}, \boldsymbol{\mu}) = \begin{cases} \sum_{s=u}^{i^\star(\boldsymbol{\mu})} \frac{(\mathfrak{I} - \mu_s)^2}{2} \boldsymbol{e}_s & \text{if } u < i^\star(\boldsymbol{\mu}), \\ \sum_{s=i^\star(\boldsymbol{\mu})}^{u} \frac{(\mathfrak{I} - \mu_s)^2}{2} \boldsymbol{e}_s, & \text{otherwise.} \end{cases} \tag{32}$$

As shown in Proposition 1, $\langle \boldsymbol{\omega}, \nabla f_u(\boldsymbol{\omega}, \boldsymbol{\mu}) \rangle = f_u(\boldsymbol{\omega}, \boldsymbol{\mu})$, and thus

$$f_\ell(\boldsymbol{\omega}, \boldsymbol{\mu}) = \begin{cases} \sum_{s=u}^{i^\star(\boldsymbol{\mu})} \omega_s \frac{(\mathfrak{I} - \mu_s)^2}{2} & \text{if } u < i^\star(\boldsymbol{\mu}), \\ \sum_{s=i^\star(\boldsymbol{\mu})}^{u} \omega_s \frac{(\mathfrak{I} - \mu_s)^2}{2}, & \text{otherwise.} \end{cases} \tag{33}$$

In view of (32), Assumptions 2 (i) holds (as $\nabla_{\boldsymbol{\omega}} f_j$ is bounded). Assumption 2 (ii) can be easily verified by differentiating (32) with respect to $\boldsymbol{\omega}$ or using the sufficient condition provided in Appendix C.

## G.2  Top-m arms indentification in dueling bandits

The top-$m$ arms identification task consists in identifying the $m$ best arms. To solve this task in dueling bandits [53], the learner is allowed to sequentially pick pairs of arms. If the pair $(i, j)$ is selected, the learner observes the realization of a Bernoulli r.v. with mean $\mu_{i,j}$. If $\mu_{i,j} > 1/2$, we say that arm $i$ is better than arm $j$. The preference matrix $\boldsymbol{\mu} = (\mu_{i,j})$ is assumed to satisfy:

(a)  $\mu_{i,j} = 1 - \mu_{j,i}, \quad \forall (i, j) \in [K]^2$.

(b)  $\mu_{i,i} = \dfrac{1}{2}$.

(c)  if $\min(\mu_{i,j}, \mu_{j,k}) \geq \dfrac{1}{2}$, then $\mu_{i,k} \geq \dfrac{1}{2}$.

Under this assumption, $\boldsymbol{\mu}$ induces a total order $\succ_{\boldsymbol{\mu}}$, defined by $i \succ_{\boldsymbol{\mu}} j$ if and only if $\mu_{i,j} \geq 1/2$. Also note that under this assumption, $\boldsymbol{\mu}$ is defined only through its entries above the diagonal, hence by $\frac{K(K-1)}{2}$ parameters. We denote by $\sigma_{\boldsymbol{\mu}}$ a permutation of $[K]$, such that $\sigma_{\boldsymbol{\mu}}(1) \succ_{\boldsymbol{\mu}} \sigma_{\boldsymbol{\mu}}(2) \succ_{\boldsymbol{\mu}} \ldots \succ_{\boldsymbol{\mu}} \sigma_{\boldsymbol{\mu}}(m) \succ_{\boldsymbol{\mu}} \ldots \succ_{\boldsymbol{\mu}} \sigma_{\boldsymbol{\mu}}(K)$. We are ready to define

$$\Lambda = \left\{ \boldsymbol{\mu} \in (0,1)^{\frac{K(K-1)}{2}} : \boldsymbol{\mu} \text{ satisfies (c) and (d)} \right\},$$

where (d) ensures that the set of $m$ best arms is unique:

$$(d) \quad \text{we can select } \sigma_{\boldsymbol{\mu}} \text{ such that } \mu_{\sigma_{\boldsymbol{\mu}}(m),\sigma_{\boldsymbol{\mu}}(m+1)} > \frac{1}{2}.$$

In our framework, the set of answers is $\mathcal{I} = \{\mathcal{A} \subset [K] : |\mathcal{A}| = m\}$ and for any $\mathcal{A} \in \mathcal{I}$, $\mathcal{S}_{\mathcal{A}} = \{\boldsymbol{\mu} \in \Lambda : \mu_{i,j} > 1/2 \text{ if } i \in \mathcal{A} \text{ but } j \notin \mathcal{A}\}$. We can readily check that $\mathcal{S}_{\mathcal{A}}$ is open.

Now let $\boldsymbol{\mu} \in \Lambda$. Assume w.l.o.g. that $\sigma_{\boldsymbol{\mu}} = Id$ (identity permutation); in particular $1 \succ_{\boldsymbol{\mu}} 2 \succ_{\boldsymbol{\mu}} \ldots \succ_{\boldsymbol{\mu}} m \succ_{\boldsymbol{\mu}} \ldots \succ_{\boldsymbol{\mu}} K$. The true answer is $[m]$. If we define:

$$\mathcal{J}_{[m]} = \{\sigma \in \Theta : \exists k > m \text{ s.t } \sigma(k) \leq m\} \quad \text{and} \quad \forall \sigma \in \mathcal{J}_{[m]}, \mathcal{C}_{\sigma}^{[m]} = \{\boldsymbol{\lambda} \in \Lambda : \sigma_{\boldsymbol{\lambda}} = \sigma\},$$

where $\Theta$ is the set of all the permutations of $[K]$, then $\text{Alt}(\boldsymbol{\mu}) = \cup_{\sigma \in \mathcal{J}_{[m]}} \mathcal{C}_{\sigma}^{[m]}$ and $\mathcal{C}_{\sigma}^{[m]}$ is a convex set (for any $\boldsymbol{\lambda}, \tilde{\boldsymbol{\lambda}} \in \mathcal{C}_{\sigma}^{[m]}$, for any of their convex combinations $\boldsymbol{\lambda}'$, we have $\sigma_{\boldsymbol{\lambda}'} = \sigma$). Assumption 1 is hence verified. For each $\sigma \in \mathcal{J}_{[m]}$, we discuss the most confusing parameter in the set $\mathcal{C}_{\sigma}^{[m]}$ against $\boldsymbol{\mu}$ at the point $\boldsymbol{\omega}$. Namely, we solve

$$\min_{\boldsymbol{\lambda} \in \mathcal{C}_{\sigma}^{[m]}} \sum_{k < \ell} \omega_{k,\ell} d(\mu_{k,\ell}, \lambda_{k,\ell}),$$

where $\omega_{k,\ell}$ is the proportion of times that $(k,\ell)$ is pulled (in dueling bandits, pulling $(k,\ell)$ is equivalent to pulling $(\ell,k)$, hence we only count for $k < \ell$). For any $k < \ell$, we can readily show that

$$\overline{\boldsymbol{\lambda}_{\sigma}(\boldsymbol{\omega}, \boldsymbol{\mu})}_{k,\ell} = \begin{cases} \frac{1}{2} & \text{if } \sigma(\ell) < \sigma(k), \\ \mu_{k,\ell}, & \text{otherwise.} \end{cases}$$

This implies that

$$\nabla_{\boldsymbol{\omega}} f_{\sigma}(\boldsymbol{\omega}, \boldsymbol{\mu}) = \sum_{\substack{k < \ell \\ \sigma(\ell) < \sigma(k)}} d(\mu_{k,\ell}, \frac{1}{2}). \tag{34}$$

As shown in Proposition 1, $\langle \boldsymbol{\omega}, \nabla f_{\sigma}(\boldsymbol{\omega}, \boldsymbol{\mu}) \rangle = f_{\sigma}(\boldsymbol{\omega}, \boldsymbol{\mu})$, and thus

$$f_{\sigma}(\boldsymbol{\omega}, \boldsymbol{\mu}) = \sum_{\substack{k < \ell \\ \sigma(\ell) < \sigma(k)}} \omega_{k,\ell} d(\mu_{k,\ell}, \frac{1}{2}). \tag{35}$$

In view of (34), Assumptions 2 (i) holds (as $\nabla_{\boldsymbol{\omega}} f_{\sigma}$ is bounded). Assumption 2 (ii) can be easily verified by differentiating (34) with respect to $\boldsymbol{\omega}$ or using the sufficient condition provided in Appendix C.

# H   Zero-sum Game: the Equivalent Linear Program

In this section, we explain how to transform the zero-sum game (11) used in our FW update to a simple Linear Program (LP). The zero-sum game is:

$$\boldsymbol{z}(t) \leftarrow \operatorname*{argmax}_{\boldsymbol{z} \in \Sigma} \min_{h \in H_{F_{\hat{\boldsymbol{\mu}}(t-1)}}(\boldsymbol{x}(t-1), r_t)} \langle \boldsymbol{z} - \boldsymbol{x}(t-1), h \rangle$$

For clarity, we use the following notations: $\boldsymbol{x} = \boldsymbol{x}(t-1) \in \mathring{\Sigma}$, $\boldsymbol{\mu} = \hat{\boldsymbol{\mu}}(t-1)$, $r = r_t$, and we assume w.l.o.g. that $j = 1, \ldots, J$ are the indexes in $\mathcal{J}_{i^\star(\boldsymbol{\mu})}$ verifying $f_j(\boldsymbol{x}, \boldsymbol{\mu}) < F_{\boldsymbol{\mu}}(\boldsymbol{x}) + r$. Hence, $H_{F_{\boldsymbol{\mu}}}(\boldsymbol{x}, r) = \operatorname{cov}(\{\nabla_{\boldsymbol{\omega}} f_j(\boldsymbol{x}, \boldsymbol{\mu})\}_{j=1}^J)$.

Define the payoff matrix $M \in \mathbb{R}^{K \times J}$ with $M_{k,j} = \langle \boldsymbol{e}_k - \boldsymbol{x}, \nabla_{\boldsymbol{\omega}} f_j(\boldsymbol{x}, \boldsymbol{\mu}) \rangle$, for all $k \in [K], j \in [J]$. Then the problem (11) can be formulated as

$$\max_{\boldsymbol{z} \in \Sigma} \min_{\boldsymbol{y} \in \mathbb{R}^J} \boldsymbol{z}^\top M \boldsymbol{y} \tag{36}$$

$$\text{s.t. } y_j \geq 0, \forall j \in [J] \text{ and } y_1 + y_2 + \ldots + y_J = 1.$$

Denote by $(\boldsymbol{z}^\star, \boldsymbol{y}^\star)$ the solution of the problem (36). Then the solution $\boldsymbol{z}(t)$ of (11) is

$$\boldsymbol{z}(t) = \boldsymbol{x} + \sum_k z_k^\star (\boldsymbol{e}_k - \boldsymbol{x}) = \boldsymbol{z}^\star.$$

Standard textbooks in game theory present procedures to solve (36) by transforming it into an LP [37, 48, 52]. We give below the method we used in our experiments.

If $\boldsymbol{z} \in \mathbb{R}^K$ is fixed, the best response of the $\boldsymbol{y}$-player is a pure strategy. The pay-off of this strategy is of course $\min\{(\boldsymbol{z}^\top M)_1, \ldots, (\boldsymbol{z}^\top M)_J\}$. As a consequence, the optimal strategy for the $\boldsymbol{z}$-player is to solve the following problem:

$$\max_{\boldsymbol{z} \in \Sigma} \left\{ \min\{(\boldsymbol{z}^\top M)_1, \ldots, (\boldsymbol{z}^\top M)_J\} \right\}. \tag{37}$$

(37) is transformed to an LP by introducing an auxiliary parameter $u \in \mathbb{R}$ as a lower bound of $(\boldsymbol{z}^\top M)_j$. The problem (37) becomes

$$\max_{\boldsymbol{z} \in \Sigma, u \in \mathbb{R}} u \tag{38}$$

$$\text{s.t. } (\boldsymbol{z}^\top M)_j \geq u, \forall j = 1, \ldots, J.$$

# I  Asymptotic Sample Complexity Upper Bound

This section is devoted to the proof of Theorem 1. This theorem summarizes our analysis of FWS, and its proof heavily relies on results presented in subsequent appendices. Specifically in Appendix J, we state and prove concentration results quantifying how $\hat{\mu}(t)$ concentrates around $\mu$, and how the FW update in FWS differs from the same update obtained assuming that $\mu$ is known. In turn, to establish these results, we will need continuity arguments presented in Appendix K (e.g., our FW update needs to be continuous in $\omega$, $\mu$ and the parameter $r$). In Appendix L, we provide useful results related to the convergence of our variant of the FW algorithm. The proof of Theorem 1 will finally require us to study the tracking rule, which is done in Appendix M.

Coming back to the present appendix, we start with the almost sure upper bound and then proceed with the expected upper bound.

## I.1  Almost sure upper bound

The proof starts by defining the event

$$\mathcal{E} = \left\{ F_{\mu}(\omega(t)) \xrightarrow{t\to\infty} F_{\mu}(\omega^{\star}(\mu)) \text{ and } \hat{\mu}(t) \xrightarrow{t\to\infty} \mu \right\}.$$

We know that $\mathbb{P}_{\mu}[\mathcal{E}] = 1$ based on the Theorem 7 in Appendix L and on the law of the large number (every arm will be pulled infinite times because of forced exploration rounds). Since $F_{\mu}(\omega)$ is continuous w.r.t. $\mu$ (Lemma 6 in Appendix K), we also have that $F_{\hat{\mu}(t)}(\omega) \xrightarrow{t\to\infty} F_{\mu}(\omega)$ uniformly over $\omega \in \overset{\circ}{\Sigma}$ almost surely. This further implies that $F_{\hat{\mu}(t)}(\omega(t)) \xrightarrow{t\to\infty} F_{\mu}(\omega^{\star}(\mu))$ a.s. (by applying triangular inequality). Let $\epsilon \in (0,1)$. Under the event $\mathcal{E}$, there exists a constant $t_1$ such that for $t \geq t_1$, $F_{\hat{\mu}(t)}(\omega(t)) \geq (1-\epsilon)F_{\mu}(\omega^{\star}(\mu))$. Hence, denoting $\mathbb{N}^* = \mathbb{N} \cup \{\infty\}$, we get:

$$\begin{aligned}
\tau &= \inf \left\{ t \in \mathbb{N}^* : t F_{\hat{\mu}(t)}(\omega(t)) \geq \beta(t,\delta) \right\} \\
&\leq t_1 \vee \inf \left\{ t \in \mathbb{N}^* : t(1-\epsilon) F_{\mu}(\omega^{\star}(\mu)) \geq \beta(t,\delta) \right\} \\
&\leq t_1 \vee \inf \left\{ t \in \mathbb{N}^* : t \geq \frac{\beta(t,\delta) T^{\star}(\mu)}{(1-\epsilon)} \right\} \\
&\leq c_1(\Lambda) \vee t_1 \vee \inf \left\{ t \in \mathbb{N}^* : t \geq \frac{\log(c_2(\Lambda)t) T^{\star}(\mu)}{(1-\epsilon)\delta} \right\}.
\end{aligned}$$

where the second inequality stems from the fact that $T^{\star}(\mu)^{-1} = F_{\mu}(\omega^{\star}(\mu))$ and the final inequality stems from (7). Finally, applying Lemma 4 (presented at the end of this appendix) with $\alpha = 1$, $c_1 = \frac{(1-\epsilon)\delta}{T^{\star}(\mu)}$ and $c_2 = c_2(\Lambda)$ to the above inequality yields that

$$\tau \leq c_1(\Lambda) + t_1 + \frac{T^{\star}(\mu)}{(1-\epsilon)\delta} \left[ \log \left( \frac{T^{\star}(\mu)c_2(\Lambda)e}{\delta(1-\epsilon)} \right) + \log\log \left( \frac{T^{\star}(\mu)c_2(\Lambda)}{\delta(1-\epsilon)} \right) \right].$$

This implies $\mathbb{P}_{\mu}\left[ \limsup_{\delta\to 0} \frac{\tau}{\log(1/\delta)} \leq T^{\star}(\mu) \right] = 1$ and $\mathbb{P}_{\mu}[\tau < \infty] = 1$, for all $\delta \in (0,1)$. The fact that the algorithm is $\delta$-PAC directly follows from the property of $\beta(t,\delta)$, i.e. (8).

## I.2  Expected upper bound

Let $\epsilon \in (0,1)$. Based on the conditions imposed on $\{r_t\}$, there exist $T_{\epsilon,L} \in \mathbb{N}$ such that

$$\sum_{s=1}^{t} r_s < t\epsilon \text{ and } tr_t > L \text{ if } t \geq T_{\epsilon,L}. \tag{39}$$

Let $M = \max\left\{ \left(\frac{32D+3L}{\epsilon}\right)^{11}, T_{\epsilon,L}^{\frac{11}{8}}, (4K+1)^{\frac{11}{8}} \right\}$ and for any $T \geq M$, define the functions

$$\begin{cases}
\underline{h}(T) = \min \left\{ t \in \mathbb{N} : t \geq T^{\frac{8}{11}} + 2, \sqrt{t/K} \in \mathbb{N} \right\}, \\
\overline{h}(T) = \min \left\{ t \in \mathbb{N} : t \geq T^{\frac{2}{11}} \underline{h}(T), \sqrt{t/K} \in \mathbb{N} \right\}.
\end{cases} \tag{40}$$

We are now ready to introduce our "good" events $\mathcal{E}_{1,\epsilon}(T) = \left(\bigcap_{t=\underline{h}(T)}^{T} \mathcal{E}_{1,\epsilon}^{(t)}\right)$ and $\mathcal{E}_{2,\epsilon}(T) = \left(\bigcap_{t=\underline{h}(T)}^{T} \mathcal{E}_{2,\epsilon}^{(t)}\right)$, where

$$\mathcal{E}_{1,\epsilon}^{(t)} = \left\{ \max_{\boldsymbol{z} \in \Sigma} \min_{h \in H_{F_{\boldsymbol{\mu}}}(\boldsymbol{x}(t-1), r_t)} \langle \boldsymbol{z} - \boldsymbol{x}(t-1), h \rangle - \epsilon < \min_{h \in H_{F_{\boldsymbol{\mu}}}(\boldsymbol{x}(t-1), r_t)} \langle \boldsymbol{z}(t) - \boldsymbol{x}(t-1), h \rangle \right\}$$

$$\mathcal{E}_{2,\epsilon}^{(t)} = \left\{ \hat{\boldsymbol{\mu}}(t) \in \mathcal{S}_{i^\star(\boldsymbol{\mu})} \text{ and } \left| F_{\hat{\boldsymbol{\mu}}_L}(\boldsymbol{\omega}) - F_{\boldsymbol{\mu}}(\boldsymbol{\omega}) \right| < \epsilon, \ \forall \boldsymbol{\omega} \in \mathring{\Sigma} \right\}.$$

$\mathcal{E}_{1,\epsilon}^{(t)}$ can be seen as the event that the error of solution in FW-update (11) is bounded by $\epsilon$, which yields that $F_{\boldsymbol{\mu}}(\boldsymbol{x}(t))$ converges to $F_{\boldsymbol{\mu}}(\boldsymbol{\omega}^\star)$. As a consequence of the tracking rule, $F_{\boldsymbol{\mu}}(\boldsymbol{\omega}(t))$ converges to $F_{\boldsymbol{\mu}}(\boldsymbol{\omega}^\star)$ as well. More precisely, as stated in Lemma 3 at the end of this appendix, under $\mathcal{E}_{1,\epsilon}(T)$, $F_{\boldsymbol{\mu}}(\boldsymbol{\omega}^\star) - F_{\boldsymbol{\mu}}(\boldsymbol{\omega}(t)) < 5\epsilon$. Now, $\mathcal{E}_{2,\epsilon}^{(t)}$ is the event that the error of objective function is bounded by $\epsilon$ uniformly, so that FWS can stop while it is close to the real maximum. Overall, under $\mathcal{E}_{1,\epsilon}(T) \cap \mathcal{E}_{2,\epsilon}(T)$, we obtain that

$$\min\{\tau, T\} \leq \overline{h}(T) + \sum_{t=\overline{h}(T)}^{T} \mathbb{1}\{\tau > t\}$$

$$\leq \overline{h}(T) + \sum_{m=\overline{h}(T)}^{T} \mathbb{1}\{t F_{\hat{\boldsymbol{\mu}}(t)}(\boldsymbol{\omega}(t)) < \beta(t, \delta)\}$$

$$\leq \overline{h}(T) + \sum_{t=\overline{h}(T)}^{T} \mathbb{1}\{t(F_{\boldsymbol{\mu}}(\boldsymbol{\omega}^\star(\boldsymbol{\mu})) - 6\epsilon) < \beta(t, \delta)\}$$

$$\leq \overline{h}(T) + \frac{\beta(T, \delta)}{F_{\boldsymbol{\mu}}(\boldsymbol{\omega}^\star(\boldsymbol{\mu})) - 6\epsilon},$$

where the third inequality is due to the fact that under event $\mathcal{E}_{2,\epsilon}(T)$ and in view of Lemma 3, when $t \geq \overline{h}(T)$, we have $F_{\hat{\boldsymbol{\mu}}(t)}(\boldsymbol{\omega}(t)) \geq F_{\boldsymbol{\mu}}(\boldsymbol{\omega}(t)) - \epsilon \geq F_{\boldsymbol{\mu}}(\boldsymbol{\omega}^\star(\boldsymbol{\mu})) - 6\epsilon$.

Now introduce a constant

$$T_0(\delta) = \inf\{T \in \mathbb{N} : \overline{h}(T) + \frac{\beta(T, \delta)}{F_{\boldsymbol{\mu}}(\boldsymbol{\omega}^\star(\boldsymbol{\mu})) - 6\epsilon} \leq T\}.$$

The above inequalities show that $\mathcal{E}_{1,\epsilon}(T) \cap \mathcal{E}_{2,\epsilon}(T) \subset \{\tau \leq T\}$. Therefore,

$$\mathbb{E}_{\boldsymbol{\mu}}[\tau] \leq \sum_{T=1}^{\infty} \mathbb{P}_{\boldsymbol{\mu}}[\tau \geq T]$$

$$\leq \left(\frac{32D + 3L}{\epsilon}\right)^{11} + T_{\epsilon, L}^{\frac{11}{8}} + (4K + 1)^{\frac{11}{8}} + T_0(\delta) + \sum_{T=M+1}^{\infty} \mathbb{P}_{\boldsymbol{\mu}}[(\mathcal{E}_{1,\epsilon}(T) \cap \mathcal{E}_{1,\epsilon}(T))^c].$$

(41)

The term $\sum_{T \geq 1} \mathbb{P}_{\boldsymbol{\mu}}[(\mathcal{E}_{1,\epsilon}(T) \cap \mathcal{E}_{2,\epsilon}(T))^c]$ on the right-hand side of the inequality (41) can be upper bounded by concentration inequalities, which we summarize in Lemma 2 and prove in the next appendix. As for $T_0(\delta)$, we further introduce another small constant $\tilde{\epsilon} \in (0, 1)$ and observe that

$$T - \overline{h}(T) \geq \frac{T}{1 + \tilde{\epsilon}} \text{ when } T \geq \left(\frac{2}{\tilde{\epsilon}}\right)^{11}.$$

Therefore, based on the above fact, and (7),

$$T_0(\delta) \leq \left(\frac{2}{\tilde{\epsilon}}\right)^{11} + \inf\left\{T \in \mathbb{N} : \frac{\beta(T,\delta)}{F_{\boldsymbol{\mu}}(\boldsymbol{\omega}^\star(\boldsymbol{\mu})) - 6\epsilon} \leq \frac{T}{1+\tilde{\epsilon}}\right\}$$

$$\leq \max\left\{\left(\frac{2}{\tilde{\epsilon}}\right)^{11}, c_1(\Lambda)\right\} + \inf\left\{T \in \mathbb{N} : \frac{1}{F_{\boldsymbol{\mu}}(\boldsymbol{\omega}^\star(\boldsymbol{\mu})) - 6\epsilon}\log(\frac{c_2(\Lambda)T}{\delta}) \leq \frac{T}{1+\tilde{\epsilon}}\right\}$$

$$\leq \max\left\{\left(\frac{2}{\tilde{\epsilon}}\right)^{11}, c_1(\Lambda)\right\}$$

$$+ \frac{1+\tilde{\epsilon}}{F_{\boldsymbol{\mu}}(\boldsymbol{\omega}^\star(\boldsymbol{\mu})) - 6\epsilon}\left[\log\left(\frac{(1+\tilde{\epsilon})c_2(\Lambda)e}{\delta(F_{\boldsymbol{\mu}}(\boldsymbol{\omega}^\star(\boldsymbol{\mu})) - 6\epsilon)}\right) + \log\log\left(\frac{(1+\tilde{\epsilon})c_2(\Lambda)}{\delta(F_{\boldsymbol{\mu}}(\boldsymbol{\omega}^\star(\boldsymbol{\mu})) - 6\epsilon)}\right)\right],$$

$$(42)$$

where the second inequality is due to (7) and the last inequality is a consequence of Lemma 4 with $\alpha = 1$, $c_1 = (F_{\boldsymbol{\mu}}(\boldsymbol{\omega}^\star(\boldsymbol{\mu})) - 6\epsilon)/(1+\tilde{\epsilon})$ and $c_2 = c_2(\Lambda)/\delta$. Substituting the upper bounds provided by (42) and Lemma 2 into (41), we obtain that

$$\limsup_{\delta \to 0} \frac{\mathbb{E}_{\boldsymbol{\mu}}[\tau]}{\log(1/\delta)} \leq \frac{1+\tilde{\epsilon}}{F_{\boldsymbol{\mu}}(\boldsymbol{\omega}^\star(\boldsymbol{\mu})) - 6\epsilon}.$$

Since $\epsilon$ and $\tilde{\epsilon}$ can be arbitrary small and $T^\star(\boldsymbol{\mu}) = \frac{1}{F_{\boldsymbol{\mu}}(\boldsymbol{\omega}^\star(\boldsymbol{\mu}))}$, we get the desired result.

### I.3 Additional lemmas

**Lemma 2.** *Under Assumptions 1, we have*

$$\sum_{T=M}^{\infty} \mathbb{P}_{\boldsymbol{\mu}}[(\mathcal{E}_{1,\epsilon}(T) \cap \mathcal{E}_{1,\epsilon}(T))^c] < \infty.$$

Refer to Appendix J for a proof.

**Lemma 3.** *For any $T \geq M = \max\{\left(\frac{32D+3L}{\epsilon}\right)^{11}, T_{\epsilon,L}^{\frac{11}{8}}, (4K+1)^{\frac{11}{8}}\}$, under event $\mathcal{E}_{1,\epsilon}(T) \cap \mathcal{E}_{2,\epsilon}(T)$ and Assumption 2, FWS algorithm with a sequence $\{r_t\}_{t \geq 1}$, satisfying (i), (ii) stated in Theorem 1 attains that*

$$F_{\boldsymbol{\mu}}(\boldsymbol{\omega}^\star) - F_{\boldsymbol{\mu}}(\boldsymbol{\omega}(t)) \leq 5\epsilon, \ \forall t = \overline{h}(T), \overline{h}(T)+1,\ldots,T.$$

Refer to Appendix L.3 for a proof.

**Lemma 4** (Lemma 18 in [20]). *For $\alpha \in [1, e/2]$, any two constants $c_1, c_2$,*

$$x = \frac{1}{c_1}\left[\log\left(\frac{c_2 e}{c_1^\alpha}\right) + \log\log\left(\frac{c_2}{c_1^\alpha}\right)\right]$$

*is such that $c_1 x \geq \log(c_2 x^\alpha)$.*

# J  Concentration Results

This section presents the proof of Lemma 2 and the necessary technical lemmas (see Appendix J.2). We restate the lemma:

**Lemma 2.** *Under Assumptions 1, we have*

$$\sum_{T=1}^{\infty} \mathbb{P}_{\boldsymbol{\mu}}[(\mathcal{E}_{1,\epsilon}(T) \cap \mathcal{E}_{1,\epsilon}(T))^c] < \infty.$$

## J.1  Proof of Lemma 2

We first derive sufficient conditions for the events $\mathcal{E}_{1,\epsilon}(T)$ and $\mathcal{E}_{2,\epsilon}(T)$ to hold separately. Then, we will conclude applying the concentration inequality.

**(i) The event $\mathcal{E}_{1,\epsilon}(T)$:**
Let $t = \underline{h}(T), \dots, T$. Applying the second part of Theorem 3 in Appendix K with $\boldsymbol{\omega} = \boldsymbol{x}(t-1)$, $r = r_t$, and $\boldsymbol{\pi} = \hat{\boldsymbol{\mu}}(t-1)$, and hence $\boldsymbol{z}(\boldsymbol{\omega}, r, \boldsymbol{\pi}) = \boldsymbol{z}(t)$, we get that: if $\|\hat{\boldsymbol{\mu}}(t-1) - \boldsymbol{\mu}\|_\infty < \xi_{1,\epsilon}$,

$$\max_{\boldsymbol{z} \in \Sigma} \min_{h \in H_{F_{\boldsymbol{\mu}}}(\boldsymbol{x}(t-1), r_t)} \langle \boldsymbol{z} - \boldsymbol{x}(t-1), h \rangle - \epsilon < \min_{h \in H_{F_{\boldsymbol{\mu}}}(\boldsymbol{x}(t-1), r_t)} \langle \boldsymbol{z}(t) - \boldsymbol{x}(t-1), h \rangle.$$

From the definition of $\mathcal{E}_{1,\epsilon}^{(t)}$, we deduce that:

$$\mathcal{E}_{1,\epsilon}^{(t)} \subset \{\|\hat{\boldsymbol{\mu}}(t-1) - \boldsymbol{\mu}\|_\infty < \xi_{1,\epsilon}\}, \ \forall t = \underline{h}(T), \dots, T.$$

**(ii) The event $\mathcal{E}_{2,\epsilon}(T)$:**
From Lemma 6 in Appendix K, we directly deduce that

$$\mathcal{E}_{2,\epsilon}^{(t)} \subset \{\|\hat{\boldsymbol{\mu}}(t) - \boldsymbol{\mu}\|_\infty < \xi_{2,\epsilon}\}, \ \forall t = \underline{h}(T), \dots, T.$$

Summarizing (i), (ii), we get that

$$\mathbb{P}_{\boldsymbol{\mu}}\left[(\mathcal{E}_{1,\epsilon}(T) \cap \mathcal{E}_{2,\epsilon}(T))^c\right] \leq \sum_{t=\underline{h}(T)-1}^{T} \sum_{k=1}^{K} \mathbb{P}_{\boldsymbol{\mu}}\left[|\hat{\mu}_k(t) - \mu_k| \geq \xi(\epsilon)\right], \tag{43}$$

where $\xi(\epsilon) = \min\{\xi_{1,\epsilon}, \xi_{2,\epsilon}\}$. To ensure the distance between the $\hat{\boldsymbol{\mu}}(t)$ and $\boldsymbol{\mu}$ is small, we need to pull each arm sufficiently often up to $t$. From Lemma 13 in Appendix M, we have

$$\min_k t x_k(t) \geq \sqrt{\frac{t}{K}} - 1, \forall t \geq 4K.$$

Hence, Lemma 12 from Appendix M implies that

$$N_k(t) \geq \sqrt{\frac{t}{K}} - K, \forall k \in [K], t \geq 4K.$$

Applying Chernoff inequalities yields that $\forall k \in [K], t \geq 4K$,

$$\mathbb{P}_{\boldsymbol{\mu}}\left[|\hat{\mu}_k(t) - \mu_k| \geq \xi(\epsilon)\right] \leq e^K \left[\exp\left(-\sqrt{t} A_k^-\right) + \exp\left(-\sqrt{t} A_k^+\right)\right], \tag{44}$$

where $A_k^- = d(\mu_k - \xi(\epsilon), \mu_k)/\sqrt{K}$ and $A_k^+ = d(\mu_k + \xi(\epsilon), \mu_k)/\sqrt{K}$. Substituting the upper bound (44) into (43), we get using a union bound for any $T \geq M$,

$$\mathbb{P}_{\boldsymbol{\mu}}\left[(\mathcal{E}_{1,\epsilon}(T) \cap \mathcal{E}_{2,\epsilon}(T))^c\right] \leq e^K \sum_{k=1}^{K} \sum_{t=\underline{h}(T)-1}^{T} \left[\exp\left(-\sqrt{t} A_k^-\right) + \exp\left(-\sqrt{t} A_k^+\right)\right]$$

$$\leq e^K \sum_{k=1}^{K} \int_{T^{\frac{8}{11}}}^{\infty} \left[\exp\left(-\sqrt{t} A_k^-\right) + \exp\left(-\sqrt{t} A_k^+\right)\right] dt,$$

where the second inequality follows from the definition (40) of $\underline{h}(T)$. We then apply Lemma 5 presented below with $\alpha = \frac{8}{11}, \beta = \frac{1}{2}$ and $A = A_k^+ (= A_k^-$ resp.) and deduce that

$$\sum_{T=M}^{\infty} \mathbb{P}_{\boldsymbol{\mu}}\left[(\mathcal{E}_{1,\epsilon}(T) \cap \mathcal{E}_{2,\epsilon}(T))^c\right] \leq e^K \sum_{k=1}^{K} \int_0^{\infty} \left(\int_{T^{\frac{8}{11}}}^{\infty} \exp\left(-\sqrt{t}A_k^-\right) + \exp\left(-\sqrt{t}A_k^+\right) dt\right) dT$$

$$= e^K \sum_{k=1}^{K} \frac{2\Gamma(\frac{19}{4})}{d(\mu_k - \xi(\epsilon), \mu_k)^{\frac{19}{4}}} + \frac{2\Gamma(\frac{19}{4})}{d(\mu_k + \xi(\epsilon), \mu_k)^{\frac{19}{4}}}$$

$$< 34 e^K \sum_{k=1}^{K} \frac{1}{d(\mu_k - \xi(\epsilon), \mu_k)^{\frac{19}{4}}} + \frac{1}{d(\mu_k + \xi(\epsilon), \mu_k)^{\frac{19}{4}}} < \infty,$$

where the second inequality is due to $\Gamma(\frac{19}{4}) < 17$. This concludes the proof.

## J.2 Technical lemmas

**Lemma 5.** *Let $\alpha, \beta \in (0, 1)$ and $A > 0$.*

$$\int_0^{\infty} \left(\int_{T^{\alpha}}^{\infty} \exp(-At^{\beta}) dt\right) dT = \frac{\Gamma(\frac{1}{\alpha\beta} + \frac{1}{\beta})}{\beta A^{\frac{1}{\alpha\beta} + \frac{1}{\beta}}}.$$

*Proof.*

$$\int_0^{\infty} \left(\int_{T^{\alpha}}^{\infty} \exp(-At^{\beta}) dt\right) dT = \int_0^{\infty} \alpha T^{\alpha} \exp(-AT^{\alpha\beta}) dT$$

$$= \frac{1}{\beta A^{\frac{1}{\alpha\beta} + \frac{1}{\beta}}} \int_0^{\infty} x^{\frac{1}{\alpha\beta} + \frac{1}{\beta} - 1} e^{-x} dx$$

$$= \frac{\Gamma(\frac{1}{\alpha\beta} + \frac{1}{\beta})}{\beta A^{\frac{1}{\alpha\beta} + \frac{1}{\beta}}}.$$

$\square$

# K  Continuity Arguments

The main goal of this section is to prove Proposition 1. We also state and prove Theorem 3 and Lemma 6. These results are used in Appendix J.

In the first subsection K.1, we present some of the ingredients used to establish our continuity results. The proofs of Proposition 1, Theorem 3 and Lemma 6 are presented in K.2, K.3 and K.4, respectively.

**Theorem 3.** *For any $\epsilon > 0$, there exist a constant $\xi_{1,\epsilon} > 0$, which depends on $\boldsymbol{\mu}$ and $\epsilon$, such that if $\|\boldsymbol{\pi} - \boldsymbol{\mu}\|_\infty < \xi_{1,\epsilon}$, then $\boldsymbol{\mu} \in \Lambda$,*

$$\left| \max_{\boldsymbol{z} \in \Sigma} \min_{h \in H_{F_{\boldsymbol{\pi}}}(\boldsymbol{\omega},r)} \langle \boldsymbol{z} - \boldsymbol{\omega}, h \rangle - \max_{\boldsymbol{z} \in \Sigma} \min_{h \in H_{F_{\boldsymbol{\mu}}}(\boldsymbol{\omega},r)} \langle \boldsymbol{z} - \boldsymbol{\omega}, h \rangle \right| < \frac{\epsilon}{2}, \ \forall (\boldsymbol{\omega}, r) \in \mathring{\Sigma} \times (0,1), \quad (45)$$

*and*

$$\left| \min_{h \in H_{F_{\boldsymbol{\pi}}(\boldsymbol{\omega},r)}} \langle \boldsymbol{z} - \boldsymbol{\omega}, h \rangle - \min_{h \in H_{F_{\boldsymbol{\mu}}(\boldsymbol{\omega},r)}} \langle \boldsymbol{z} - \boldsymbol{\omega}, h \rangle \right| < \frac{\epsilon}{2}, \ \forall (\boldsymbol{z}, \boldsymbol{\omega}, r) \in \Sigma \times \mathring{\Sigma} \times (0,1). \quad (46)$$

*As a consequence, if we fix some $(\boldsymbol{\omega}, r, \boldsymbol{\pi}) \in \mathring{\Sigma} \times (0,1) \times \Lambda$, where $\|\boldsymbol{\pi} - \boldsymbol{\mu}\|_\infty < \xi_{1,\epsilon}$, and further select $\boldsymbol{z}(\boldsymbol{\omega}, r, \boldsymbol{\pi}) \in \mathrm{argmax}_{\boldsymbol{z} \in \Sigma} \min_{h \in H_{F_{\boldsymbol{\pi}}}(\boldsymbol{\omega},r)} \langle \boldsymbol{z} - \boldsymbol{\omega}, h \rangle$, the above two inequalities yield that*

$$\max_{\boldsymbol{z} \in \Sigma} \min_{h \in H_{F_{\boldsymbol{\mu}}}(\boldsymbol{\omega},r)} \langle \boldsymbol{z} - \boldsymbol{\omega}, h \rangle - \epsilon < \min_{h \in H_{F_{\boldsymbol{\mu}}}(\boldsymbol{\omega},r)} \langle \boldsymbol{z}(\boldsymbol{\omega}, r, \boldsymbol{\pi}) - \boldsymbol{\omega}, h \rangle.$$

**Lemma 6.** *For any $\epsilon > 0$, there is $\xi_{2,\epsilon} > 0$, which depends on $\boldsymbol{\mu}$ and $\epsilon$, s.t. if $\|\boldsymbol{\pi} - \boldsymbol{\mu}\|_\infty < \xi_{2,\epsilon}$, then*

$$\boldsymbol{\pi} \in \mathcal{S}_{i^\star(\boldsymbol{\mu})} \text{ and } |F_{\boldsymbol{\pi}}(\boldsymbol{\omega}) - F_{\boldsymbol{\mu}}(\boldsymbol{\omega})| < \epsilon, \forall \boldsymbol{\omega} \in \mathring{\Sigma}.$$

## K.1  Continuity and differentiablity of value functions

We introduce some definitions and results taken from [16], and also used recently in [11, 10] in the bandit literature. [11] concerns the continuity of the optimal allocation when there are multiple correct answers for active learning. [10] applies it for the regret minimization problem, but it is restricted to the single-valued analysis.

**Definition 2.** *Let $f : U \to \mathbb{R}$ be a function where $U$ is a non-empty subset of a topological space. The level sets of $f$ is defined as for $y \in \mathbb{R}$,*

$$L_f(y, U) = \{x \in U \ : \ f(x) \leq y\},$$
$$L_f^<(y, U) = \{x \in U \ : \ f(x) < y\}.$$

*We say that $f$ is **lower semi-continuous** on $U$ if all the level sets $L_f(y, U)$ are closed. It is **inf-compact** on $U$ if all these level sets are compact. And it is **upper semi-continuous** if all the strict level sets $L_f^<(y, U)$ are open.*

Suppose $\mathbb{X}$ and $\mathbb{Y}$ are Hausdorff topological spaces. Let $u : \mathbb{X} \times \mathbb{Y} \to \mathbb{R}$ be a function and $\Phi : \mathbb{X} \rightrightarrows \mathbb{S}(\mathbb{Y})$ be a set-valued function, where $\mathbb{S}(\mathbb{Y})$ is the set of non-empty subsets of $\mathbb{Y}$. We are interest in a minimization problem of the form:

$$v(x) = \inf_{y \in \Phi(x)} u(x, y),$$
$$\Phi^*(x) = \{y \in \Phi(x) \ : \ u(x, y) = v(x)\}.$$

For $U \subset \mathbb{X}$, let the graph of $\Phi$ restricted to $U$ be $Gr_U(\Phi) = \{(x, y) \in U \times \mathbb{Y} \ : \ y \in \Phi(x)\}$.

**Definition 3.** *A function $u : \mathbb{X} \times \mathbb{Y} \to \overline{\mathbb{R}}$ is called $\mathbb{K}$-**inf-compact** on $Gr_{\mathbb{X}}(\Phi)$ if for all non-empty compact subset $C$ of $\mathbb{X}$, $u$ is inf-compact on $Gr_C(\Phi)$.*

There are two versions of Berge's theorem used in our paper. The first one asks $\Phi$ to be compact-valued. The second one relaxes this assumption but requires an additional assumption on the object function $u$. Besides, we introduce $\mathbb{K}(\mathbb{X}) = \{F \in \mathbb{S}(\mathbb{X}) : F$ is compact $\}$.

**Theorem 4** ([4]). *Let $\mathbb{X}$ and $\mathbb{Y}$ be Hausdorff topological spaces. Assume that*

- *$\Phi : \mathbb{X} \rightrightarrows \mathbb{K}(\mathbb{X})$ is continuous (i.e. both lower and upper hemicontinuous),*
- *$u : \mathbb{X} \times \mathbb{Y} \to \mathbb{R}$ is continuous.*

*Then the function $v : \mathbb{X} \to \mathbb{R}$ is continuous and the solution multifunction $\Phi^* : \mathbb{X} \to \mathbb{S}(\mathbb{Y})$ is upper hemicontinuous and compact valued.*

**Theorem 5** ([16]). *Assume that*

- *$\mathbb{X}$ is compactly generated,*
- *$\Phi : \mathbb{X} \rightrightarrows \mathbb{S}(\mathbb{Y})$ is lower hemicontinuous,*
- *$u : \mathbb{X} \times \mathbb{Y} \to \mathbb{R}$ is $\mathbb{K}$-inf-compact and upper semi-continuous on $Gr_{\mathbb{X}}(\Phi)$.*

*Then the function $v : \mathbb{X} \to \mathbb{R}$ is continuous and the solution multifunction $\Phi^* : \mathbb{X} \rightrightarrows \mathbb{S}(\mathbb{Y})$ is upper hemicontinuous and compact valued.*

All the above theorems are about the continuity of the value function of a parameterized optimization problem. We also need an additional lemma to guarantee the differentiability. The following lemma is one of the variant of the envelope theorem, which provides an important tool in optimization and has several applications in economics.

**Lemma 7** (Corollary 299 in [6]). *Let $\mathbb{X}$ be a metric space and $Y$ is a nonempty open subset in $\mathbb{R}^K$. Let $u : \mathbb{X} \times Y \to \mathbb{R}$ and assume $\frac{\partial u}{\partial y}$ exists and is continuous in $\mathbb{X} \times Y$. For each $y \in Y$, let $x^\star(y)$ minimizes $u(x, y)$ over $x \in \mathbb{X}$. Set*

$$v(y) = u(x^\star(y), y).$$

*Assume that $x^\star : Y \to \mathbb{X}$ is a continuous function. Then $v$ is continuously differentiable and*

$$\frac{d}{dy}v(y) = \frac{\partial u}{\partial y}\left(x^\star(y), y\right).$$

### K.2 Proof of Proposition 1

With fixed $j \in \mathcal{J}_i$, we prove the proposition in two steps.

**(i) $f_j$ is continuous on $\Sigma \times \mathcal{S}_i$ and $\overline{\lambda}_j$ is unique, continuous on $\mathring{\Sigma} \times \mathcal{S}_i$.**

We apply Theorem 4 with the following substitutions:

- $\mathbb{X} = \Sigma \times \mathcal{S}_i$,
- $\mathbb{Y} = \mathrm{cl}(\mathcal{C}_j^i)$,
- $\Phi(\boldsymbol{\omega}, \boldsymbol{\mu}) = \mathrm{cl}(\mathcal{C}_j^i)$,
- $u(\boldsymbol{\omega}, \boldsymbol{\mu}, \boldsymbol{\lambda}) = \sum_{k=1}^K \omega_k d(\mu_k, \lambda_k)$.

As $\Phi$ is a constant correspondence and $u$ is a continuous mapping, we immediate obtain that $\overline{\lambda}_j$ is upper hemicontinuous and $f_j$ is continuous on $\Sigma \times \mathcal{S}_i$ by Theorem 4.

Observe that $d(\mu, \cdot)$ is strictly convex on $\mathrm{cl}(\mathcal{C}_j^i)$ when the distributions are from a one-parameter exponential family and recall $\min_k \omega_k > 0$ for all $\boldsymbol{\omega} \in \mathring{\Sigma}$, so is the weighted sum. We conclude that the uniqueness of the solution function, $\overline{\lambda}_j$, stems from the strict convexity of the objective function. Thus, the continuity of $\overline{\lambda}_j$ holds as the consequence of the uniqueness and its upper hemicontinuity.

**(ii) $f_j$ is differentiable on $\mathring{\Sigma} \times \mathcal{S}_i$ and $\nabla_{\boldsymbol{\omega}} f_j(\boldsymbol{\omega}, \boldsymbol{\mu}) = \sum_{k=1}^K d(\mu_k, \overline{\lambda_j(\boldsymbol{\omega}, \boldsymbol{\mu})}_k)$ on $\mathring{\Sigma} \times \mathcal{S}_i$.**

This is a consequence of Lemma 7 using the following substitutions:

- $\mathbb{X} = \mathrm{cl}(\mathcal{C}_j^i)$,
- $Y = \mathring{\Sigma} \times \mathcal{S}_i$,
- $x^\star(\boldsymbol{\omega}, \boldsymbol{\mu}) = \overline{\lambda_j(\boldsymbol{\omega}, \boldsymbol{\mu})}$,
- $u(\overline{\lambda_j(\boldsymbol{\omega}, \boldsymbol{\mu})}, \boldsymbol{\omega}, \boldsymbol{\mu}) = \sum_{k=1}^K \omega_k d(\mu_k, \overline{\lambda_j(\boldsymbol{\omega}, \boldsymbol{\mu})}_k)$.

Under these substitutions, $f_j$ is continuously differentiable as $x^\star$ is continuous from (i). The results follow directly.

### K.3 The continuity of solution of (11) – Proof of Theorem 3

Before we prove the theorem, we state and prove some preliminary results. For the simplicity and clarity, we make a convention that $\boldsymbol{\mu} \in \mathcal{S}_i$ for some $i \in \mathcal{I}$. We also define the maps $\psi_1$ and $\psi_2$ as:

$$\psi_1 : (\boldsymbol{\omega}, r, \boldsymbol{\pi}, \boldsymbol{z}) \mapsto \min_{h \in H_{F_{\boldsymbol{\pi}}}(\boldsymbol{\omega}, r)} \langle \boldsymbol{z} - \boldsymbol{\omega}, h \rangle,$$

$$\psi_2 : (\boldsymbol{\omega}, r, \boldsymbol{\pi}) \mapsto \max_{\boldsymbol{z} \in \Sigma} \min_{h \in H_{F_{\boldsymbol{\pi}}}(\boldsymbol{\omega}, r)} \langle \boldsymbol{z} - \boldsymbol{\omega}, h \rangle.$$

**Lemma 8.** $\psi_1(\boldsymbol{\omega}, r, \boldsymbol{\pi}, \boldsymbol{z})$ *is continuous on* $\mathring{\Sigma} \times (0, 1) \times \mathcal{S}_i \times \Sigma$.

*Proof.* We apply Theorem 4 with the following substitutions:

- $\mathbb{X} = \mathring{\Sigma} \times (0, 1) \times \mathcal{S}_i \times \Sigma$,
- $\mathbb{Y} = \mathbb{R}^K$,
- $\Phi(\boldsymbol{\omega}, r, \boldsymbol{\pi}, \boldsymbol{z}) = H_{F_{\boldsymbol{\pi}}}(\boldsymbol{\omega}, r)$,
- $u(\boldsymbol{\omega}, r, \boldsymbol{\pi}, \boldsymbol{z}, h) = \langle \boldsymbol{z} - \boldsymbol{\omega}, h \rangle$.

As $u$ is obviously continuous, it only remains to prove that $\Phi$ is continuous.

Let $\{(\boldsymbol{\omega}_n, r_n, \boldsymbol{\pi}_n)\}_{n=1}^\infty$ be a sequence converging to $(\boldsymbol{\omega}, r, \boldsymbol{\pi}) \in \mathring{\Sigma} \times (0, 1) \times \mathcal{S}_i$. Also, let $H_{F_{\boldsymbol{\pi}}}(\boldsymbol{\omega}, r) = \mathrm{cov}\{\nabla_{\boldsymbol{\omega}} f_{j_m}(\boldsymbol{\omega}, \boldsymbol{\pi})\}_{m=1}^M$ for some $\{j_m\}_{m=1}^M \in \mathcal{J}_i$. Arbitrarily select $h \in H_{F_{\boldsymbol{\pi}}}(\boldsymbol{\omega}, r)$. Then there exists $\alpha_1, \ldots, \alpha_m \geq 0$ such that

$$\sum_{m=1}^M \alpha_{j_m} = 1 \text{ and } h = \sum_{m=1}^M \alpha_{j_m} \nabla_{\boldsymbol{\omega}} f_{j_m}(\boldsymbol{\omega}, \boldsymbol{\pi}).$$

As $(\boldsymbol{\omega}_n, r_n, \boldsymbol{\pi}_n) \xrightarrow{n \to \infty} (\boldsymbol{\omega}, r, \boldsymbol{\pi})$ and $\{f_j\}_{j \in \mathcal{J}_i}$ are continuous from Proposition 1, there is $N \in \mathbb{N}$ such that

$$\nabla_{\boldsymbol{\omega}} f_{j_m} \in H_{F_{\boldsymbol{\pi}}(\boldsymbol{\omega}_n, r_n)}, \text{ or equivalently } f_{j_m}(\boldsymbol{\omega}_n, \boldsymbol{\pi}_n) < F(\boldsymbol{\omega}_n, \boldsymbol{\pi}_n) + r_n,$$

for all $m = 1, \ldots, M, \ n \geq N$. In the following, we show lower and upper hemicontinuity for $\Phi$ separately.

**Lower hemicontinuity:**
For $n \geq N$, we select $h_n = \sum_{m=1}^M \alpha_{j_m} \nabla_{\boldsymbol{\omega}} f_{j_m}(\boldsymbol{\omega}_n, \boldsymbol{\pi}_n)$ then $h_n \xrightarrow{n \to \infty} h$ as $\nabla_{\boldsymbol{\omega}} f_{j_m}$'s are continuous by Proposition 1. This implies the lower hemicontinuity of $\Phi$

**Upper hemicontinuity:**
Let $\mathcal{U}$ be an open set containing $H_{F_{\boldsymbol{\pi}}}(\boldsymbol{\omega}, r) = \mathrm{cov}\{\nabla_{\boldsymbol{\omega}} f_{j_m}(\boldsymbol{\omega}, \boldsymbol{\pi})\}_{m=1}^M$. Because $\mathcal{U}$ is open, there exist $\epsilon > 0$ such that $H_{F_{\boldsymbol{\pi}}}(\boldsymbol{\omega}, r) + B(0, \epsilon) \subset \mathcal{U}$, where $+$ is a Minkowski addition and $B(0, \epsilon)$ is the $K$-dimensional ball with diameter $\epsilon$. According to Proposition 1, there exists an integer $N' \geq N$ such that $\|\nabla_{\boldsymbol{\omega}} f_{j_m}(\boldsymbol{\omega}_n, \boldsymbol{\pi}_n) - \nabla_{\boldsymbol{\omega}} f_{j_m}(\boldsymbol{\omega}, \boldsymbol{\pi})\|_\infty < \epsilon$ for all $m = 1, \ldots, M, \ n \geq N'$. Thus, if $n \geq N'$,

$$H_{F_{\boldsymbol{\pi}_n}}(\boldsymbol{\omega}_n, r_n) = \mathrm{cov}\{\nabla_{\boldsymbol{\omega}} f_{j_m}(\boldsymbol{\omega}_n, \boldsymbol{\pi}_n)\}_{m=1}^M\} \subset H_{F_{\boldsymbol{\pi}}}(\boldsymbol{\omega}, r) + B(0, \epsilon) \subset \mathcal{U},$$

and the upper hemicontinuity follows.

Summarizing, by continuity of $\Phi$ and $u$, we conclude that $\psi_1$ is also continuous by Theorem 4. $\quad\square$

**Lemma 9.** $\psi_2(\boldsymbol{\omega}, r, \boldsymbol{\pi})$ *is continuous on* $\mathring{\Sigma} \times (0, 1) \times \mathcal{S}_i$.

*Proof.* We apply Theorem 4 with the following substitutions:

- $\mathbb{X} = \mathring{\Sigma} \times (0, 1) \times \mathcal{S}_i$,
- $\mathbb{Y} = \Sigma$,
- $\Phi(\boldsymbol{\omega}, r, \boldsymbol{\pi}) = \Sigma$,
- $u(\boldsymbol{\omega}, r, \boldsymbol{\pi}, \boldsymbol{z}) = \psi_1(\boldsymbol{\omega}, r, \boldsymbol{\pi}, \boldsymbol{z})$.

From Lemma 8, $\psi_1$ is continuous. Notice that $\Phi$ is a constant map and hence continuous, so Theorem 4 directly implies the conclusion.

$\square$

We are now ready to prove the theorem.

**Proof of Theorem 3:** We prove the inequality (45) using Lemma 9. The inequality (46) can be obtained analogously using Lemma 8. Let $\phi$ be a function defined on $\mathcal{S}_i$ as

$$\phi(\boldsymbol{\pi}) = \min\left\{-|\psi_2(\boldsymbol{\omega}, r, \boldsymbol{\pi}) - \psi_2(\boldsymbol{\omega}, r, \boldsymbol{\mu})| : (\boldsymbol{\omega}, r) \in \mathring{\Sigma} \times (0, 1)\right\}.$$

**$\phi$ is a continuous function on $\mathring{\Sigma} \times \mathcal{S}_i$:**

We apply Theorem 5 with the following substitutions:

- $\mathbb{X} = \mathcal{S}_i$,
- $\mathbb{Y} = \mathring{\Sigma} \times (0, 1)$,
- $\Phi(\boldsymbol{\pi}) = \mathring{\Sigma} \times (0, 1)$,
- $u(\boldsymbol{\lambda}, \boldsymbol{\omega}, r) = -|\psi_2(\boldsymbol{\omega}, r, \boldsymbol{\pi}) - \psi_2(\boldsymbol{\omega}, r, \boldsymbol{\mu})|$.

As $\mathbb{X} = \mathcal{S}_i$ is a metric space, it is compactly generated. $\Phi$ is continuous for it is a constant map. As for $u$, the upper semi-continuity follows from Lemma 9. It only remains to show that $u$ is $\mathbb{K}$-inf compact. Let $C \subset \mathcal{S}_i$ be a compact set and let $y \in \mathbb{R}$. We show that $L_u(y, C \times \mathring{\Sigma} \times (0, 1))$ is a compact by checking that it is bounded and closed. Boundness directly follows from the fact $\mathring{\Sigma} \times (0, 1)$ is bounded and $C$ is compact. As for closeness, $u$ is a continuous function from Lemma 9, which also implies that $L_u(y, C \times \mathring{\Sigma} \times (0, 1))$ is closed. Thus Theorem 5 implies that $\phi$ is a continuous function.

By definition of $\phi$, $\phi(\boldsymbol{\mu}) = 0$. Since $\phi$ is continuous, there exists $\xi_{1,\epsilon}$ such that $\phi(\boldsymbol{\pi}) > -\epsilon/2$ for all $|\boldsymbol{\pi} - \boldsymbol{\mu}| < \xi_{1,\epsilon}$. In other words, the inequality (45) holds.

## K.4 Proof of Lemma 6

Assume $i^\star(\boldsymbol{\mu}) = i$ for some $i \in \mathcal{I}$ for clarity. According to Assumption 1, $\mathcal{S}_i$ is open, and we know that $\boldsymbol{\pi} \in \mathcal{S}_i$ when $\boldsymbol{\pi}$ is close enough to $\boldsymbol{\mu}$. Hence, it remains to show that there exists a constant $\xi_{2,\epsilon} > 0$ such that $|F_{\boldsymbol{\pi}}(\boldsymbol{\omega}) - F_{\boldsymbol{\mu}}(\boldsymbol{\omega})| < \frac{\epsilon}{2}$, for all $|\boldsymbol{\pi} - \boldsymbol{\mu}| < \xi_{2,\epsilon}$. We consider a function $\phi$, which is defined below, and show its continuity.

$$\phi(\boldsymbol{\pi}) = \min_{\boldsymbol{\omega} \in \mathring{\Sigma}} -|F_{\boldsymbol{\pi}}(\boldsymbol{\omega}) - F_{\boldsymbol{\mu}}(\boldsymbol{\omega})|, \ \forall \boldsymbol{\pi} \in \mathcal{S}_i.$$

**$\phi$ is a continuous function on $\mathcal{S}_i$:**

We apply Theorem 5 with the following substitutions:

- $\mathbb{X} = \mathcal{S}_i$,
- $\mathbb{Y} = \mathring{\Sigma}$,
- $\Phi(\boldsymbol{\lambda}) = \mathring{\Sigma}$,
- $u(\boldsymbol{\lambda}) = -|F_{\boldsymbol{\pi}}(\boldsymbol{\omega}) - F_{\boldsymbol{\mu}}(\boldsymbol{\omega})|$.

As $\mathbb{X} = \mathcal{S}_i$ is a metric space, it is compactly generated. $\Phi$ is continuous for it is a constant map. As for $u$, the upper semi-continuity is followed by Proposition 1. It only remains to show that $u$ is $\mathbb{K}$-inf compact. Let $C \subset \mathcal{S}_i$ be a compact set and let $y \in \mathbb{R}$. We show that $L_u(y, C \times \mathring{\Sigma})$ is a compact by checking that it is bounded and closed. Boundness directly follows from the fact $\mathring{\Sigma}$ is bounded and $C$ is compact. As for closeness, $u$ is a continuous function from Proposition 1, which also implies that $L_u(y, C \times \mathring{\Sigma})$ is closed. Theorem 5 hence implies that $\phi$ is a continuous function.

By the definition of $\phi$, $\phi(\boldsymbol{\mu}) = 0$. Since $\phi$ is continuous, there exists $\xi_{2,\epsilon}$ such that $\phi(\boldsymbol{\lambda}) > -\epsilon/2$, or equivalently $|F_{\boldsymbol{\pi}}(\boldsymbol{\omega}) - F_{\boldsymbol{\mu}}(\boldsymbol{\omega})| < \frac{\epsilon}{2}$, for all $|\boldsymbol{\pi} - \boldsymbol{\mu}| < \xi_{2,\epsilon}$. This completes the proof.

# L   Convergence of the Frank-Wolfe Algorithm

In this appendix, we study the performance of our variant of the FW algorithm. We assume that the real parameter $\mu$ is used in the updates rather than its estimate.

**Notations.** In the following, for brevity, we drop the subscript $\mu$. For instance, $F_\mu$ is replaced by $F$; $\mathcal{J}_{i^\star(\mu)}$ and $i^\star(\mu)$ become $\mathcal{J}$ and $i^\star$. We also use $\nabla f_j$ instead of $\nabla_\omega f_j$ (as we will not differentiate $f_j$ w.r.t. another argument).

## L.1   Smoothness of the objective function

We state below the main properties of the objective function $F$. These properties will be instrumental in our convergence analysis.

### L.1.1   $F$ is Lipschitz

**Proposition 3.** *$F$ is a $L$-Lipschitz function on $\Sigma$ with respect to the infinity norm.*

*Proof.* Recall Assumption 2 and apply of mean value theorem. We get that $f_j$'s are $L$-Lipschitz on $\mathring{\Sigma}$. As $f_j$'s are continuous functions on $\Sigma$ (see K.2 (i)), we can further extend the Lipschitzness from $\mathring{\Sigma}$ to $\Sigma$. Next we show that $F$ is $L$-Lipstchitz. For any $\boldsymbol{x}, \boldsymbol{y} \in \Sigma$, we have that

$$F(\boldsymbol{x}) = \min_{j \in \mathcal{J}} f_j(\boldsymbol{x}) \geq \min_{j \in \mathcal{J}} \left( f_j(\boldsymbol{y}) - L \left\| x - y \right\|_\infty \right)$$
$$\geq \min_{j \in \mathcal{J}} f_j(\boldsymbol{y}) - L \left\| x - y \right\|_\infty = F(\boldsymbol{y}) - L \left\| x - y \right\|_\infty.$$

This concludes the proof. $\qquad\qquad\square$

### L.1.2   Curvature of $F$

The definition (5) of the curvature and Assumption 2 allow us to bound the curvature of $f_j$ inside $\Sigma_\gamma$. The following Proposition states that inside $\Sigma_\gamma$, the first order approximation of $f_j$ remains controlled.

**Proposition 4.** *Let $\gamma \in (0, \frac{1}{K})$, $\boldsymbol{x} \in \Sigma_\gamma$ and $\boldsymbol{z} \in \Sigma$. Under Assumption 2, we have*

$$f_j(\boldsymbol{x}) + \langle \boldsymbol{y} - \boldsymbol{x}, \nabla f_j(\boldsymbol{x}) \rangle - f_j(\boldsymbol{y}) \leq \frac{8 D \alpha^2}{\gamma},$$

*where $j \in \mathcal{J}$ and $\boldsymbol{y} = \boldsymbol{x} + \alpha(\boldsymbol{z} - \boldsymbol{x})$ for some $\alpha \in (0, \frac{1}{2}]$.*

*Proof.* Let us drop the subscript $j$ in $f_j$ for clarity. Consider

$$\boldsymbol{u} = \frac{1}{2}(\boldsymbol{x} + \boldsymbol{z}). \tag{47}$$

As $\boldsymbol{x}, \boldsymbol{y}, \boldsymbol{z}, \boldsymbol{u}$ are on the same line, we can re-write $\boldsymbol{y}$ as:

$$\boldsymbol{y} = \boldsymbol{x} + \alpha(\boldsymbol{z} - \boldsymbol{x}) = 2\alpha \boldsymbol{u} + (1 - 2\alpha)\boldsymbol{x}.$$

The definition of $\boldsymbol{u}$ implies that $\boldsymbol{x}, \boldsymbol{u} \in \Sigma_{\frac{\gamma}{2}}$. Let $\alpha' = 2\alpha \in [0, 1]$, Assumption 2 (ii) and the definition (5) lead to

$$f(\boldsymbol{x}) + \langle \boldsymbol{y} - \boldsymbol{x}, \nabla f(\boldsymbol{x}) \rangle - f(\boldsymbol{y}) \leq \frac{2 D \alpha'^2}{\gamma} = \frac{8 D \alpha^2}{\gamma}.$$

$$\square$$

Next, consider $F = \min_{j \in \mathcal{J}} f_j$ instead of $f_j$.

**Corollary 1.** *Let* $\gamma \in (0, 1/K), r \in (0,1)$, $\boldsymbol{x} \in \Sigma_\gamma$ *and* $\boldsymbol{z} \in \Sigma$. *If* $\alpha$ *is a positive number s.t.* $\alpha < \min\{\frac{1}{2}, \frac{r}{L}\}$, *then*

$$F(\boldsymbol{y}) \geq F(\boldsymbol{x}) + \alpha \min_{h \in H_F(\boldsymbol{x}, r)} \langle \boldsymbol{z} - \boldsymbol{x}, h \rangle - \frac{8D\alpha^2}{\gamma},$$

*where* $\boldsymbol{y} = (1 - \alpha)\boldsymbol{x} + \alpha\boldsymbol{z}$.

*Proof.* If $F(\boldsymbol{x}) = f_j(\boldsymbol{x}) = f_j(\boldsymbol{y}) = F(\boldsymbol{y})$ for some $j \in \mathcal{J}$, the result directly follows from Proposition 4 as $\nabla f_j(\boldsymbol{x}) \in H_F(\boldsymbol{x}, r)$.

Otherwise, assume that we have two distinct $j_1, j_2 \in \mathcal{J}$ such that $F(\boldsymbol{x}) = f_{j_1}(\boldsymbol{x}) < f_{j_2}(\boldsymbol{x})$ and $F(\boldsymbol{y}) = f_{j_2}(\boldsymbol{y}) < f_{j_1}(\boldsymbol{y})$. As shown in the proof of Proposition 3, $f_j$ is $L$-Lipschitz and $\|\boldsymbol{x} - \boldsymbol{y}\|_\infty = \alpha \|\boldsymbol{z}\|_\infty < \frac{r}{L}$. We deduce that $f_{j_2}(\boldsymbol{x}) < F(\boldsymbol{x}) + r$, which is equivalent to $\nabla f_{j_2}(\boldsymbol{x}) \in H_F(\boldsymbol{x}, r)$. Consequently, choosing $h = \nabla f_{j_2}(\boldsymbol{x})$ yields that

$$F(\boldsymbol{x}) + \langle \boldsymbol{y} - \boldsymbol{x}, h \rangle - F(\boldsymbol{y}) \leq f_{j_2}(\boldsymbol{x}) + \langle \boldsymbol{y} - \boldsymbol{x}, h \rangle - f_{j_2}(\boldsymbol{y}) \leq \frac{8D\alpha^2}{\gamma},$$

where the last inequality is from Proposition 4. The corollary is proved. $\square$

### L.2 Properties of $H_\Phi(\boldsymbol{x}, r)$

Here we consider some functions, $\phi_1, \ldots, \phi_n$ on $\Sigma$ and define $\Phi(\boldsymbol{x}) = \min_i \phi_i(\boldsymbol{x}), \forall \boldsymbol{x} \in \Sigma$. It is clear that $\partial \Phi(\boldsymbol{x}) \subset H_\Phi(\boldsymbol{x}, r)$. The following result relates $H_\Phi(\boldsymbol{x}, r)$ to the $r$-subdifferential of $\Phi$. Recall that for $r \in (0, 1)$, the $r$-subdifferential of $\Phi$ is defined as $\partial_r \Phi(\boldsymbol{x}) = \{h \in \mathbb{R}^K : \Phi(\boldsymbol{y}) < \Phi(\boldsymbol{x}) + \langle \boldsymbol{y} - \boldsymbol{x}, h \rangle + r \text{ for all } \boldsymbol{y} \in \Sigma\}$.

**Lemma 10.** *If* $\Phi = \min_{i \in [n]} \phi_i$ *where* $\{\phi_j\}_{j=1}^n$ *are concave differentiable functions defined on* $\mathring{\Sigma}$, *then*

$$H_\Phi(\boldsymbol{x}, r) \subset \partial_r \Phi(\boldsymbol{x}), \forall x \in \mathring{\Sigma}, \ r > 0. \tag{48}$$

*Proof.* Let $\boldsymbol{x} \in \mathring{\Sigma}, r > 0$ be fixed and $\mathcal{A} = \{i \in [n] : \phi_i(\boldsymbol{x}) < \Phi(\boldsymbol{x}) + r\}$. Let $h \in H_\Phi(\boldsymbol{x}, r)$. It can be written as $h = \sum_{i \in \mathcal{A}} \alpha_i \nabla \phi_i(\boldsymbol{x}) \in H_\Phi(\boldsymbol{x}, r)$, where $\alpha_i \geq 0, \forall i \in \mathcal{A}$ and $\sum_{i \in \mathcal{A}} \alpha_i = 1$. Observe that for any $\boldsymbol{y} \in \mathring{\Sigma}, \Phi(\boldsymbol{y}) \leq \phi_i(\boldsymbol{y}), \forall i \in \mathcal{A}$. Thus,

$$\Phi(\boldsymbol{y}) - \Phi(\boldsymbol{x}) - \langle \boldsymbol{y} - \boldsymbol{x}, h \rangle < \sum_{i \in \mathcal{A}} \alpha_i \left[ \phi_i(\boldsymbol{y}) - \phi_i(\boldsymbol{x}) + r - \langle \boldsymbol{y} - \boldsymbol{x}, \nabla \phi_i(\boldsymbol{x}) \rangle \right] \leq \sum_{i \in \mathcal{A}} \alpha_i r = r,$$

where the last inequality stems for the concavity of the $\phi_i$'s. The above inequality, valid for any $h \in H_\Phi(\boldsymbol{x}, r)$, implies that $H_\Phi(\boldsymbol{x}, r) \subset \partial_r \Phi(\boldsymbol{x})$. $\square$

The next property is sometimes called primal-dual gap, see [24]. Interestingly, this property together with Lemma 10 tell us that the maxmin value computed at each iteration (11) can serve as an estimate of the gap.

**Lemma 11** ([41]). *Let* $\Phi = \min_{i \in [n]} \phi_i$ *where* $\{\phi_j\}_{j=1}^n$ *are concave differentiable functions defined on* $\Sigma$. *Then, for any* $\boldsymbol{x} \in \Sigma$,

$$\max_{\boldsymbol{z} \in \Sigma} \min_{h \in \partial_r \Phi(\boldsymbol{x})} \langle \boldsymbol{z} - \boldsymbol{x}, h \rangle \geq \max_{\boldsymbol{y} \in \Sigma} \Phi(\boldsymbol{y}) - \Phi(\boldsymbol{x}) - r.$$

### L.3 The convergence of `FWS` under $\mathcal{E}_{1,\epsilon}(T) \cap \mathcal{E}_{2,\epsilon}(T)$

Recall the definition of our "good" event (see Appendix I): $\mathcal{E}_{1,\epsilon}(T) = \left( \bigcap_{t=\underline{h}(T)}^T \mathcal{E}_{1,\epsilon}^{(t)} \right)$ and $\mathcal{E}_{2,\epsilon}(T) = \left( \bigcap_{t=\underline{h}(T)}^T \mathcal{E}_{2,\epsilon}^{(t)} \right)$ where

$$\mathcal{E}_{1,\epsilon}^{(t)} = \left\{ \max_{\boldsymbol{z} \in \Sigma} \min_{h \in H_{F_\mu}(\boldsymbol{x}(t-1), r_t)} \langle \boldsymbol{z} - \boldsymbol{x}(t-1), h \rangle - \epsilon < \min_{h \in H_{F_\mu}(\boldsymbol{x}(t-1), r_t)} \langle \boldsymbol{z}(t) - \boldsymbol{x}(t-1), h \rangle \right\},$$

$$\mathcal{E}_{2,\epsilon}^{(t)} = \left\{ \hat{\boldsymbol{\mu}}(t) \in \mathcal{S}_{i^\star(\boldsymbol{\mu})} \text{ and } \left| F_{\hat{\boldsymbol{\mu}}(t)}(\boldsymbol{\omega}) - F_\mu(\boldsymbol{\omega}) \right| < \epsilon, \forall \boldsymbol{\omega} \in \mathring{\Sigma} \right\}.$$

In what follows, we use the notation: $\Delta_t = F(\boldsymbol{\omega}^\star) - F(\boldsymbol{x}(t))$. In our convergence analysis, we first show that $\Delta_t$ is a decreasing sequence under $\mathcal{E}_{1,\epsilon}(T)$. Then, we prove that $\Delta_t$ becomes small when $t \geq \overline{h}(T)$.

**Theorem 6.** *Let $t \in \mathbb{N}$ satisfying that $\lfloor \sqrt{\frac{t}{K}} \rfloor \notin \mathbb{N}$ and $t \geq 4K$. Under the event $\mathcal{E}_{1,\epsilon}^{(t)} \cap \mathcal{E}_{2,\epsilon}^{(t)}$ and iteration (11) with parameter such that $L < r_t t$, we have*

$$\Delta_t \leq \frac{t-1}{t} \Delta_{t-1} + \frac{r_t + \epsilon}{t} + \frac{16D\sqrt{K}}{t^{\frac{3}{2}}}. \tag{49}$$

*Proof.* To simplify our presentation, we denote $\boldsymbol{y} = \boldsymbol{x}(t)$, $\boldsymbol{x} = \boldsymbol{x}(t-1)$ and $\boldsymbol{z} = \boldsymbol{z}(t)$. Also, let $\alpha$ be the step size $\frac{1}{t}$ and $r = r_t$.

Lemma 13 implies that $\boldsymbol{x} \in \Sigma_{\frac{1}{2\sqrt{tK}}}$ (when $t \geq 4K$), and hence, Corollary 1 with $\gamma = \frac{1}{2\sqrt{tK}}$ yields:

$$F(\boldsymbol{y}) \geq F(\boldsymbol{x}) + \alpha \min_{h \in H_F(\boldsymbol{x},r)} \langle \boldsymbol{z} - \boldsymbol{x}, h \rangle - 16D\alpha^2\sqrt{tK}$$

$$\geq F(\boldsymbol{x}) + \alpha \left( \max_{\boldsymbol{\omega} \in \Sigma} \min_{h \in H_F(\boldsymbol{x},r)} \langle \boldsymbol{\omega} - \boldsymbol{x}, h \rangle - \epsilon \right) - 16D\alpha^2\sqrt{tK}, \tag{50}$$

where the second inequality directly follows from the selection of $\boldsymbol{z}$ and the event $\mathcal{E}_{1,\epsilon}^{(t)}$. As $H_F(\boldsymbol{x},r) \subset \partial_r F(\boldsymbol{x})$, shown in Lemma 10, Lemma 11 implies that the second term in the right-hand side of inequality (50) can be lower bounded as:

$$\max_{\boldsymbol{\omega} \in \Sigma} \min_{h \in H_F(\boldsymbol{x},r)} \langle \boldsymbol{\omega} - \boldsymbol{x}, h \rangle - \epsilon \geq \max_{\boldsymbol{\omega} \in \Sigma} \min_{h \in \partial_r F(\boldsymbol{x})} \langle \boldsymbol{\omega} - \boldsymbol{x}, h \rangle - \epsilon$$

$$\geq \Delta_{t-1} - r - \epsilon. \tag{51}$$

Substituting the inequalities (51) into (50), we obtain that

$$F(\boldsymbol{y}) \geq F(\boldsymbol{x}) + \alpha \left( \Delta_{t-1} - r - \epsilon \right) - 16D\alpha^2\sqrt{tK}.$$

Subtracting $F(\boldsymbol{\omega}^\star)$ on both sides of the above inequality, we get that

$$\Delta_t \leq (1-\alpha)\Delta_{t-1} + \alpha(r+\epsilon) + 16D\alpha^2\sqrt{tK}. \tag{52}$$

The result follows from the inequality (52) and $\alpha = \frac{1}{t}$.

$\square$

The following theorem states the convergence of FWS. This convergence is obtained by repeatedly applying Theorem 6.

**Theorem 7.** *Let $\{r_t\}_{t\geq 1}$ be a sequence of positive numbers satisfying (i) $\lim_{t\to\infty} \frac{1}{t} \sum_{s=1}^t r_s = 0$, and (ii) $\lim_{t\to\infty} t r_t = \infty$. Suppose $T \geq \max\{ \left( \frac{32D+3L}{\epsilon} \right)^{11}, T_{\epsilon,L}^{\frac{11}{8}}, (4K+1)^{\frac{11}{8}} \}$, where $T_{\epsilon,L}$ is defined in (39). Then, under event $\mathcal{E}_{1,\epsilon}(T) \cap \mathcal{E}_{2,\epsilon}(T)$, applying FWS algorithm with $\{r_t\}_{t\geq 1}$, we have:*

$$F(\boldsymbol{\omega}^\star) - F(\boldsymbol{x}(t)) \leq 4\epsilon, \ \forall t = \overline{h}(T), \overline{h}(T)+1, \ldots, T.$$

*Proof.* We start the proof by dividing the time horizon into several blocks, where each block consists of $K$ successive rounds. We introduce $m$ as the index of the block, with a slight abuse of notation, we denote $\tilde{\Delta}_m = \Delta_{mK}$ as the gap at the end of the $m$-th block.

We provide recursive properties of $\tilde{\Delta}_m$ in two cases (a) $m$ is a square number, (b) $m$ is not a square number for $mK \geq \underline{h}(T)$.

**Step 1. Recursive properties of $\tilde{\Delta}_m$ under (a) and (b).**

For (a), $\boldsymbol{x}(mK) = \frac{1}{m}(\frac{1}{K}, \ldots, \frac{1}{K}) + \frac{m-1}{m}\boldsymbol{x}(mK - K)$. Proposition 3 directly yields that

$$\tilde{\Delta}_m \leq \tilde{\Delta}_{m-1} + \frac{L}{m}.$$

Since $\tilde{\Delta}_{m-1}$ is bounded by $L$, the above equation implies that

$$m\tilde{\Delta}_m \leq (m-1)\tilde{\Delta}_{m-1} + 2L. \tag{53}$$

For (b), recall that $T \geq T_{\epsilon,L}^{\frac{11}{8}}$, (39) and (40), for any $t \geq \underline{h}(T)$, we have that $t \geq \max\{T_{\epsilon,L}, 4K\}$. Thus, letting $Z = 16D\sqrt{K}$, Theorem 6 can be applied to derive a series of inequalities for $t = (m-1)K+1, \ldots, mK$:

$$[(m-1)K+1]\Delta_{(m-1)K+1} \leq (m-1)K\Delta_{(m-1)K} + \epsilon + \frac{Z}{[(m-1)K+1]^{\frac{1}{2}}} + r_{(m-1)K+1},$$

$$[(m-1)K+2]\Delta_{(m-1)K+2} \leq [(m-1)K+1]\Delta_{(m-1)K+2} + \epsilon$$
$$+ \frac{Z}{[(m-1)K+2]^{\frac{1}{2}}} + r_{(m-1)K+2},$$

$$\vdots$$

$$(mK)\Delta_{mK} \leq (mK-1)\Delta_{mK-1} + \epsilon + \frac{Z}{(mK)^{\frac{1}{2}}} + r_{mK}.$$

Summing over them and dividing by $K$ on both sides, we obtain that:

$$m\tilde{\Delta}_m \leq (m-1)\tilde{\Delta}_{(m-1)} + \epsilon + Z(m) + r(m), \tag{54}$$

where $Z(m) = \sum_{t=(m-1)K+1}^{mK} \frac{Z}{K\sqrt{t}}$ and $r(m) = \sum_{t=(m-1)K+1}^{mK} \frac{r_t}{K}$.

Our next step consists in studying the recursion between two successive square numbers. We introduce:

$$\underline{p} = \sqrt{\frac{\underline{h}(T)}{K}}, \quad \overline{p} = \sqrt{\frac{\overline{h}(T)}{K}}, \quad \text{and } \mathcal{P}(T) = \{p \in \mathbb{N} : \underline{h}(T) \leq Kp^2 \leq T\}. \tag{55}$$

**Step 2. For any $q \geq \overline{p}$, $\tilde{\Delta}_q \leq 3\epsilon$.**

We first fix some $p \in \mathcal{P}(T)$, summing the inequalities (54) over $m = p^2+1, \ldots, (p+1)^2 - 1$ and inequality (53) with $m = (p+1)^2$ gives:

$$(p+1)^2\tilde{\Delta}_{(p+1)^2} \leq p^2\tilde{\Delta}_{p^2} + 2p\epsilon + 2L + \sum_{m=p^2+1}^{(p+1)^2} Z(m) + r(m). \tag{56}$$

Then for any $q \geq \overline{p}$, we sum the inequalities (56) from $p = \underline{p}$ to $p = q-1$ and get that

$$q^2\tilde{\Delta}_{q^2} \leq \underline{p}^2\tilde{\Delta}_{\underline{p}^2} + 2\epsilon\sum_{p=\underline{p}}^{q-1} p + 2L(q - \underline{p} - 2) + \sum_{m=\underline{p}^2}^{q^2} Z(m) + r(m)$$

$$\leq \underline{p}^2 L + 2\epsilon\int_0^q t\,dt + 2Lq + \int_0^{q^2K} \frac{Z}{K\sqrt{t}}\,dt + \sum_{t=1}^{q^2K} \frac{r_t}{K}$$

$$\leq \frac{\underline{h}(T)L}{K} + \epsilon q^2 + 2Lq + \frac{2Zq}{\sqrt{K}} + \sum_{t=1}^{q^2K} \frac{r_t}{K}. \tag{57}$$

Recall from (40) and (55) that $\overline{h}(T) \geq T^{\frac{2}{11}}\underline{h}(T)$ and $q^2K \geq \overline{p}^2K = \overline{h}(T) \geq \max\{T^{\frac{2}{11}}K, T_{\epsilon,L}\}$. Divide by $q^2$ both sides of the inequality (57). We obtain:

$$\tilde{\Delta}_{q^2} \leq \frac{L}{T^{\frac{2}{11}}} + \epsilon + \frac{2(Z/\sqrt{K} + L)}{T^{\frac{1}{11}}} + \frac{1}{q^2K}\sum_{t=1}^{q^2K} r_t$$

$$\leq \epsilon + \frac{2Z/\sqrt{K} + 3L}{T^{\frac{1}{11}}} + \frac{1}{q^2K}\sum_{t=1}^{q^2K} r_t \leq 3\epsilon,$$

where the last inequality stems from $T \geq \left(\frac{32D+3L}{\epsilon}\right)^{11} = \left(\frac{2Z/\sqrt{K}+3L}{\epsilon}\right)^{11}$ and the definition of $T_{\epsilon,L}$ (see (39)).

**Step 3. For any $t \geq \overline{h}(T)$, we have $\Delta_t \leq 4\epsilon$.**

Now suppose $t \in \{Kq^2 + 1, \dots, K(q+1)^2 - 1\}$ for some $q \geq \overline{p}$. Recall that

$$\boldsymbol{x}(t) = \frac{Kq^2}{t}\boldsymbol{x}(Kq^2) + \frac{(t-Kq^2)}{t}\boldsymbol{u}, \text{ for some } \boldsymbol{u} \in \Sigma,$$

which yields that

$$\left\|\boldsymbol{x}(Kq^2) - \boldsymbol{u}\right\|_\infty \leq \frac{t-Kq^2}{t}\left\|\boldsymbol{u}\right\|_\infty \leq \frac{K(2q+1)}{Kq^2} \leq \frac{3}{q} \leq \frac{\epsilon}{L},$$

as (40) implies that $q \geq \frac{T^{\frac{2}{11}}\underline{h}(T)}{K} \geq \frac{3L}{\epsilon}$. Consequently, Proposition 3 yields

$$\left|F(\boldsymbol{x}(t)) - F(\boldsymbol{x}(Kq^2))\right| \leq \epsilon.$$

Combining this with the inequality from **Step 2**, we conclude that $F(\boldsymbol{\omega}^\star) - F(\boldsymbol{x}(t)) \leq 4\epsilon$ for all $t \geq \overline{h}(T)$.

$\square$

A consequence of Lemma 12 about the tracking rule (presented in the next appendix) and Theorem 7 is Lemma 3.

**Proof of Lemma 3.** Lemma 12 implies that $\left\|\boldsymbol{\omega}(t) - \boldsymbol{x}(t)\right\|_\infty \leq \frac{K-1}{t}$ and then by Proposition 3,

$$F(\boldsymbol{\omega}(t)) \geq F(\boldsymbol{x}(t)) - \frac{(K-1)L}{t} \geq F(\boldsymbol{x}(t)) - \epsilon,$$

where the last inequality is due to the fact that $t \geq \overline{h}(T) \geq T^{\frac{2}{11}}\underline{h}(T) \geq \frac{KL}{\epsilon}$ (see definition of $\underline{h}(T)$ and $\overline{h}(T)$ (40)). Combining Theorem 7 and the above inequality leads to the desired result. $\square$

# M    Tracking Rule

This section presents the analysis of the tracking rule in FWS and related results.

**Lemma 12** (Lemma 7 in [12]). *Let $\{z(s)\}_{s \in \mathbb{N}} \in \Sigma$ be a sequence of vectors such that $z(1), \ldots, z(K)$ are $e_1, \ldots, e_K$. We recursively define for $t \geq K$,*

$$\forall k \in [K], N_k(K) = 1,$$

$$\forall t \geq K + 1, \ A_t \in \operatorname*{argmax}_{k'} \frac{\sum_{s=1}^{t} z_{k'}(s)}{N_{k'}(t-1)}, \ \forall k \in [K], \ N_k(t) = \sum_{s=1}^{t} \mathbb{1}\{A_s = k\},$$

*(where the tie-breaking rule in the* argmax *is arbitrary). Then for all $t \geq K$, all $k \in [K]$,*

$$\sum_{s=1}^{t} z_k(s) - (K - 1) \leq N_k(t) \leq \sum_{s=1}^{t} z_k(s) + 1.$$

**Lemma 13.** *At any time $t \geq 4K$, under* FWS, *we have $x(t) \in \Sigma_{\sqrt{\frac{1}{Kt}} - \frac{1}{t}} \subset \Sigma_{\frac{1}{2\sqrt{tK}}}$.*

*Proof.* This lemma directly follows from the forced exploration procedure of FWS when $\lfloor t/K \rfloor$ is a square number, $x(t)$ move to the center of $\Sigma$ for $K$ successive rounds. Hence, for all $k = 1, \ldots, K$

$$t x_k(t) = \sum_{s=1}^{t} z_k(s) \geq \frac{1}{K} \sum_{s=1}^{t} \mathbb{1}\{z(s) = (1/K, \ldots, 1/K)\}$$

$$= \sqrt{\lfloor \frac{t}{K} \rfloor} \geq \sqrt{\frac{t}{K}} - 1 \geq \frac{1}{2}\sqrt{\frac{t}{K}}.$$

Dividing $t$ on the both sides, we get the result. $\qquad\square$

# N Non-asymptotic Sample Complexity

Looking back at our asymptotic sample complexity analysis, we note that the reason why we could not derive results for the mild confidence regime (non-asymptotic) is that we cannot quantify the cost paid for events $\mathbb{P}\left[(\mathcal{E}_{1,\epsilon} \cap \mathcal{E}_{2,\epsilon})^c\right]$ (see (41)-(42)). Looking in more details, the probability of these events cannot be precisely controlled because our continuity arguments (see Lemma 6 and Theorem 3 in Appendix K) rely on maximal theorems, and the constants $\xi_{1,\epsilon}$ and $\xi_{2,\epsilon}$ involved there have an unknown dependence in $\epsilon$.

To get non-asymptotic sample complexity upper bounds, we use mean value theorems instead, and obtain simple upper bounds of $\xi_{1,\epsilon}$ and $\xi_{2,\epsilon}$. This section is organized as follows: we present a stronger version of Lemma 6 and Theorem 3 in N.1 and N.2, respectively; the proof of Theorem 2 is then provided in N.3.

For convenience, we restate the additional assumption and our non-asymptotic sample complexity upper bound.

**Assumption 3.** *For any $\boldsymbol{\mu} \in \Lambda$, there exist constants $\kappa, E > 0$, s.t. if $\|\boldsymbol{\pi} - \boldsymbol{\mu}\|_\infty \leq \kappa$, then $\boldsymbol{\pi} \in \mathcal{S}_{i^\star(\boldsymbol{\mu})}, \forall \boldsymbol{\omega} \in \mathring{\Sigma}, \ j \in \mathcal{J}_{i^\star(\boldsymbol{\mu})}, \nabla_{\boldsymbol{\pi}} d(\pi_k, \overline{\boldsymbol{\lambda}_j(\boldsymbol{\omega}, \boldsymbol{\pi})}_k)$ is continuous and $\left\|\nabla_{\boldsymbol{\pi}} d(\pi_k, \overline{\boldsymbol{\lambda}_j(\boldsymbol{\omega}, \boldsymbol{\pi})}_k)\right\|_1 \leq E, \ \forall k = 1, \ldots, K$.*

**Theorem 2.** *Consider* `FWS` *algorithm with a sequence $\{r_t\}_{t \geq 1}$ as in Theorem 1. Under Assumptions 1, 2, and 3, the sample complexity $\tau$ of the algorithm satisfies: for any $\boldsymbol{\mu} \in \Lambda, \delta \in (0, 1)$, and any $\epsilon < \min\{\kappa E/2, 1\}, \tilde{\epsilon} < 1$,*

$$
\begin{aligned}
\mathbb{E}_{\boldsymbol{\mu}}\left[\tau\right] \leq &\frac{1 + \tilde{\epsilon}}{F_{\boldsymbol{\mu}}(\boldsymbol{\omega}^\star(\boldsymbol{\mu})) - 6\epsilon} \left[\log\left(\frac{(1 + \tilde{\epsilon})c_2(\Lambda)e}{\delta(F_{\boldsymbol{\mu}}(\boldsymbol{\omega}^\star(\boldsymbol{\mu})) - 6\epsilon)}\right) + \log\log\left(\frac{(1 + \tilde{\epsilon})c_2(\Lambda)}{\delta(F_{\boldsymbol{\mu}}(\boldsymbol{\omega}^\star(\boldsymbol{\mu})) - 6\epsilon)}\right)\right] \\
&+ \Psi(K, D, E, L, c_1(\Lambda), \epsilon) + T_{\epsilon, L}^{\frac{5}{4}},
\end{aligned}
$$

*where $T_{\epsilon, L}$ is a constant such if $t \geq T_{\epsilon, L}$, then $\sum_{s=1}^t r_s < t\epsilon$ and $tr_t > L$. The constant $\Psi$ is polynomial in $(D, E, L, c_1(\Lambda), 1/\epsilon)$ and exponential in $K$. The precise definition of $\Psi$ is given at the end of this section.*

## N.1 Continuity of the primal problem

First, we present the analogue of Lemma 6 (Appendix K).

**Lemma 14.** *Under Assumptions 1 and 3, for any $\boldsymbol{\mu} \in \Lambda$ and $\epsilon \in (0, \kappa E)$, if $\|\boldsymbol{\mu} - \boldsymbol{\pi}\|_\infty \leq \frac{\epsilon}{E}$, then*

$$
|F_{\boldsymbol{\pi}}(\boldsymbol{\omega}) - F_{\boldsymbol{\mu}}(\boldsymbol{\omega})| < \epsilon, \forall \boldsymbol{\omega} \in \mathring{\Sigma}.
$$

*Proof.* Fix $j \in \mathcal{J}_{i^\star(\boldsymbol{\mu})}$, and let $\boldsymbol{\omega} \in \mathring{\Sigma}$. Define the function $g : [0, 1] \to \mathbb{R}$ as

$$
g(t) = f_j(\boldsymbol{\omega}, t\boldsymbol{\mu} + (1 - t)\boldsymbol{\pi}).
$$

Since $\|\boldsymbol{\mu} - \boldsymbol{\pi}\|_\infty \leq \frac{\epsilon}{E} \leq \kappa$, Assumption 3 says that $t\boldsymbol{\mu} + (1 - t)\boldsymbol{\pi} \in \mathcal{S}_{i^\star(\boldsymbol{\mu})}$, which implies $g$ is well-defined. Based on the mean value theorem, there is $t \in (0, 1)$ s.t.

$$
\langle g'(t), \boldsymbol{\mu} - \boldsymbol{\pi} \rangle = f_j(\boldsymbol{\omega}, \boldsymbol{\mu}) - f_j(\boldsymbol{\omega}, \boldsymbol{\pi}). \tag{58}
$$

For clarity, we denote $t\boldsymbol{\mu} + (1 - t)\boldsymbol{\pi} = \tilde{\boldsymbol{\pi}}$ and its $k$-th component as $\tilde{\pi}_k$. Also, $\frac{\partial f_j}{\partial \boldsymbol{\mu}}(\boldsymbol{\omega}, \boldsymbol{\mu})$ is the partial derivative of $f_j(\boldsymbol{\omega}, \boldsymbol{\mu})$ w.r.t. $\boldsymbol{\mu}$. (58) yields that

$$
\begin{aligned}
|f_j(\boldsymbol{\omega}, \boldsymbol{\mu}) - f_j(\boldsymbol{\omega}, \boldsymbol{\pi})| = |g'(t)| &= \left|\left\langle \frac{\partial f_j}{\partial \boldsymbol{\mu}}(\boldsymbol{\omega}, \tilde{\boldsymbol{\pi}}), \boldsymbol{\mu} - \boldsymbol{\pi} \right\rangle\right| \\
&= \sum_{k=1}^K \omega_k \langle \nabla_{\tilde{\boldsymbol{\pi}}} d(\tilde{\pi}_k, \overline{\boldsymbol{\lambda}_j(\boldsymbol{\omega}, \tilde{\boldsymbol{\pi}})}_k), \boldsymbol{\mu} - \boldsymbol{\pi} \rangle \\
&\leq \sum_{k=1}^K \omega_k \left\|\nabla_{\tilde{\boldsymbol{\pi}}} d(\tilde{\pi}_k, \overline{\boldsymbol{\lambda}_j(\boldsymbol{\omega}, \tilde{\boldsymbol{\pi}})}_k)\right\|_1 \|\boldsymbol{\mu} - \boldsymbol{\pi}\|_\infty \leq \epsilon, \tag{59}
\end{aligned}
$$

where the last inequality is the result of Assumption 3, $\|\boldsymbol{\mu} - \boldsymbol{\pi}\|_\infty < \frac{\epsilon}{E}$ and $\boldsymbol{\omega} \in \overset{\circ}{\Sigma}$.

As for the objective function, we have

$$F_{\boldsymbol{\pi}}(\boldsymbol{\omega}) - F_{\boldsymbol{\mu}}(\boldsymbol{\omega}) = \min_j f_j(\boldsymbol{\omega}, \boldsymbol{\pi}) - \min_j f_j(\boldsymbol{\omega}, \boldsymbol{\mu}) \leq \min_j f_j(\boldsymbol{\omega}, \boldsymbol{\mu}) + \epsilon - \min_j f_j(\boldsymbol{\omega}, \boldsymbol{\mu}) = \epsilon,$$

where the inequality holds in view of (59). The other inequality follows similarly, which completes the proof. □

## N.2 Envelope theorem at a saddle point

As Lemma 14 in Appendix K, we wish to apply the envelop theorem for the perturbation analysis of the equation (11). For clarity, we redefine the notations used in Appendix K.

Let $\mathbb{X}$ and $\mathbb{Y}$ be Hausdorff topological spaces and $u : \mathbb{X} \times \mathbb{Y} \times [0, 1] \to \mathbb{R}$ be a function. We introduce $\mathbb{K}(\mathbb{X})$ (resp. $\mathbb{K}(\mathbb{Y})$) as the collection of all compact sets in $\mathbb{X}$ (resp. $\mathbb{Y}$). Assume that $X : [0, 1] \rightrightarrows \mathbb{K}(\mathbb{X})$ and $Y : [0, 1] \rightrightarrows \mathbb{K}(\mathbb{Y})$ are nonempty correspondences. We say that $(x^\star(t), y^\star(t))$ is a *saddle point* of $u$ at some fixed $t \in [0, 1]$ if it satisfies that

$$\max_{x \in X(t)} u(x, y^\star(t), t) \leq u(x^\star(t), y^\star(t), t) \leq \min_{y \in Y(t)} u(x^\star(t), y, t). \tag{60}$$

It is well-known that the existence of the above saddle point $(x^\star(t), y^\star(t))$ implies that (see e.g. [15] Chapter 6. Proposition 1.2)

$$\max_{x \in X(t)} \min_{y \in Y(t)} u(x, y, t) = \min_{y \in Y(t)} \max_{x \in X(t)} u(x, y, t) = u(x^\star(t), y^\star(t), t). \tag{61}$$

Next, introduce the value function $v(t)$ for $t \in [0, 1]$ as $v(t) = \max_{x \in X(t)} \min_{y \in Y(t)} u(x, y, t)$. The existence of the saddle point $(x^\star(t), y^\star(t))$ further implies that there exist subsets $X^\star(t) \subseteq X(t)$, $Y^\star(t) \subset Y(t)$ such that ([15] Chapter 6. Proposition 1.4)

$$u(x, y, t) = u(x^\star(t), y^\star(t), t), \ \forall (x, y) \in X^\star(t) \times Y^\star(t).$$

In this case, we can derive an envelope theorem at the saddle points [39].

**Theorem 8** (Theorem 5 in [39])**.** *Let $u$ and its derivative with respect to $t$, $u_t$, be continuous functions on $X \times Y \times [0, 1]$. Let $X, Y$ be continuous correspondences such that the existence of saddle point is guaranteed for all $t \in [0, 1]$. Then $v(t)$ is differentiable in $(0, 1)$, and*

$$v'(t) = \max_{x \in X(t)} \min_{y \in Y(t)} u_t(x, y, t) = \min_{y \in Y(t)} \max_{x \in X(t)} u_t(x, y, t), \ \forall t \in (0, 1).$$

Using Theorem 8, we are able to develop the stronger version of Theorem 3.

**Theorem 9.** *Let $\boldsymbol{\mu} \in \Lambda$, $\epsilon \in (0, \kappa E)$. For any $r \in (0, 1)$, and $\boldsymbol{\omega} \in \overset{\circ}{\Sigma}$, if another parameter $\boldsymbol{\pi} \in \Lambda$ satisfies that $\|\boldsymbol{\pi} - \boldsymbol{\mu}\|_\infty \leq \frac{\epsilon}{2E}$, then under Assumptions 1 and 3, we get:*

$$\left| \max_{z \in \Sigma} \min_{h \in H_{F_{\boldsymbol{\pi}}}} \langle \boldsymbol{z} - \boldsymbol{\omega}, h \rangle - \max_{z \in \Sigma} \min_{h \in H_{F_{\boldsymbol{\mu}}}} \langle \boldsymbol{z} - \boldsymbol{\omega}, h \rangle \right| \leq \frac{\epsilon}{2}, \ \forall (\boldsymbol{\omega}, r) \in \overset{\circ}{\Sigma} \times (0, 1), \tag{62}$$

*and*

$$\left| \min_{h \in H_{F_{\boldsymbol{\pi}}}} \langle \boldsymbol{z} - \boldsymbol{\omega}, h \rangle - \min_{h \in H_{F_{\boldsymbol{\mu}}}} \langle \boldsymbol{z} - \boldsymbol{\omega}, h \rangle \right| \leq \frac{\epsilon}{2}, \ \forall (\boldsymbol{z}, \boldsymbol{\omega}, r) \in \Sigma \times \overset{\circ}{\Sigma} \times (0, 1). \tag{63}$$

*Proof.* We prove (62). (63) will hold for similar reasons as discussed later. Fix $\boldsymbol{\mu} \in \Lambda$, $r \in (0, 1)$ and $\boldsymbol{\omega} \in \overset{\circ}{\Sigma}$.

### (i) Verifying the conditions of Theorem 8.

We apply Theorem 8 with $\mathbb{X} = \Sigma - \boldsymbol{\omega} = \{\boldsymbol{x} \in \mathbb{R}^K : \exists \boldsymbol{z} \in \Sigma \text{ s.t } \boldsymbol{x} = \boldsymbol{z} - \boldsymbol{\omega}\}$ and $\mathbb{Y} = \Sigma(\mathcal{J}_{i^\star(\boldsymbol{\mu})})$, which denotes the $\left| \mathcal{J}_{i^\star(\boldsymbol{\mu})} \right| - 1$-simplex. As $\|\boldsymbol{\pi} - \boldsymbol{\mu}\|_\infty \leq \frac{\epsilon}{2E} < \kappa$, Assumption 3 holds. Thus,

$\boldsymbol{\pi} \in \mathcal{S}_{i^\star(\boldsymbol{\mu})}$, we then define $u$ and its derivative with respect to $t$ as:

$$u(\boldsymbol{x}, \boldsymbol{y}, t) = \sum_k \sum_j x_k y_j d(\tilde{\pi}(t)_k, \overline{\boldsymbol{\lambda}_j(\boldsymbol{\omega}, \tilde{\boldsymbol{\pi}}(t))}_k),$$

$$u_t(\boldsymbol{x}, \boldsymbol{y}, t) = \sum_k \sum_j x_k y_j \langle \nabla_{\tilde{\boldsymbol{\pi}}(t)} d(\tilde{\pi}(t)_k, \overline{\boldsymbol{\lambda}_j(\boldsymbol{\omega}, \tilde{\boldsymbol{\pi}}(t))}_k), \boldsymbol{\mu} - \boldsymbol{\pi} \rangle.$$

where $\tilde{\boldsymbol{\pi}}(t) = t\boldsymbol{\mu} + (1-t)\boldsymbol{\pi}$, for all $t \in [0, 1]$. Observe that both $u, u_t$ are continuous on $\mathbb{X} \times \mathbb{Y} \times [0, 1]$. Further define the correspondences

$$\begin{cases} X(t) = \mathbb{X} = \Sigma - \boldsymbol{\omega}, \\ Y(t) = \{ \boldsymbol{y} \in \Sigma(\mathcal{J}_{i^\star(\boldsymbol{\mu})}) : y_j = 0 \text{ if } f_j(\boldsymbol{\omega}, \tilde{\boldsymbol{\pi}}(t)) \geq F_{\tilde{\boldsymbol{\pi}}(t)}(\boldsymbol{\omega}) + r \}. \end{cases}$$

$X(t)$ is a constant so it is continuous. As for the continuity of $Y(t)$, the argument is similar to that used to prove the hemicontinuity of $H_{F_{\boldsymbol{\pi}}}(\boldsymbol{\omega}, r)$ (see the proof of Lemma 8). Since $\max_{x \in X(t)} \min_{y \in Y(t)} u(x, y, t)$ forms a zero-sum matrix game for any $t \in [0, 1]$, the saddle point always exist (von Neumann minimax theorem, see [52] chapter 20). Thus, the conditions of Theorem 8 are verified.

#### (ii) Applying mean value theorem.

Observe that

$$v(0) = \max_{\boldsymbol{x} \in \Sigma - \boldsymbol{\omega}} \min_{\boldsymbol{y} \in Y(0)} \sum_k \sum_j x_k y_j d(\pi_k, \overline{\boldsymbol{\lambda}_j(\boldsymbol{\omega}, \boldsymbol{\pi})}_k)$$

$$= \max_{\boldsymbol{x} \in \Sigma - \boldsymbol{\omega}} \min_{h \in H_{F_{\boldsymbol{\pi}}}(\boldsymbol{\omega}, r)} \langle \boldsymbol{x}, h \rangle = \max_{z \in \Sigma} \min_{h \in H_{F_{\boldsymbol{\pi}}}(\boldsymbol{\omega}, r)} \langle \boldsymbol{z} - \boldsymbol{\omega}, h \rangle. \tag{64}$$

Likewise, we have

$$v(1) = \max_{z \in \Sigma} \min_{h \in H_{F_{\boldsymbol{\mu}}}(\boldsymbol{\omega}, r)} \langle \boldsymbol{z} - \boldsymbol{\omega}, h \rangle. \tag{65}$$

Theorem 8 implies that $v(t)$ is differentiable and the mean value theorem yields that there exists $t_0 \in (0, 1)$ such that $v(1) - v(0) = \max_{\boldsymbol{x} \in X(t_0)} \min_{\boldsymbol{y} \in Y(t_0)} u_t(\boldsymbol{x}, \boldsymbol{y}, t_0)$. Therefore,

$$\begin{aligned} |v(1) - v(0)| &= \left| \max_{\boldsymbol{x} \in X(t_0)} \min_{\boldsymbol{y} \in Y(t_0)} u_t(\boldsymbol{x}, \boldsymbol{y}, t_0) \right| \\ &\leq \max_{k,j} \left\| \nabla_{\tilde{\boldsymbol{\pi}}(t_0)} d(\tilde{\pi}(t_0)_k, \overline{\boldsymbol{\lambda}_j(\boldsymbol{\omega}, \tilde{\boldsymbol{\pi}}(t_0))}_k) \right\|_1 \|\boldsymbol{\pi} - \boldsymbol{\mu}\|_\infty \\ &\leq (E)(\frac{\epsilon}{2E}) \leq \frac{\epsilon}{2}, \end{aligned} \tag{66}$$

where the first inequality is Hölder inequality and the second inequality stems from Assumption 3. By substituting equations (64)-(65) into the left-hand side of the inequality (66), we deduce the inequality (62) claimed in the theorem.

As for the inequality (63), the argument will hold by replacing $X(t) = \{\boldsymbol{z}\}$.

$\square$

### N.3 Completing the non-asymptotic analysis

Based on Theorem 9 and Lemma 14 in this section, we can state the new concentration result that will replace Lemma 2 (Appendix I.3).

**Lemma 15.** *Under Assumptions 1 and 3, for any $\boldsymbol{\mu} \in \Lambda, \epsilon \in (0, \kappa E)$, under* FWS,

$$\sum_{T=1}^\infty \mathbb{P}_{\boldsymbol{\mu}} \left[ (\mathcal{E}_{1,\epsilon}(T) \cap \mathcal{E}_{2,\epsilon}(T))^c \right] \leq \sum_{k=1}^K \frac{34 e^K}{d(\mu_k - \frac{\epsilon}{2E}, \mu_k)^{\frac{19}{4}}} + \frac{34 e^K}{d(\mu_k + \frac{\epsilon}{2E}, \mu_k)^{\frac{19}{4}}}.$$

*Proof.* Replacing $\xi(\epsilon)$ by $\frac{\epsilon}{2E}$ in the proof of Lemma 2 (see Appendix J.1), we obtain the lemma (as in the proof of Lemma 2).

$\square$

**Proof of Theorem 2**

Plugging the inequality derived in Lemma 15 and (42) in (41), we conclude that

$$
\mathbb{E}\left[\tau\right] \leq \sum_{k=1}^{K} \left( \frac{34 e^K}{d(\mu_k - \frac{\epsilon}{2E}, \mu_k)^{\frac{19}{4}}} + \frac{34 e^K}{d(\mu_k + \frac{\epsilon}{2E}, \mu_k)^{\frac{19}{4}}} \right) + \left( \frac{32D + 3L}{\epsilon} \right)^{11}
$$

$$
+ (4K+1)^{11} + \max \left\{ c_1(\Lambda), \left( \frac{2}{\tilde{\epsilon}} \right)^{11} \right\}
$$

$$
+ \frac{1 + \tilde{\epsilon}}{F_{\boldsymbol{\mu}}(\boldsymbol{\omega}^\star(\boldsymbol{\mu})) - 6\epsilon} \left[ \log \left( \frac{(1 + \tilde{\epsilon}) c_2(\Lambda) e}{\delta(F_{\boldsymbol{\mu}}(\boldsymbol{\omega}^\star(\boldsymbol{\mu})) - 6\epsilon)} \right) + \log \log \left( \frac{(1 + \tilde{\epsilon}) c_2(\Lambda)}{\delta(F_{\boldsymbol{\mu}}(\boldsymbol{\omega}^\star(\boldsymbol{\mu})) - 6\epsilon)} \right) \right].
$$

This is the upper bound claimed in Theorem 2. □