# OpenReview forum: "Fast Pure Exploration via Frank-Wolfe"
_NeurIPS.cc/2021/Conference — NeurIPS 2021 Poster_

### Official Review · Reviewer_2Czm · 2021-07-12

**Rating:** 6
**Confidence:** 3

**Summary:**

This paper studies the best arm identification problem in the fixed confidence setting. Frank Wolfe algorithm is employed to solve the minimax problem for the optimal arm allocation. They establish the asymptotic optimality of their methods in a wide class of pure exploration settings. The sample complexity is empirically confirmed in the numerical simulations.

**Limitations And Societal Impact:**

This paper is well-written and self-contained. However, it would be great if the authors clarify the main challenges in more detail.

**Main Review:**

Concern on the originality :
The lower bound of MAB pure exploration is proved in  [21], and most existing papers solve a lower bound optimization which is introduced in equation (1) at each round t. Such algorithms achieves the asymptotically optimal, but solving (1) at every round is  computationally inefficient.
In this paper, instead of solving (1) in each round as in [21], they propose an online iterative method to converge the optimal allocation of arm pulls \omega^*.
This concept is already discussed in the work of [39], which performs an online lazy mirror ascent to solve a lower bound optimization (1). Their method can avoid computing (1) at every round since it is sufficient to compute at each steps only a subgradient to maximize the concave but not smooth objective function over the simplex.

The addition by this paper is to connect FW-algorithms with pure exploration problems for solving non-smooth functions. There is a surprising work of [42] that can be applied to solving non-smooth objective functions; this enables us to leverage the analysis of the Frank-Wolfe algorithm in pure exploration bandit problems.

To sum-up, it is not so clear that this paper’s contribution is significant. The drawback of the computational issue of [21] is already discussed in [39]. Best Challenger type algorithm is also not new, which was already introduced in [21].  Analysis for  non-smooth objective function by FW-algorithms is given by [42]. This paper combines those existing work’s ideas but it is not really clear which part is the novel developed and what is the difficulty to  leverage them.

[21] Aurélien Garivier and Emilie Kaufmann. Optimal best arm identification with fixed confidence. In Proc. of COLT, 2016.

[39] Pierre Ménard. Gradient ascent for active exploration in bandit problems. arXiv, 2019.

[42] Sathya N Ravi, Maxwell D Collins, and Vikas Singh. A deterministic nonsmooth frank wolfe algorithm with coreset guarantees. Informs Journal on Optimization, 2019.

============
Thank you for your feedback,  my concern is clearly addressed by author's response.


**Time Spent Reviewing:**

6,7 hours

---

> ### Author Response · Authors · 2021-08-08
> **Author response**
>
> Thank you for your comments. We will address them and revise our paper accordingly. Please find answers to your comments below. Reference numbers correspond to those provided in the supplementary material (long version of the paper).
>
> 1. Comparison with [39]. [39] presents a gradient ascent algorithm for pure exploration problems. It is close in spirit to ours as we mention in the introduction. As also noticed in [32], to make LMA work, we need to tune its learning rate as a function of the unknown parameter $\mu$ (which is impossible in practice). When using the learning rate proposed in [39], LMA is close to uniform sampling and has high sample complexity. BC has a much better performance than LMA even when applying a learning rate depending on $\mu$ (as reported in our paper). Finally, we have to say that we are not sure about the theoretical performance guarantees of LMA for any structure, e.g., for Lipschitz bandits.
>
> 2. Non-smooth FW algorithms. It is impossible to use the method of [42] for the generic pure exploration problems as there, the update requires the knowledge of the subgradient of $F_\mu$ on the neighborhood of the current allocation. As already argued in [8], computing such subgradients is most often impossible. Our FW update circumvents this difficulty thanks to two non-trivial observations (i) Proposition 1 and (ii) the construction of r-subdifferential spaces. To the best of our knowledge, these observations are novel.
>
> 3. Our contributions. Overall, we propose a new way to adapt the FW algorithm so that when used as a subroutine in BC, a pure exploration algorithm, BC has asymptotically optimal sample complexity for any kind of pure exploration problems in structured bandits. FW-based algorithms were mentioned in the literature earlier as a very efficient way of solving pure exploration problems, see e.g. [21] and [39]. We formalize and substantiate this observation.
> Importantly, we applied BC to different pure exploration problems, including BAI in linear bandits, and for the first time in the literature in BAI in Lipschitz bandits! \
> \
> Note that we do not claim that the lower bound on the sample complexity was a contribution of the paper since indeed, it can be established using standard technique. That is also why we present it in the introduction.

---

### Official Review · Reviewer_8vBR · 2021-07-13

**Rating:** 6
**Confidence:** 4

**Summary:**

The paper studies pure exploration in the fixed confidence setting for generic stochastic bandits. They develop a new algorithm based on Frank-Wolfe, give conditions under which it matches the lower bound (asymptotically as delta goes to 0), and empirically evaluate it.

**Limitations And Societal Impact:**

I forsee no potential negative societal impact.

**Main Review:**

The paper argues that most prior work solves the optimization problem (1) at each round, which can be computationally expensive. To address this, they give an algorithm that just takes one step of Frank-Wolfe at each round instead.

I found the assumptions 2 and 3 difficult to interpret. I think it would be useful to give more explanation in the main text on when these conditions hold, especially assumption 2. Given that prior work (e.g., [9]) has already solved this problem in such a generic setting, it would be useful to understand how restrictive assumption 2 is to compare to prior work. Are these assumptions strong enough to make the sample complexity result substantially weaker than prior work? Or are these assumptions relatively benign?

The authors say in lines 100-106 that the work of [29] is quite similar to the current work, except that LMA does worse empirically. Is this the only advantage of the current paper, superior empirical performance? I think it is important to distinguish the current paper from prior work more clearly so that the contribution is clear.

The authors write in line 56 that inner optimization problem in (1) can be hard. This was a bit unclear to me. Do they mean it is NP-hard or just intensive? Could examples be provided?

Building on this, if the main advantage of the current work is being computationally lightweight, it would be useful to demonstrate this in experiments or examples of settings that are difficult for prior algorithms. Are there settings where the proposed algorithm does well, but existing algorithms do not scale to them? Essentially, are there problems that the current algorithm solves, but prior algorithms do not?


After the rebuttal: thank you for the helpful clarification. I increased my score.


**Time Spent Reviewing:**

3

---

> ### Author Response · Authors · 2021-08-08
> **Author response**
>
> Thank you for your comments. We will address them and revise our paper accordingly. Please find answers to your comments below. Reference numbers correspond to those provided in the main document (short version of the paper).
>
> 1. Our contributions. The main contribution of the paper is to adapt the FW algorithm so that when used as a subroutine in BC, our pure exploration algorithm, BC has asymptotically optimal sample complexity for any kind of pure exploration problems in structured bandits. FW-based algorithms were mentioned in the literature earlier as a very efficient way of solving pure exploration problems, see e.g. [13] and [29]. We formalize and substantiate this observation.\
> Importantly, we applied BC to different pure exploration problems, including BAI in linear bandits, and for the first time in the literature, BAI in Lipschitz bandits! (Note that [8] and [29] just applied their algorithms to very simple unstructured bandit problems). \
> Next, we comment on the differences between our submission and [8] and [29]. The contributions presented in these papers are inspiring. Nevertheless, we believe that our submission brings interesting new insights and results.
>
> 2. Comparison with [8]. We are unsure about the applicability of the algorithm proposed in [8] to any kind of structure. The authors of [8] themselves mention in [9] that an application of the algorithm in [8] does not work for linear bandits (see Page 2 right column of [9] -- Our guess is that the analysis of [8] would require the use of more involved concentration results than those presented in [8]). The authors of [19] also noticed that the algorithm of [8] is difficult to apply to combinatorial bandits. The algorithm proposed in [8] requires two zero-regret algorithms as subroutines, and it is not clear to us how to implement such algorithms for generic structure. Note also that the analysis of [8] requires that the mean rewards are bounded (Assumption 2 in [8]). Importantly, our algorithm, BC, outperforms the algorithm proposed in [8] in all problems we tested. Finally, note that [8] only evaluated their algorithms on simple unstructured bandit problems.
>
> 3. Comparison with [29]. [29] presents a gradient ascent algorithm for pure exploration problems. It is close in spirit to ours as we mention in the introduction. As also noticed in [23], to make LMA work, we need to tune its learning rate as a function of the unknown parameter $\mu$ (which is impossible in practice). When using the learning rate proposed in [29], LMA is close to uniform sampling and has high sample complexity. BC has a better performance than LMA even when applying a learning rate depending on $\mu$ (as reported in our paper). Finally, we have to say that we are not sure about the theoretical performance guarantees of LMA for any structure, e.g., for Lipschitz bandits.\
> As a final remark, note that [8] and [29] only assess the performance of their algorithms for simple unstructured bandit problems.
>
> 4. About Assumption 2 (A2). A2 is in general needed for the analysis of FW algorithms. Notice that A2.(i) is weaker than Assumption 1 in LMA [29] and Assumption 4 in [8]. Moreover, A2.(ii) is weaker than the standard assumption used in the analysis of FW, see e.g. [17]. As explained in Appendix C, A2 holds for most relevant pure exploration problems (the sufficient condition for A2 that we provide in Appendix C is very easy to check). Please finally observe that in [8, 29], the authors only verify their assumptions on easy-to-solve problems, actually only in unstructured bandits, while we checked A2 in unstructured bandits, linear, Lipschitz, and dueling bandits.
>
> 5. About Assumption 3. As described in the proof sketch of Theorem 1 (Section 4.3), we need to ensure that the functions are continuous in $\mu$. To conduct a non-asymptotic analysis of the sample complexity (i.e., Theorem 2), we need even stronger continuity results (at least Lipschitz continuity). The technical Assumption 3 allows us to upper bound the gradients of the functions $F_\mu$ and of that defined in (11) (as it turns out, these gradients involve the gradient of the function d, hence Assumption 3). Usually, Assumption 3 is easy to verify, and all the problems discussed in Appendix D, E, F, G satisfy this assumption.
>
> 6. Hardness of the inner optimization problem in (1). The hardness of this problem is difficult to evaluate and depends on the structure. We may imagine structures where it becomes intractable. Even for the case of Lipschitz bandits, solving this problem is not easy. We will look for yet harder structures. We note that to circumvent the difficulty of solving the inner optimization problem (1), we present in Proposition 1 an interesting new intermediate result that allows us to simplify the optimization problem leading to the optimal allocation. This result is exploited in the design of BC.
>
> 7. Does BC solve problems that other algorithms cannot solve? As we discuss above, BC is the first algorithm that works well for Lipschitz bandits (LMA needs a learning rate that depends on $\mu$).

---

### Official Review · Reviewer_ABte · 2021-07-16

**Rating:** 7
**Confidence:** 4

**Summary:**

This paper studies the pure exploration problem for stochastic bandits. In particular, the authors focus on the fixed-confidence setting. The main contributions of the paper are the following:

- First, the authors propose a variant of the BC algorithm by Garivier and Kaufmann, 2016, where the sampling rule is based on a Frank-Wolfe-type algorithm adapted to non-smooth functions.
- Second, the authors show that the proposed algorithm is asymptotically optimal (in the sense that it matches the lower bound derived by Garivier and Kaufmann, 2016).
- Experimental illustration is given on various scenarios including both structured (like linear BAI and BAI for Lipschitz bandits) and unstructured BAI. All the experiments are against several existing baselines (though probably better to add some more past and recent baselines as mentioned in the 'Limitations and Societal Impact' section).


**Limitations And Societal Impact:**

Societal Impact: not applicable

Major questions:
- I think it could be helpful to compare the time complexity of the FW update against its counterpart in other algorithms (for example, you can compare the average execution time of different sampling rules), as in my opinion this is one important reason why we are interested in FW here.
- I am quite curious about the forced exploration phase, do you think it is possible to get rid of it?

Minor comments:
- What is LT-H?
- It would probably be better to include LinGapE (Xu et al., 2018) and RAGE (Fiez et al., 2019) as well in the experiments for linear BAI.
- Probably better to also include top two algorithms (Russo, 2016; Qin et al., 2019; Shang et al., 2020) in the experiments (in particular for unstructured bandits).
- It would probably be better to place the acronyms (like T-D, D-C, etc) directly into the plot captions.


**Main Review:**

Clarity: The paper is well written and the delivered message is clear.

Quality: I did not check the all the proofs in detail (54 pages is quite long honestly), but they seem to be sound in general. Proof-related questions are listed in the 'Limitations And Societal Impact' section.

Originality and significance: This paper proposed a variant of BC with a modified FW (adapted to non-smooth functions) that achieves the asymptotic optimality in a wide range of pure exploration scenarios. For me it is an interesting result and valuable contribution to the BAI community (at least for fixed-confidence BAI), since it has a wide range of practical usage thanks to FW and in the mean while avoids the $\beta$-tuning issue for top-two algorithms. The technical novelty is also generally sound (please see below for some technical comments).

Some technique-related comments:
- Line 240: What if $N_k(t) = 0$?
- Line 990: How do we know that $\pi = \hat{\mu}^{t-1}\in \Lambda$?
- Line 1012: $\mu$ needs to be in $\Lambda$ right?
- Proposition 3: $F$ is Lipschitz with respect to which norm?
- Line 1171: Are you sure about the last inequality $||z-x||_{\infty}^2$?
- Line 1252: $z_k(t)$ -> $z_k(s)$.
- Line 1277: It seems that $\Psi$ is exponential in the number of arms $K$? Could it be a problem when dealing with many arms?

Overall, I vote for accept for this paper. Please see the next section for major questions and minor comments.


**Time Spent Reviewing:**

10

---

> ### Author Response · Authors · 2021-08-08
> **Author response**
>
> Thank you for your insightful comments. We will address them and revise our paper accordingly. Please find answers to each of your comments below. Reference numbers correspond to those provided in the main document (short version of the paper).
>
>
> Major questions:
> 1. Complexity of the algorithm BC. In each round, BC solves a small LP with m variables where m is the dimension of the r-subdifferential space. We do not have guarantees on m but experiments suggest that it remains small – see Appendix D Fig. 1 where 40% of the time $m=1$ for unstructured bandits, and Appendix E Fig. 3 where m is almost always equal to 1 for linear bandits. For Lipschitz bandits, we re-run our experiments, and in about 70% of rounds $m=1$; we will add these results in the paper.\
> \
> In comparison, in the case of Best Arm Identification (BAI) in unstructured bandits, the TaS algorithm [13] and LMA [29] have a complexity equal to $O(K\log(\frac{1}{\epsilon})^2))$ (where $\epsilon$ is the precision of the binary search algorithm used to compute the update) and O(K) per round, respectively. For the gamification approach [8], the time complexity mainly depends on the two regret minimization algorithms used as subroutines. The paper [29] provides a precise account of the complexity of various baseline algorithms (see Appendix B in [29]). From the above discussion, it is difficult to compare the complexity of the various algorithms analytically. We will provide results from experiments: BC has a similar running time as TaS but worse than LMA.\
> \
> In the case of BAI in structured bandits, it is still difficult to compare the running time of various algorithms analytically.  However, BC is the only algorithm with strong theoretical and empirical performance guarantees and good running time. In linear bandits, all the presented algorithms are fast. LT-H [18] is the fastest but has a much higher sample complexity than BC; BC is faster than LinGame and LinGame-C [9]. In Lipschitz bandits, LMA [29] is faster than BC and BC is faster than TaS. But, as also noticed in [23], to make LMA work, we need to tune its learning rate as a function of the unknown parameter $\mu$ (which is impossible in practice). When using the learning rate proposed in [29], LMA is close to uniform sampling and has high sample complexity. In addition, we are not sure about its theoretical performance guarantees for Lipschitz bandits.
>
> 2. Avoiding the forced exploration phase. When the curvature remains bounded (e.g., in most threshold bandit problems), forced exploration can be avoided and upper confidence bounds as [5] can be used instead. When the curvature may become infinite, we need to exploit the specific structure of the problem to be able to remove the forced exploration phase. For example, for BAI in linear bandits, we only need to explore each dimension as done in [26].
>
>
> Minor comments:
>
> •	LT-H is defined in Appendix E L748, and stands for Lazy Track-and-Stop with a heuristic stopping rule introduced in [18] (see Appendix A.1 of NeurIPS version). For convenience, we will add this definition in the main paper.
>
> •	We will add LinGapE but omit RAGE since it is outperformed by LT-H [18].
>
> •	We will make the proposed modifications.
>
>
>
> Technique-related comments:
>
> •	Line 240: Then it should be defined as 0.
>
> •	Line 990: We will include $\{\hat{\mu}(t-1)\in \Lambda}$ into the good event.
>
> •	Line 1012: Yes
>
> •	Proposition 3: Infinity norm.
>
> •	Line 1171: $|| z − x ||_{\infty}^2$ should be removed
>
> •	Line 1277: Our upper bound in Theorem 2 is conservative because it is valid for any structure. We can leverage specific structures to derive tighter results. For example, for BAI in linear bandits, we only need to explore each dimension as done in [18] – and in turn we hope to remove the dependency in the number of arms.

---

### Official Review · Reviewer_5onQ · 2021-07-25

**Rating:** 7
**Confidence:** 4

**Summary:**

The authors propose a new framework for structured (and unstructured) pure exploration problems. They propose a new algorithm based on modified FW where the motivation comes from the BC algorithm from the TaS paper that can be interpreted as an FW step. They show the asymptotic optimality of the algorithm and provide a finite-time analysis. The empirical results on various bandit problems show the superiority of the proposed framework.


**Ethical Concerns:**

no concerns.

**Limitations And Societal Impact:**

I don't see a negative societal impact.

**Main Review:**

high originality, mediocre quality (see my comments on experiments below), clarity is good, and the significance is above bar.

* The main contribution is very clear: they provide a new framework and recipe for solving generic pure exploration problems. I think the authors did a nice job of interpreting best challenger from Garivier & Kauffman as a frank-wolfe step and find a way to do it right.
* I feel the experiments are quite biased in that the baseline methods are missing a few popular algorithms including RAGE, LinGapE, and ALBA.  For the unstructured problem, lil ucb and LUCB are missing. Thus, I am not  convinced about its performance.
* I was excited to read the abstract that we take only one step in each iteration since that would mean the computational complexity is much improved. However, it seems to require solving the linear program. Could the authors discuss the time complexity of the algorithm? For pure exploration problems, It seems to be worse than TaS.
* Please add a discussion on the significance of Theorem 2 in the main content of the paper. I guess the first term looks fine, but how should readers take \Psi() and T_{\epsilon,L}? Does the authors think the exponential dependence on K should be improvable?

minor points
* L136: "close the boundary" => close to the boundeary?
* L151-152: "the optimism-in-front-of-uncertainty ": I've seen it called 'optimism in the face of uncertainty' more frequently.
* L680: did you mean [21] rather than [29] here? there are other places like L698.
* Okay, I am struggling with finding the precise definition of $\Lambda$​. So, $\Lambda$​ is the set of possible instances for which there is only a unique best arm? Then, it is not a convex set? I ask this to understand the if statement in Algorithm 1 that has $\hat \mu(t-1) \not \in \Lambda$​​ . I could guess but I would appreciate if the definition of $\Lambda$ is more explicit.
* Assumption 3: is \forall symbol missing in front of $\pi \in \mathcal{S}_{i^*(\mu)}$?
* L290: con() was defined as con{}.

----
(after rebuttal)
I am mostly satisfied with the rebuttal. I now remember that lil'ucb was evaluated in "Best-arm identification algorithms for multi-armed bandits in the fixed confidence setting" and was empirically shown to be worse than LUCB in practical regimes anyways. But it would be great to have that remark in the final version.

For Assumption 3, I understand it now. I rephrase and expand the sentence to make it easier to read.

**Time Spent Reviewing:**

6

---

> ### Author Response · Authors · 2021-08-08
> **Author response**
>
> Thank you for your insightful comments. We will address them and revise our paper accordingly. Please find answers to each of your comments below. Reference numbers correspond to those provided in the main document (short version of the paper).
>
> 1. About our experiments for linear bandits. The idea was to compare the performance of our algorithm, BC, to that of algorithms with the same theoretical guarantees, namely with instance-specific asymptotically optimal sample complexity. That is why we compared BC to algorithms in [9] and [18] only. This choice was also motivated by results presented in [10] and [18]: in [10], the authors evaluate the performance of RAGE, LinGapE, and ALBA, and show that RAGE outperforms the others; whereas in [18], LT-H is shown to be better than RAGE (hence we kept LT-H and omitted the other algorithms). We are currently running all algorithms and plan to report their performance in the paper.
>
> 2. Experiments for unstructured bandits. In [13], the authors compare their algorithm, TaS, to LUCB, and show that TaS exhibits better performance. That is why we omitted LUCB. About lil’ UCB, almost all existing works do not consider this algorithm, but we can add it.
>
> 3. Experiments in general and their role in the paper. The main contribution of the paper is to provide a generic algorithm with asymptotically optimal instance-specific sample complexity, whatever the underlying structure and the pure exploration task are. We illustrate the versatility of our approach on several difficult structured bandits (linear bandits and for the first time in the literature, Lipschitz bandits). Note that the two other papers [8] and [29] with the same objective as our paper just evaluate numerically their algorithms on easy-to-solve problems, actually only in unstructured bandits!
>
> 4. Implementation complexity of BC. Indeed, BC solves a small LP in each round with m variables where m is the dimension of the r-subdifferential space. We do not have guarantees on m but experiments suggest that it remains small – see Appendix D Fig. 1 where 40% of the time, $m=1$ for unstructured bandits, and Appendix E Fig. 3 where $m$ is almost always equal to 1 in linear bandits. For Lipschitz bandits, we re-run our experiments, and in about 70% of rounds $m=1$; we will add these results in the paper. Hence, BC is very fast for structured bandit problems, whereas for unstructured bandits, its running time is close to and sometimes better than that of TaS (TaS has a complexity equal to $O(K\log(\frac{1}{\epsilon})^2))$, where epsilon is the precision of the binary search algorithm used to compute the update). We will include the empirical running time of all algorithms in the paper.
>
>
> 5. About Theorem 2. To the best of our knowledge, the only attempts to devise algorithms with both (i) asymptotic instance-specific optimal sample complexity and (ii) non-asymptotic sample complexity guarantees for generic pure exploration problems are reported in [8] and in the present submission. Our analysis is valid for any structure; this versatility does not seem to be shared by [8], since as mentioned in [9] (the same authors as in [8]) and [19], the analysis in [8] is not applicable to linear and combinatorial bandits. As you noticed, the second term in our sample complexity upper bound scales exponentially in $K$, but it is the result of the application of crude concentration inequalities. By exploiting specific structures, we believe we can improve this dependency.
>
> Minor points:
> Definition of $\Lambda$. We will clarify. It is defined around L30. This is the set of all possible parameters. The set captures the structure of the problem if any; it is a usual way of encoding the known structural properties of the problem.
>
> L136, L151-L152, L290: Thanks for noting these. We will modify accordingly.
>
> Assumption 3: No, $\pi$ is some vector in $\mathbb{R}^K$ satisfying L352.

---

### Decision · Program_Chairs · 2021-09-28

**Decision:**

Accept (Poster)

**Comment:**

This paper has initially received mixed reviews, with some reviewers expressing concerns about the significance of the contribution. These concerns were sufficiently addressed by the authors' response, and, after a short discussion, all reviewers agreed that the paper is strong enough for publication at the conference. I concur with this assessment and recommend that this paper be accepted.

**Consistency Experiment:**

NeurIPS has a long history of experimentation. In 2014, NeurIPS ran an experiment in which 10% of submissions were reviewed by two independent committees to quantify the randomness in the review process. This year, we repeated a variant of this experiment to see how the quality of the review process has changed over time.  This paper was part of the experiment and was therefore assigned to two committees (consisting of reviewers, an Area Chair, and a Senior Area Chair) that reached independent decisions.  If both committees made the same recommendation, this recommendation was followed. If a single committee recommended acceptance, the paper was accepted (with the exception of a few cases in which the other committee identified what we considered a fatal flaw, e.g., an error in a key result).

Both committees reached the same decision: **Accept (Poster)**

The other committee assigned to the paper recommended **Accept (Poster)**.  You can find the other set of reviews, along with any follow up discussion with the authors here:
https://openreview.net/forum?id=cD2Ls4qXTc